# Multiple social encounters can eliminate Crozier's paradox and stabilise genetic kin recognition

Thomas W. Scott [1✉], Alan Grafen [1,2] & Stuart A. West [1,2]

Crozier's paradox suggests that genetic kin recognition will not be evolutionarily stable. The problem is that more common tags (markers) are more likely to be recognised and helped. This causes common tags to increase in frequency, and hence eliminates the genetic variability that is required for genetic kin recognition. It has therefore been assumed that genetic kin recognition can only be stable if there is some other factor maintaining tag diversity, such as the advantage of rare alleles in host-parasite interactions. We show that allowing for multiple social encounters before each social interaction can eliminate Crozier's paradox, because it allows individuals with rare tags to find others with the same tag. We also show that rare tags are better indicators of relatedness, and hence better at helping individuals avoid interactions with non-cooperative cheats. Consequently, genetic kin recognition provides an advantage to rare tags that maintains tag diversity, and stabilises itself.

[1] Department of Zoology, University of Oxford, Oxford OX1 3SZ, UK. [2] These authors contributed equally: Alan Grafen, Stuart A. West.
✉email: thomas.scott@zoo.ox.ac.uk

K in selection theory predicts that, at all levels of biology, from bacteria to humans, individuals should preferentially cooperate with closer relatives[1]. Individuals are favoured to help relatives because they share genes, and so by helping a relative reproduce, an individual is still passing its genes to the next generation, just indirectly. Closer relatives are more likely to share genes, and so there is a greater indirect benefit from preferentially helping closer relatives. Individuals are therefore expected to evolve *kin discrimination*, which is the conditional helping of relatives that are identified (*kin recognition*) through either genetic or environmental cues[2].

It has become widely accepted that kin recognition via genetic cues is not usually evolutionarily stable[2–8]. The problem is that more common tags (markers) at the recognition locus are more likely to be recognised[8] (Fig. 1a). Consequently, individuals with more common tags are more likely to be helped, increasing their fitness. In contrast, individuals with rare tags are less likely to be recognised and helped, reducing their relative fitness. This means that common tags will increase in frequency, and rare tags will decrease in frequency and be lost (Fig. 1b). Therefore, genetic kin recognition eliminates the genetic variability that is required for genetic kin recognition. This is Crozier's paradox—genetic kin recognition drives its own ruin[8].

Crozier's paradox provides the framework for the current understanding of kin recognition. State-of-the-art population genetic analyses have supported Crozier's argument, finding that genetic kin recognition is only stable under restrictive conditions[7,9]. It has therefore been assumed that genetic kin recognition will generally not be stable, and so kin discrimination is constrained to be based on environmental cues, such as a song learnt from relatives[4–6,8,10,11]. There are many examples of kin recognition based on environmental cues, especially in birds and mammals[11–14]. However, genetic kin recognition has also been observed in a range of animals, microorganisms and plants[15–22]. In these instances of genetic kin recognition, it has been assumed that an additional factor unrelated to social behaviour is maintaining variation at the tag locus, such as the advantage of rare MHC alleles in host-parasite interactions[4–8].

We hypothesise that Crozier's paradox can be eliminated by allowing for more natural forms of social behaviour, where individuals can encounter multiple individuals before taking part in a social interaction. Previous theory has assumed that, when an individual encounters a partner with a different tag, the opportunity to socially interact is wasted[4,7,8,23–26]. In contrast, in many animals where kin discrimination occurs, individuals can encounter multiple individuals within their group or larger social network before deciding who to help. For example, cooperatively breeding vertebrates can choose which group to help at, or which individuals to help in a group.

We show that, if multiple encounters occur before each interaction, then individuals with rare tags can find individuals with the same tag and receive as much help as individuals with common tags. In Fig. 1a this is represented by the orange birds finding and pairing up with other orange birds. Technically, this means that common tags do not increase in frequency, as assumed by Crozier's paradox, and so genetic variability is not eliminated at the recognition locus. In addition, we show that rare tags are better indicators of relatedness, and hence better at helping individuals avoid interactions with non-cooperative cheats. This means that the process of genetic kin recognition provides an advantage to rare tags that maintains tag diversity, and stabilises itself.

## Results and discussion

**Multiple encounters**. We model a scenario where individuals can potentially encounter several other individuals before settling on one to potentially help. We assume an infinite population of haploids, partitioned into an infinite number of groups (infinite island model). We ignore stochastic variation in the genetic composition of groups, which is reasonable if there are a large number of individuals ($N$) in each group. Each individual has a recognition allele (tag). The maximum number of tags that may simultaneously segregate in the population (genetic constraint) is given by $L_{max}$.

Individuals can potentially have many social encounters before committing to a given social interaction. Each generation, each individual encounters a random member of its group. If an individual shares a tag with its partner, it interacts and potentially helps—the social encounter becomes a social interaction. In contrast, if an individual does not share a tag with its partner, what happens depends upon the encounter parameter, $\alpha$. With a probability $\alpha$, an individual with a tag-mismatched partner will abandon that partner and re-associate for a new social encounter, with a new individual drawn at random from its group (Fig. 2a). With a probability $1-\alpha$, an individual with a tag-mismatched partner remains with that partner, but it does not interact (the opportunity to socially interact is wasted; Fig. 2a). Each time an individual abandons a partner and re-associates for a new social encounter, it pays a fecundity cost of $c_{search}$.

The encounter parameter, $\alpha$, puts a form of individual agency into the theory, by allowing individuals to search for another individual with the same tag. When $\alpha = 1$, individuals are free to have encounters with all the other individuals in their group, if need be, to find a tag-matched individual to interact with. In this case, individuals with a rare tag will keep searching until they encounter an individual with the same tag. At the other extreme, when $\alpha = 0$, any individual who does not encounter a tag-matched individual on its first try does not get to initiate a social interaction. Previous theory has implicitly considered the scenario where $\alpha = 0$[7,23–26].

We assume that, when an individual encounters a partner with the same tag, it interacts and potentially helps. Whether an individual helps depends upon its allele at the helping (trait) locus. Individuals with the 'conditional helping' allele will help, paying a fecundity cost of $c$ to give a benefit of $b$ to their social partner. Individuals with the 'defect' allele do not help. We assume that selection is weak (low magnitude of $b$ and $c$).

After social interactions have taken place, haploid individuals produce a very large number of gametes, before dying, where an individual's fecundity is given by how well it fared in social interactions and its investment in partner search. Each gamete

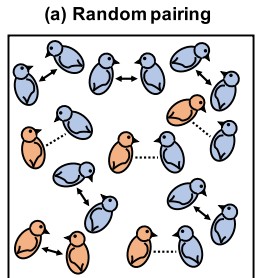
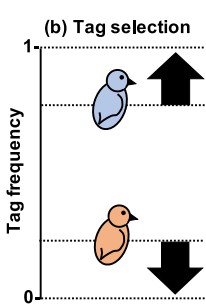

**(a) Random pairing**

**(b) Tag selection**

Tag frequency

**Fig. 1 Crozier's paradox. a** Birds with a more common genetic tag (blue) are more likely to encounter birds with the same tag, compared to birds with a less common tag (orange). Consequently, blue birds are more likely to be recognised and helped (arrows). **b** The blue tag will increase in frequency, while the orange tag will decrease in frequency and be eliminated (positive frequency dependence). Bird cartoons adapted from Levin, Caro, Griffin & West, *Evolution Letters* (ref. [59]), Creative Commons (https://creativecommons.org/licenses/by/3.0/).

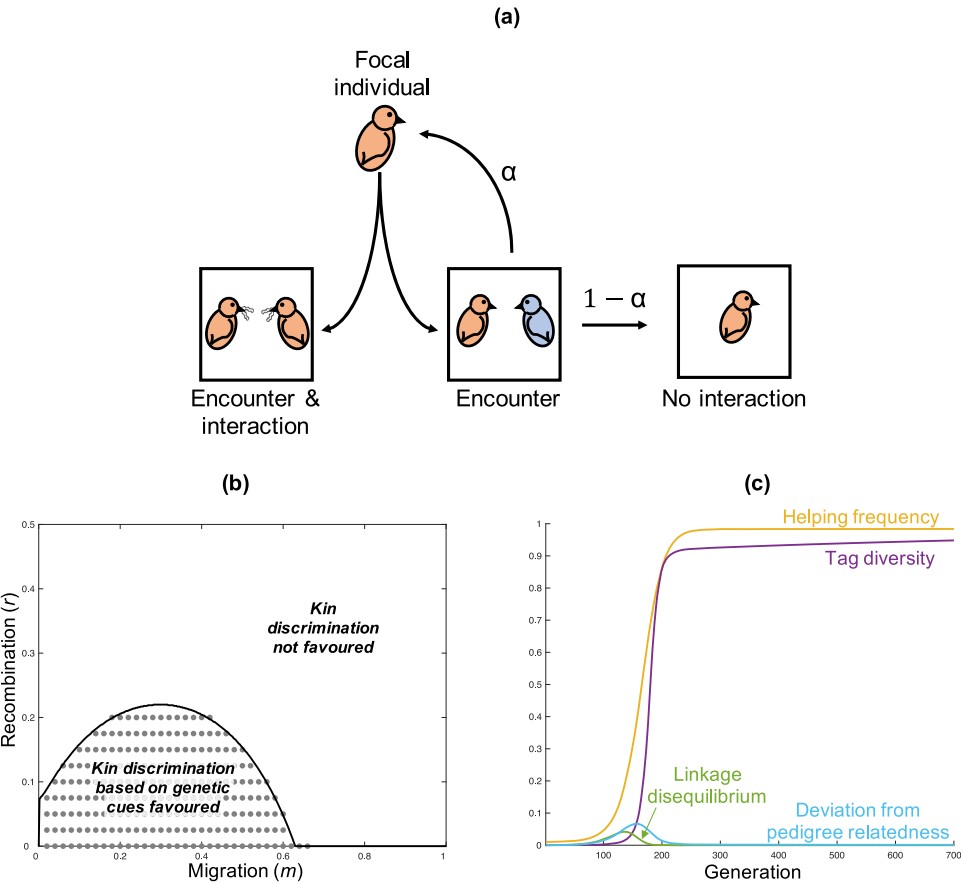

**Fig. 2 Stable genetic kin recognition. a** Social encounters and social interactions. If the focal individual encounters a tag-matched individual (both orange), it socially interacts. Conversely, if the focal individual encounters a tag-mismatched individual (one orange; one blue), the focal individual may encounter a new partner ($\alpha$), or forgo the social search ($1-\alpha$). Higher values of the encounter parameter ($\alpha$) correspond to individuals having more encounters to find a matching partner. During an interaction with a (tag-matched) partner, the focal individual may help or not (defect), depending upon its allele at the trait locus. **b** We plot the results of our population genetic island model when encounters are unrestricted and uncostly ($\alpha=1$ & $c_{search}=0$). The area under the solid line shows where kin discrimination is favoured by kin selection (Eq. 1 satisfied). The data points show parameter combinations where kin discrimination based on genetic cues is stable. These two areas match—whenever kin discrimination is favoured, sufficient tag diversity is maintained to allow genetic kin recognition. We assumed $\mu_{Trait}=0.001$, $b=0.3$, $c=0.1$, $L_{max}=100$, $\alpha=1$, $c_{search}=0$, $N=30$. **c** An illustrative single trial from panel B with: $r=0.08$; $m=0.3$. Rare tags become statistically associated with helping (linkage disequilibrium increases), which increases relatedness at the trait locus above that expected from pedigree. Consequently, rare tags and helping increase in frequency. As rare tags increase in frequency, they lose their statistical association with helping (linkage disequilibrium decreases), and relatedness at the trait locus converges on that expected from pedigree (see the sections 'Derivation of linkage disequilibrium' and 'Definitions of the four outputs plotted in Fig. 2c'; Supplementary Discussion 3). Bird cartoons adapted from Levin, Caro, Griffin & West, *Evolution Letters* (ref. [59]), Creative Commons (https://creativecommons.org/licenses/by/3.0/).

has a $1-m$ probability of staying in its native group, and a $m$ probability of emigrating to a different, randomly chosen group. Then, gametes fuse randomly within groups to produce diploid zygotes, and this is followed immediately by meiosis, with recombination between tag and trait loci occurring with probability $r$, and mutation at the trait locus occurring with probability $\mu_{Trait}$. We do not allow mutation at the tag locus, because we want to determine when selection can maintain tag diversity. Finally, $N$ haploid adults are sampled randomly from the haploid juveniles to provide the members of the group in the next generation (local competition), which completes the lifecycle. We assume that conditional helping and tag diversity are initially low, then iterate the lifecycle to find the equilibrium frequency of the conditional helping allele and the equilibrium tag diversity.

**Stable genetic kin recognition**. We find, in contrast to Crozier's prediction, and previous theory, that genetic kin recognition can be maintained in a relatively large area of parameter space

(Fig. 2b). Kin discrimination based on genetic cues (tags) is favoured when two conditions are met. First, kin discrimination must be favoured by kin selection. By this, we mean that conditional helping (help if matching tag) must have a higher fitness payoff than both defection (never help) and indiscriminate helping (always help, irrespective of tag). Second, rare tags must be maintained in the population, so that there is sufficient genetic diversity at the tag locus to allow genetic kin recognition.

*Kin discrimination.* Examining the first condition, kin discrimination is favoured by kin selection when:

$$R_{tag}b - c - (b-c)R_{comp} > 0, \qquad (1)$$

where $R_{tag}$ is the relatedness between actors and their (tag-matched) social interactants, and $R_{comp}$ is the relatedness between actors and the individuals who are displaced by competition (derived in 'Individual-level analysis (finding the right area of parameter space)')[27–30]. Here, relatedness technically means genetic similarity at the trait locus, but at evolutionary

equilibrium, this will usually be equal to the probability that individuals share common ancestry (pedigree/genealogical relatedness; e.g., 1/2 for full siblings, 1/8 for cousins; Supplementary Discussion 3)[1,29,31].

Equation 1 is a form of Hamilton's rule, showing that altruistic helping based on kin discrimination is more likely to be favoured if: helping is cheap to actors (lower $c$) and beneficial to recipients (higher $b$); individuals reside in groups with a high variance in relatedness, so that there are highly related individuals to help (high $R_{tag}$), and poorly related individuals to avoid helping (low $R_{comp}$)[1,27,32–35]. If this Hamilton's rule condition is not satisfied, defection is favoured.

*Maintaining tag diversity.* Examining the second condition, we found that, when individuals can search freely for social partners at no cost ($\alpha = 1$ & $c_{search} = 0$), tag diversity is maintained for the same area of parameter space where kin discrimination is favoured by kin selection (Fig. 2b; see 'Model construction and analysis'). This means that Eq. 1 also predicts when genetic kin recognition will be stable. Our result contrasts with Crozier's prediction, where tag diversity is lost, meaning genetic kin recognition is not stable.

In our model, genetic diversity at the tag locus is maintained by coevolution between helping and kin recognition[31]. As tags become more common, they will become less useful cues of the individual's common ancestry (pedigree relatedness; Supplementary Discussion 3), and so kin selection is less likely to favour the helping of tag-matched individuals (Fig. 3). Consequently, defection can invade at common tags (Fig. 3). In contrast, rare tags will be good indicators of relatedness, and so kin selection will favour the helping of tag-matched individuals. This means that rare tags cannot be invaded by defectors. Technically, a statistical association between genes for helping and rare tags builds up (linkage disequilibrium; Fig. 3). Crozier's original statement of the paradox did not permit defectors, meaning this coevolution between helping and kin recognition could not be captured[8,31]. More recent models have permitted defectors[5,7,23–26].

The consequence of this coevolution is that individuals with rare tags will have a greater average payoff from social interactions, meaning rare tags increase in frequency, maintaining tag diversity (negative frequency dependence)[31]. This prediction is in the opposite direction to Crozier's paradox, where common tags were favoured, except under restrictive conditions (positive frequency dependence)[4,7,8,23–26]. Our conclusions still tended to hold when we relaxed the assumptions of infinite population size and weak selection, and allowed the genetic composition of groups to vary stochastically (see 'Finite population (agent-based) simulation')[9,36,37].

**Encounter rate and search cost.** As the encounter parameter ($\alpha$) decreases, or the search cost parameter ($c_{search}$) increases, the area where genetic kin recognition is stable is reduced (Figs. 4 and 5). When $\alpha = 1$ & $c_{search} = 0$, there is negligible cost to having a rare tag, because individuals have multiple uncostly encounters until they find another individual with a matching tag. As $\alpha$ decreases, individuals with rare tags become relatively less likely to find an individual with a matching tag, favouring common tags, as suggested by Crozier. As $c_{search}$ increases, individuals with rare tags incur a relatively higher search cost on their way to finding another individual with a matching tag, also favouring common tags.

As the encounter parameter ($\alpha$) decreases, there is a decrease in the likelihood that genetic kin recognition is stable (Fig. 4c). Consequently, a high value of this encounter parameter ($\alpha$) is

required for genetic kin recognition to evolve. At this point, we need to think about what our encounter parameter ($\alpha$) is capturing. Our use of the encounter parameter ($\alpha$) assumes that individuals search randomly and with replacement, to make our model mathematically tractable. In nature, the search for potential partners could be more efficient, by focusing on individuals not previously encountered and by searching where relatives are more likely to be encountered, based upon environmental or spatial cues. Our model allows us to conceptually capture these more realistic scenarios if we think about how the encounter parameter ($\alpha$) determines the likelihood with which individuals can find another individual with the same tag. As $\alpha$ decreases, individuals become less likely to encounter another individual with the same tag, especially if the individual is using a rare tag (Fig. 4d).

Different species would correspond to different likelihoods of being able to find another individual with the same tag, and hence different values of $\alpha$. At one extreme, in many cooperatively breeding birds and mammals, individuals are likely to be able to find another individual with the same tag to interact with. This corresponds to a high $\alpha$, possibly even $\alpha \approx 1$. The reason for this is that individual animals: (a) can move around and choose who to help; and (b) live in family groups, within spatially structured populations, and so will encounter close relatives, who are likely to share the same tag. The probability of finding an individual with the same tag could be increased by a number of 'nonrandom' behaviours, such as using spatial or environmental cues to streamline the search.

In other organisms, such as bacteria, limited dispersal can still make encounters with relatives likely, but individuals have less ability to move around and choose who to interact with, and so $\alpha$ could be lower. When searching for partners is relatively cheap (low $c_{search}$), and individuals are using a limitingly rare tag, the probability of social interactions often needs to drop significantly below 1.0 before genetic kin recognition is likely to be less favoured (Fig. 4e). Therefore, although the stability of genetic kin recognition is susceptible to a drop off in the mathematical encounter parameter ($\alpha$), a biological interpretation of our mathematical parameter—as a proxy for the probability of socially interacting—implies that genetic kin recognition could evolve relatively permissively, as long as the risk of foregoing social interactions whilst using a rare tag is relatively low.

In addition, the influence of a decrease in the encounter parameter ($\alpha$) is reduced by strong selection. Our theoretical results were derived for the case where the strength of selection on social behaviour, captured by the magnitude of $c$ and $b$, is low (weak selection). We focused on the weak selection case because it is the most likely scenario in animals, where behaviours are generally underpinned by many genes of small effect. However, selection on social behaviour may be stronger in bacteria and other microorganisms, where one or a few genes of large effect may underpin social behaviours like the production of public goods such as iron-scavenging molecules (siderophores)[38]. In the case where selection is strong: genetic kin recognition is sometimes stable even when there is no chance for multiple encounters ($\alpha = 0$)[7]. Although, an increase in the encounter parameter ($\alpha$) still increases the strength of balancing selection on tags, and therefore increases the likelihood that genetic kin recognition is stable (Fig. 4f; see 'Finite population (agent-based) simulation').

The search cost ($c_{search}$) is incurred every time an individual abandons a partner and re-associates for a new social encounter. Consequently, the total search cost can be much higher than $c_{search}$. For limitingly rare tags under conditions of limitingly low

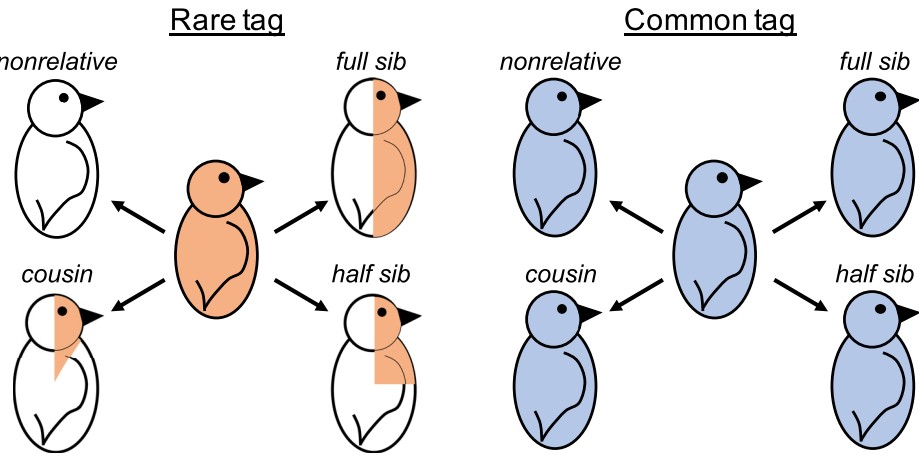

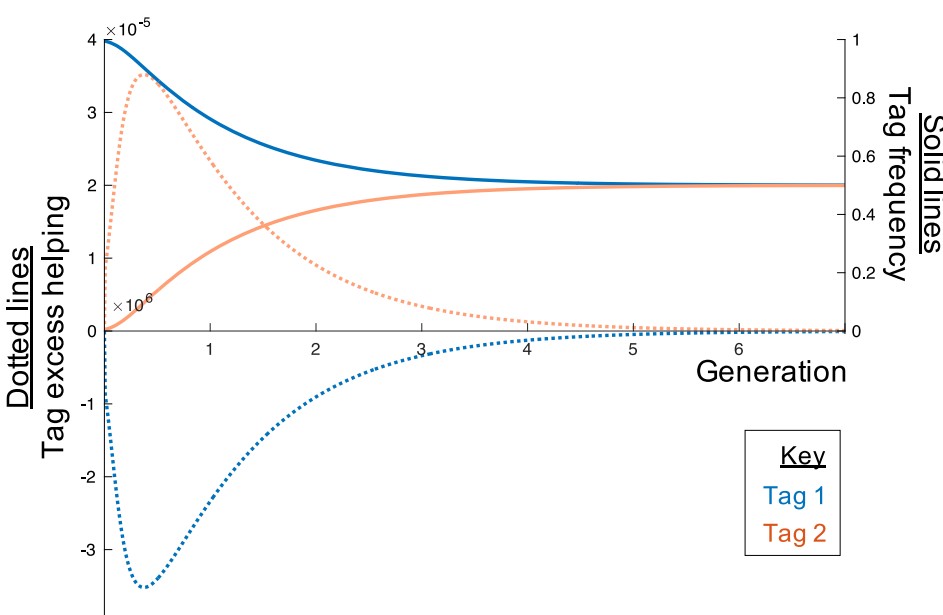

**Fig. 3 Advantage for rare tags. a** For birds with rare tags (orange), the probability of matching tags with someone (proportion shaded) is given by pedigree relatedness. For birds with common tags (blue), the probability of matching tags with someone (proportion shaded) is always high, regardless of pedigree relatedness. More common tags are therefore worse indicators of both relatednesses at the trait locus and pedigree relatedness. **b** An illustrative single trial, with two tags (blue and orange). The solid lines show tag frequency and the dotted lines show the probability above the population average that a social interaction results in help being received (tag excess helping) (see 'Definition of tag excess helping plotted in Fig. 3b'). The initially rare tag (orange) is a better indicator of relatedness, and as a result, it gains extra helpers (orange dotted line increases) relative to the initially common tag (blue dotted line decreases) (linkage disequilibrium). This causes the rare (orange) tag to increase in frequency (orange solid line increases), and the common (blue) tag to decrease in frequency (blue solid line decreases). As the orange tag increases in frequency, it loses its extra helpers (dotted lines tend to zero), and the fitness of the two tags converge. We assumed $L_{max} = 2$, $\mu_{Trait} = 0.0001$, $b = 0.015$, $c = 0.005$, $\alpha = 1$, $c_{search} = 0$, $m = 0.3$, $r = 0.1$, $N = 30$. Bird cartoons adapted from Levin, Caro, Griffin & West, *Evolution Letters* (ref. [59]), Creative Commons (https://creativecommons.org/licenses/by/3.0/).

relatedness, the total search cost will be $\frac{\alpha c_{search}}{1-\alpha}$. The search cost can capture many different things empirically, such as an increased predation risk whilst out searching for a social partner, or a loss of time that could be spent doing other things like foraging for food. Although an increased search cost reduces the area of parameter space for which genetic kin recognition is stable, this effect is relatively minor, and genetic kin recognition can still be stable with an appreciable partner search cost (Fig. 5). For instance, when $c_{search} = 0.009$, $c = 0.1$, $\alpha = 0.999$, genetic kin recognition is still stable in over half of the parameter space where kin discrimination is favoured by kin selection (for biologically reasonable parameter evaluations and weak social selection;

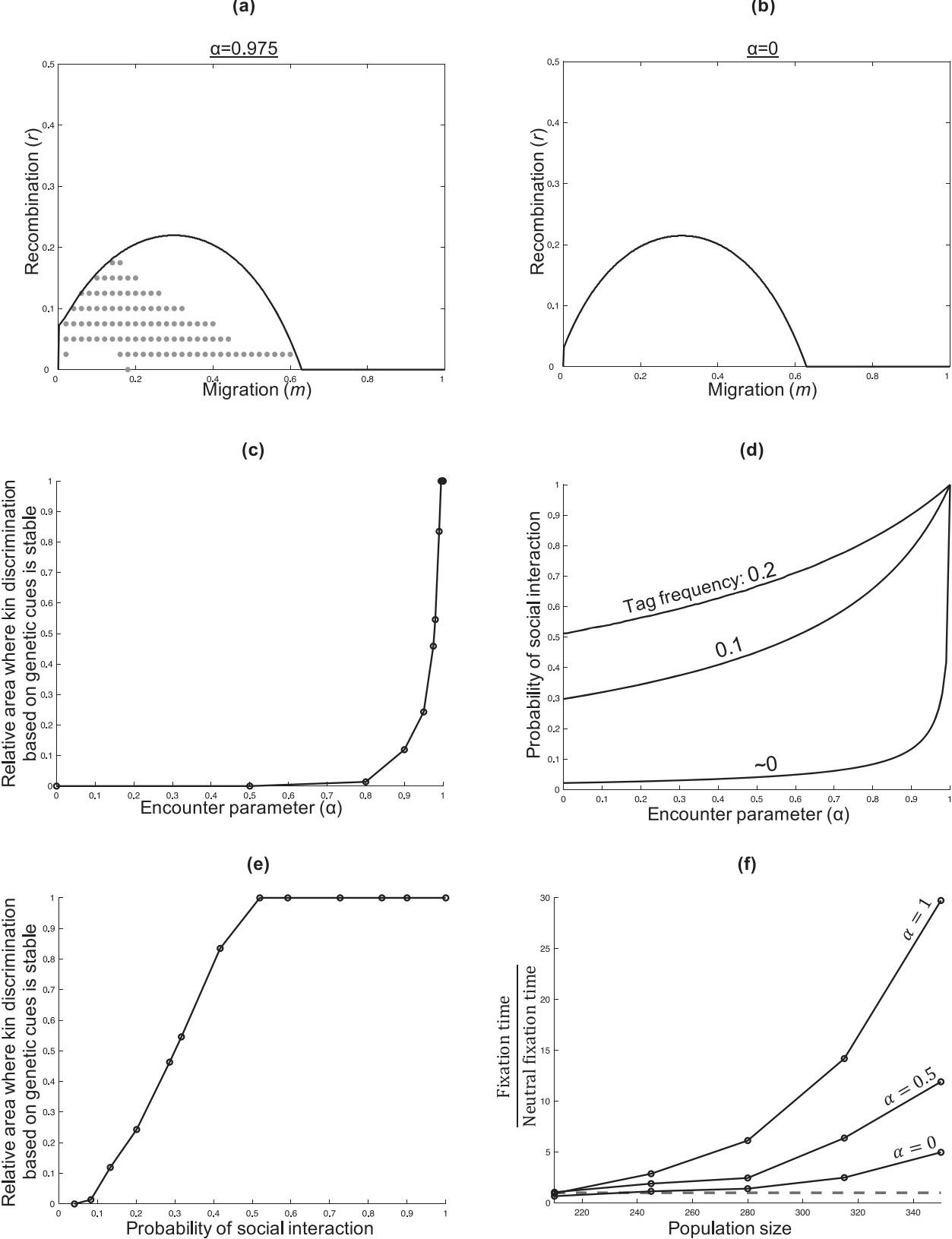

Fig. 5a). In this case, the total search cost for limitingly rare tags under low relatedness would be nine times higher than the cost of helping (0.9 versus 0.1). The search cost ($c_{search}$) has a smaller influence because as it increases, it does not reduce the likelihood of receiving help (compared to when $\alpha$ decreases), and so rare tags still gain an appreciable benefit (see 'Full analysis (solving the model)'). Furthermore, if the partner search rate ($\alpha$) is reduced, the partner search cost ($c_{search}$) has even less of a destabilising effect, simply because the search cost will be paid less often (Fig. 5b). However, if the search cost ($c_{search}$) is increased high enough, kin recognition will eventually be destabilised.

**Fig. 4 Social encounter rate. a** and **b** show when genetic kin recognition is stable. The area under the solid line shows where kin discrimination is favoured by kin selection (Eq. 1 satisfied). The data points show where, for a given value of $\alpha$, kin discrimination based on genetic cues is stable. **c**, **e** The Y axis represents the area of parameter space where kin discrimination based on genetic cues is stable for a given value of $\alpha$, divided by the area it is stable when $\alpha = 1$. **c** As the encounter parameter ($\alpha$) decreases, the area where genetic kin recognition is favoured decreases. **d** As the encounter parameter ($\alpha$) increases, the per-generation probability of encountering and interacting with a tag-matched individual increases. The different lines represent different population tag frequencies (~0, 0.1, 0.2). **e** As the probability of a social interaction increases, the area where genetic kin recognition is favoured also increases. **f** The Y axis represents the time taken for tag diversity to be lost, relative to the neutral scenario (no selective effects), in a finite-population model where social selection (b,c) is strong. This increases with population size, and the increase is steeper for higher $\alpha$, indicating that the strength of balancing selection at the recognition locus increases with the encounter parameter. We assumed: $c_{search} = 0$; **a–e** $\mu_{Trait} = 0.001$, $b = 0.3$, $c = 0.1$, $L_{max} = 100$, $N = 30$; kin discrimination based on genetic cues is stable when >10 tags maintained and helping frequency >0.4; **e** tag frequency ~0; **f** $\mu_{Trait} = 0.005$, $b = 4.5$, $c = 0.5$, $L_{max} = 2$, $N = 7$, $m = 0.01$, $r = 0.01$.

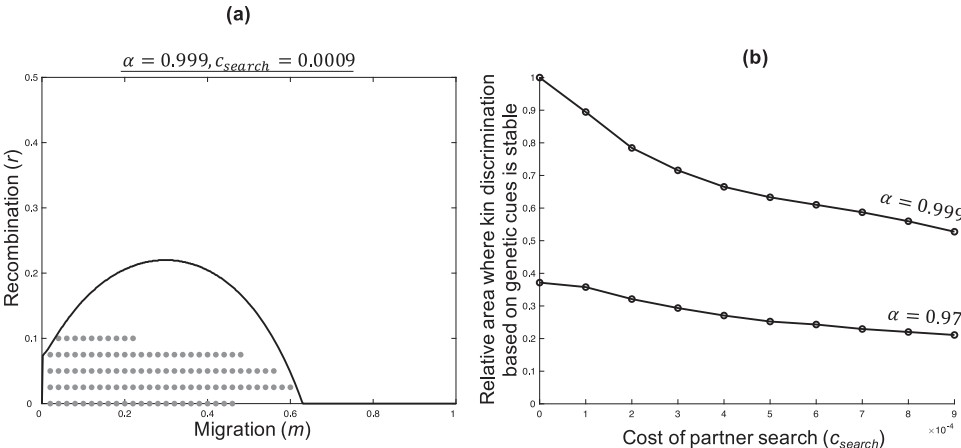

**Fig. 5 Search cost.** Panel **a** shows when genetic kin recognition is stable. The area under the solid line shows where kin discrimination is favoured by kin selection (Eq. 1 satisfied). The data points show where, for a partner search cost of $c_{search} = 0.0009$ and encounter parameter of $\alpha = 0.999$, kin discrimination based on genetic cues is stable. **b** The Y axis represents the area of parameter space where kin discrimination based on genetic cues is stable for given values of $c_{search}$ & $\alpha$, divided by the area it is stable when $c_{search} = 0$ & $\alpha = 1$. As the search cost ($c_{search}$) increases, the area where genetic kin recognition is favoured drops off relatively slowly, especially when $\alpha$ is lower. We assumed: $\mu_{Trait} = 0.001$, $b = 0.3$, $c = 0.1$, $L_{max} = 100$, $N = 30$; kin discrimination based on genetic cues is stable when >10 tags maintained and helping frequency >0.4.

**Alternative scenarios and genetic architecture**. A difference between ours and previous theory is that we show how high levels of both conditional helping and tag diversity can be maintained relatively easily, without tag mutation, strong selection on social behaviour, or additional selection pressures unrelated to social behaviour. Previous studies found that, in general, tag diversity could not be maintained by selection on social behaviour alone, recovering Crozier's paradox[4,7,23–26]. These previous studies did find restrictive conditions where kin discrimination based on genetic cues could evolve, but it was characterised by relatively low levels of both conditional helping and tag diversity (Supplementary Discussion 1). In addition, our theory modelled a relatively unfavourable scenario for genetic kin recognition, and so our finding that it can be stable may be conservative. In particular, we assumed: an island model with extreme local competition for resources[7,27,28,33,35]; no tag mutation; conditional helping is a discrete (all or nothing) trait, where cheating can be purged by selection; and that encounters are not restricted to individuals yet to have a social interaction (Supplementary Discussion 2).

A possible alternative solution to Crozier's paradox is if the recognition alleles have an additional role unrelated to social behaviour that maintains tag diversity. In such cases, negative frequency dependence could arise from selection acting on the recognition alleles' additional role, rather their role in social behaviour[5,7,8]. For instance, if recognition alleles are also MHC alleles, negative frequency dependence may arise because rare MHC alleles fare better in host-parasite interactions[4–8]. Alternatively, if recognition alleles are also mate-choice tags, negative

frequency dependence may arise because rare genetic mating cues are more reliable indicators of inbreeding[5]. These resolutions need not, in principle, be mutually exclusive, and could even act concordantly, with selection on social behaviour combining with extrinsic forces to increase the overall level of balancing selection at the recognition locus. However, there are possible complications, such as host-parasite coevolution leading to fluctuating allele frequencies, and it has yet to be shown that natural selection would 'choose' a locus under extrinsic balancing selection for a kin recognition tag, as opposed to a locus that was otherwise neutral. Formal theoretical modelling is required to examine the consequences of allowing the genetic architecture to evolve.

**Empirical implications**. To conclude, our findings have four implications for empirical research. First, the hunt for alternative factors to maintain tag diversity in species with genetic kin recognition, such as host-parasite interactions, may not be necessary[3–5,7,17,39]. We have shown how genetic kin recognition can maintain the tag diversity that it requires, without any other factor. Second, our theory emphasises the need to measure the frequency with which individuals have social encounters with other individuals (Fig. 4d). When this frequency is high, corresponding to a high $\alpha$, genetic kin recognition is more likely to be stable. Third, Eq. 1 can explain the variation that has been observed in the extent of kin discrimination across cooperatively breeding birds and mammals. Kin discrimination is greater in species where helping provides a greater benefit, and where relatedness within groups is more variable[10,11].

Finally, there is a need to develop and test adaptive hypotheses for variation across species in whether environmental or genetic cues are used for kin discrimination. It had been concluded that the use of environmental cues represents the 'best of a bad job', on the grounds that Crozier's paradox prevents the use of genetic cues[10,11]. Instead, our results suggest that we need to examine the relative costs and benefits of using different sorts of cues. Genetic cues could be more likely to be favoured when there is greater opportunity for multiple low-cost social encounters (higher $\alpha$ & lower $c_{search}$), for instance, when social groups are more compact (dense social networks). Environmental cues could be less likely to be favoured when they are less reliable, for instance, when: (i) there is high promiscuity, meaning 'sibling' becomes a less reliable cue of relatedness; (ii) offspring have less opportunity to learn cues of relatedness, such as when there are low or non-existent parental/offspring associations. Theory investigating different scenarios will allow us to explain the vast diversity of kin recognition systems observed across the natural world.

## Methods

We develop a theoretical model to track the evolution of genetic kin recognition, and examine the theoretical viability of Crozier's paradox. The 'Methods' section is organised as follows. In 'Background', we briefly go over the key conceptual issues that crop up throughout our analyses: Crozier's paradox, Grafen's linkage disequilibrium, partner search and kin selection. In 'Model assumptions', we list our model assumptions, and in 'Model construction and analysis', we construct and analyse the model. In 'Additional definitions and derivations', we give mathematical definitions and derivations for the summary statistics plotted in Figs. 2 and 3.

### Background

*Crozier's paradox.* Crozier argued that genetic kin recognition will often be evolutionarily unstable[8]. The argument is that more common tags are more likely to be recognised and helped than rare tags, generating positive frequency dependence at the tag locus, and resulting in a loss of tag diversity, destabilising kin recognition (Fig. 1).

*Grafen's linkage disequilibrium.* Grafen suggested that Crozier's paradox could be solved by coevolution between helping and kin recognition[31]. Specifically, as tags become more common, they will become less useful cues of relatedness, and so can be invaded by non-cooperative cheats. Rare tags will be good indicators of relatedness, preventing invasion by cheats (Fig. 3). Technically, a statistical association between genes for helping and rare tags will build up (linkage disequilibrium). This leads to individuals with rare tags having greater payoffs from social interactions. Grafen argued that this advantage for rare tags (reduced risk of being cheated) could exceed Crozier's advantage for common tags (increased chance of being recognised and helped), meaning rare tags gain an overall advantage, maintaining tag diversity.

However, a problem with this argument is that linkage disequilibrium is constantly broken down by direct selection at the tag locus (for increased social interaction rate) and recombination. As a result, linkage disequilibrium may not be strong enough to give rare tags an overall advantage. This was confirmed in a mathematical analysis by Rousset and Roze[7]. They found that, except in restrictive (biologically unnatural) scenarios, Grafen's linkage advantage for rare tags is less than Crozier's advantage for common tags, meaning tag diversity is lost.

*Partner search.* We hypothesise that more realistic forms of social interactions create forces that maintain genetic diversity at a tag (matching) locus, and so eliminate Crozier's paradox. Specifically, if individuals can have multiple encounters, testing out each individual, before settling on one tag-matched social partner, then individuals with rare tags can be recognised as much as individuals with common tags. Technically, this means that Crozier's advantage for common tags (increased opportunity to be recognised and helped) would be minimised or eliminated. As a result, Grafen's advantage for rare tags (less chance of being cheated), even if weak, could dominate, giving rare tags an overall advantage, maintaining tag diversity.

*Kin selection.* We hypothesise that kin discrimination based on genetic cues may only evolve in the region of parameter space where it is favoured by individual-level selection (kin selection)[26]. Previous theory has not always restricted itself to this area, and hence, in some cases, has been searching for kin discrimination in situations where either indiscriminate helping or indiscriminate defection is favoured. However, a failure to find genetic kin recognition in these situations is expected from kin selection theory. We will explicitly derive the regions of parameter space where kin discrimination is favoured by kin selection, and search for genetic kin recognition in these regions.

Our hypothesis is that genetic kin recognition will evolve if two conditions are met: (1) kin discrimination is favoured by individual-level kin selection (it confers greater inclusive fitness returns than both indiscriminate helping and indiscriminate defection); (2) individuals can engage in sufficiently many social *encounters* before committing to a given social *interaction* (essentially "trying out" multiple individuals to see if any are tag-matched, before having to commit to a social interaction with one of them).

### Model assumptions

*Tag & trait loci.* We assume an infinite population of individuals and that, at the start of each generation, individuals are haploid. Each individual encodes a given phenotype (tag). Tags are recognised by other individuals, and distinguishable. There are $L_{max}$ possible tags, and each tag is encoded by a specific allele at the 'tag locus' ($L_{max}$ possible alleles at the tag locus). $L_{max}$ therefore gives the upper bound on the number of tags that can be distinguished between ('tag availability'). This upper bound ($L_{max}$) is set by the efficacy of the sensory system responsible for recognising the tags (evolutionary constraint). More sophisticated sensory systems will be capable of reliably distinguishing between greater numbers of tags (higher $L_{max}$). Each allele at the tag locus is denoted by a number, $i$, within the set $i \in \{1, 2, …, L_{max}\}$. At a given point in time, the number of segregating tags (i.e. number of tags present at non-zero population frequency) is denoted by $L$ ($1 \le L \le L_{max}$). At equilibrium, the number of segregating tags is denoted by $L^*$ ($L$ tends to $L^*$ in the evolutionary long term).

Each individual also adopts a given 'trait', which dictates how it behaves in social interactions. There are two possible traits, and each trait is encoded by a specific allele at the 'trait locus' (2 possible alleles at the trait locus). Trait allele '1' encodes (conditional) helping and trait allele '0' encodes defection. In addition to conditional helping and defection, there is a third possible trait phenotype—indiscriminate helping (help everyone). However, for simplicity, we do not explicitly track an indiscriminate helping allele at the trait locus. Instead, we note that the 'indiscriminate helping' phenotype can still evolve in our model, if tag diversity is lost (one tag goes to fixation) and the conditional helping allele goes to fixation.

*Genotype frequency notation.* The population frequency of conditional helpers bearing a given tag $i$ is denoted by $x_{i1}$. The population frequency of defectors bearing a given tag $i$ is denoted by $x_{i0}$. The overall population frequency of a given tag $i$ is given by $x_{i1} + x_{i0}$, and denoted by $x_i$. The proportion of individuals bearing a given tag ($i$) that are helpers ('helper proportion') is given by $x_{i1}/x_i$, and denoted by $p_i$. The proportion of individuals bearing a given tag ($i$) that are cheaters ('cheater load') is therefore given by $1-p_i$.

*Social encounters and interactions.* For each individual, in each generation, we define an 'interaction group'. This is the group of neighbours with whom the individual is close enough to socially interact with[40]. Social interactions are pairwise and asymmetrical, comprising one actor (who may give help) and one recipient (who may receive help).

Each individual has one 'social search' per generation. In a given social search, a focal individual encounters a random member of its interaction group (partner). If the focal individual and its partner share the same tag, they interact, with the focal individual potentially giving help (actor) and its partner potentially receiving help (recipient)—the social encounter becomes a social interaction (successful social search).

In contrast, if the focal individual and its partner do not share the same tag, what happens depends upon the search parameter, $\alpha$. With a probability $\alpha$, the focal individual abandons its tag-mismatched partner and re-associates for a new social encounter, with a new partner drawn at random from its interaction group (with replacement of previously encountered individuals) (Fig. 2a). With a probability $1-\alpha$, the focal individual remains with its tag-mismatched partner, but they do not interact—the opportunity to socially interact is wasted (failed social search).

We reiterate that a given individual has one social search per generation, meaning it socially interacts as an actor either once (successful social search) or zero (failed social search) times per generation. However, a given individual may be chosen once, zero or multiple times per generation, by other individuals on their social searches. A given individual may therefore socially interact as a recipient once, zero or multiple times per generation.

When $\alpha = 1$, individuals are free to have encounters with all the other individuals in their interaction group, if need be, to find a tag-matched individual to interact with. In this case, even individuals with a rare tag are still likely to interact with another individual with the same tag. At the other extreme, when $\alpha = 0$, any individual who does not encounter a tag-matched individual on its first try does not get to engage in a social interaction, and so population tag frequency will determine the rate of interaction.

We assume that searching for partners is costly. Specifically, each time an individual abandons a social partner for a new social encounter, it pays a fecundity cost of $c_{search}$.

*Cooperative game.* For each social interaction, which comprises pairs of individuals sharing the same tag, there is one actor and one recipient. The actor and recipient

play a (nonreciprocal) cooperation game. Actors provide help if they have the conditional helping allele, suffering a fecundity cost of $c$ to give a benefit of $b$ to the recipient. Actors do not help if they have the defection allele. Recipients never help. There is a net benefit to helping ($b > c$).

*Mutation.* We include the possibility for mutation at the tag and trait loci. We assume that, each generation, trait mutation occurs with probability $\mu_{Trait}$, and tag mutation occurs with probability $\mu_{Tag}$. However, we assume, except where specified, that there is no mutation at the tag locus ($\mu_{Tag} = 0$). We make this assumption because tag mutation can maintain tag diversity even when it is disfavoured by selection, leading to spurious (non-adaptive) kin recognition. In general, we are interested in when selection maintains tag diversity.

*Assigning interaction groups.* In this (island) model, the population is split into distinct physical groups of individuals (demes) of size $N$, and social interactions take place amongst individuals on the same deme. Therefore, for each individual, in each generation, the interaction group comprises the $N-1$ other individuals (i.e. discounting themselves) on the deme.

We note that the mathematical model we construct in 'Model construction and analysis' is only completely accurate for the cases where $\alpha = 0$ and/or $N = \infty$, and accuracy in the $\alpha > 0$ case is reduced as deme size ($N$) is reduced. However, in the 'Finite population (agent-based) simulation' section, we verify using agent-based simulation that the results of the mathematical model tend to hold even for small deme sizes ($N$). The reason why our mathematical model is only accurate for the $\alpha = 0$ and/or $N = \infty$ case is that, for analytical tractability, we do not account for stochasticity in the genetic composition of demes. However, some stochasticity will arise whenever demes are finite, and stochasticity will matter (affect evolution) whenever individuals can have multiple social encounters before each social interaction ($\alpha > 0$). More discussion of this can be found under the "Relation to Rousset and Roze[7]" heading below, and in the 'Finite population (agent-based) simulation' section.

*Lifecycle.* We assume an infinite population of haploids, partitioned into demes[9,28,41,42]. At the beginning of the lifecycle, each deme has $N$ haploid individuals. Firstly, haploid individuals have the opportunity to socially interact, as detailed under the "Social encounters and interactions" heading above. Next, haploid individuals produce a very large number of gametes, before dying, where an individual's fecundity is given by how well it fared in social interactions and its investment in partner search. Each gamete has a $1-m$ probability of staying in its native deme, and a $m$ probability of emigrating to a different, randomly chosen deme. Then, gametes fuse randomly within demes to produce diploid zygotes, and this is followed immediately by meiosis, with recombination between tag and trait loci occurring with probability $r$. The haploid juveniles undergo mutation, as detailed under the "Mutation" heading above. Finally, $N$ haploid adults are sampled randomly from the haploid juveniles in each deme (population regulation occurs at the deme, or 'local', level). This completes the lifecycle.

*Relation to Rousset and Roze[7].* Our lifecycle assumptions are almost identical to those taken by Rousset and Roze (henceforth: R&R). However, there are three differences.

R&R assumed that a social encounter with a tag-mismatched partner never results in a social interaction (no partner search). We, on the other hand, allow for multiple social encounters before each social interaction, meaning an initial encounter with a tag-mismatched partner could still ultimately result in a social interaction, if the tag-mismatched partner is abandoned and the focal individual successfully re-associates with a tag-matched partner. In other words, in our model, but not in R&R, partner search is permitted. We hypothesise that this model generalisation will facilitate the evolution of genetic kin recognition. We note that R&R's case (no partner search) is recovered in our broader framework for the special case where partner search is absent ($\alpha = 0$).

Secondly, R&R placed no restrictions on deme size ($N$). However, in our model, when there is partner search ($\alpha > 0$), our model is only accurate for the case where deme size ($N$) is infinite, with reduced accuracy as deme size ($N$) decreases. This inaccuracy arises because, for analytical tractability, we account for repeat social encounters ($\alpha > 0$) in a deterministic modelling framework, without accounting for stochastic effects. To appreciate this, note that, if demes are small (small $N$), each deme will only comprise a subset of the genetic variation (genotypes) present in the wider population. This means that, if an individual abandons a tag-mismatched partner in search of a tag-matched partner to socially interact with, whether or not it ends up with a tag-matched social partner will depend, not only on the genotypes present in the population, but also on stochastic effects—namely, whether the individual happens to have entered into a deme that comprises someone else with the same tag. Conversely, if demes are large (high $N$), every genotype present in the population is also likely to be present in every deme in the population. This means that the outcome of partner search can be largely determined with sole reference to population-wide characteristics (demographic parameters and population genotype frequencies). We focus on such population-wide characteristics, and ignore the stochastic variation in deme composition associated with low deme size ($N$). This means that our model is only technically accurate for the case where there is no partner search ($\alpha = 0$), or where there is no stochastic variation in the genetic

composition of demes ($N = \infty$). Having said that, in 'Finite population (agent-based) simulation', we verify using an agent-based simulation that our theoretical results still tend to hold even when stochasticity in deme composition is incorporated and deme size ($N$) is low.

Thirdly, R&R assumed that each individual has $N-1$ social encounters each generation, one with each other member of its deme, and that the fecundity benefit and cost of helping are respectively given by $b/(N-1)$ and $c/(N-1)$. We, on the other hand, assume that each individual, in its social search, initiates one (replaceable, if the social partner is abandoned to obtain a new one) social encounter per generation, with an individual drawn randomly (discounting itself) from its deme, and that the fecundity benefit and cost of helping are respectively given by $b$ and $c$. We note that the alternative conceptualisations are mathematically equivalent under our assumptions of small $b$ & $c$ (weak selection) and no stochastic deme variation[9]. In both cases: the total fecundity of all individuals sharing a common genotype is the same; the maximal generational fecundity benefit and cost to an individual is the same (and given by $b$ and $c$). However, the reason why we have adopted a slightly different conceptualisation to R&R is that our conceptualisation extends more naturally to the scenario where individuals can search for social partners.

## Model construction and analysis

**Model construction and analysis**. We now mathematically formulate and analyse our model. We do so gradually, taking the following steps:

(a) Constructing the model. We write equations to describe how genotype frequencies change every generation due to: (i) selection; (ii) recombination; (iii) mutation. We combine these equations (i, ii, iii) to obtain recursions describing how genotype frequencies change across a generation. For simplicity, we assume that selection and mutation are weak (of low magnitude), population size is infinite, and there is no stochastic variation in the genetic composition of demes (a reasonable assumption if deme size, $N$, is large).

(b) Individual-level analysis (finding the right area of parameter space). We derive coefficients of relatedness ($R_{tag}$, $R_{competitor}$), which allows us to formulate a condition, based on Hamilton's rule, to show when kin discrimination (help relatives) confers greater fitness returns than indiscriminate defection (never help) and indiscriminate helping (help everyone), meaning it is favoured by an actor striving to maximise its inclusive fitness[1]. This is the area of parameter space where we should look for genetic kin recognition.

(c) Full analysis (solving the model). We numerically solve our model to see what genotype frequencies arise at equilibrium. This allows us to examine whether natural selection can maintain genetic variability at the tag locus, alongside conditional helping, and hence allow kin discrimination based on genetic cues to evolve and be stable.

(d) Finite population (agent-based) simulation. We check if our theoretical results still hold when selection and mutation are stronger, the population is finite, and the genetic composition of demes vary realistically (stochastically).

(e) Key points and implications.

**Constructing the model**. Based on our lifecycle assumptions (see 'Model assumptions'), we write a recursion to describe how the population frequency of a given genotype changes across a generation. Specifically, we write three equations, partitioning the respective effects of selection, recombination and mutation, on genotype frequency change. We assume that: (1) Selection takes the frequency of a genotype ($ij$) from $x_{ij}$ to $x_{ij}'$, where $i$ gives tag identity, and $j$ gives trait identity, with $j = 1$ for a conditional helper and $j = 0$ for a defector. (2) Recombination takes genotype frequency from $x_{ij}'$ to $x_{ij}''$. (3) Mutation takes genotype frequency from $x_{ij}''$ to $x_{ij}'''$. We partition our model in this way—as three successive equations—because the logic behind our model is clearer when the effects of selection, recombination and mutation are presented in isolation from each other. Taken together, the equations give a 'recursion', describing the change in frequency of a genotype ($ij$) from one generation ($x_{ij}$) to the next ($x_{ij}'''$).

We note that the selection equation comprises the consequences of both reproduction, which occurs at the start of the lifecycle (before recombination and mutation), and competitive displacement (population regulation), which occurs at the end of the lifecycle (after recombination and mutation). In an iterative evolutionary process, in which one generation follows seamlessly from the next, the "start" of one generation follows the "end" of the previous generation. For this reason, we are justified in our decision to "move" competitive displacement from the end of the generation to the start, to consider it alongside reproduction in a single "selection" equation.

We construct the *selection* equation first, followed by the *recombination* equation, and finally the *mutation* equation. The selection equation is most cumbersome to construct, and must be done in multiple stages, in which higher-level variables are written in terms of increasingly low-level, mechanistic variables, until the equation is specified solely in terms of fundamental model parameters (dynamic sufficiency[43,44]).

*Selection (high-level expressions).* We construct our selection equation in stages from high-level to low-level details. Before doing so, we go over some features of our lifecycle again, this time with a slightly different emphasis, so that the relation between the verbal lifecycle description, and the algebraic model we are about to construct, is clear.

First, we reiterate that social interactions are pairwise and asymmetrical, with one actor potentially giving help, and one recipient potentially receiving help (see 'Model assumptions'). Every generation, each individual has one social search, culminating in one opportunity to socially interact as the actor. For each opportunity, an individual may engage in a series of social encounters (partner search), before finally settling on an "ultimate partner". This ultimate partner is either tag-matched, resulting in social interaction (where the focal individual is the actor, and its ultimate partner is the recipient), or tag mismatched, resulting in no social interaction (missed opportunity).

We now explicitly define a few probabilities. $M_{interact}$ is the per-generation probability that a given individual engages in a social interaction as the actor (i.e. obtains a tag-matched ultimate partner). $M_{ind\_helped}$ is the expected number of times per-generation that a given individual receives help (i.e. is chosen as a tag-matched helper's ultimate partner). $M_{deme\_helped}$ is the expected number of times per-generation that a random individual drawn from the focal individual's deme (i.e. not necessarily, but possibly, the focal individual) receives help (i.e. is chosen as a tag-matched helper's ultimate partner). $M_{pop\_helped}$ is the expected number of times that a random individual drawn from a different deme to the focal individual receives help (i.e. is chosen as a tag-matched helper's ultimate partner). $M_{ind\_abandon}$ is the expected number of times per social search (i.e. per-generation) that a given individual abandons its social partner for a new social encounter. $M_{deme\_abandon}$ is the expected number of times per social search (i.e. per-generation) that a random individual drawn from the focal individual's deme abandons its social partner for a new social encounter. $M_{pop\_abandon}$ is the expected number of times per social search (i.e. per-generation) that a random individual drawn from a different deme to the focal individual abandons its social partner for a new social encounter.

$M_{pop\_helped}$ and $M_{pop\_abandon}$ are the same for each individual in the population, regardless of the individual's tag or trait identity—that is, having a particular tag or trait does not affect the likelihood that a random individual drawn from a non-native deme will receive help or abandon social partners. This is because, in the infinite island model, there is no genetic correlation (relatedness) between individuals drawn from different patches (coalescence takes an infinite amount of time).

However, $M_{deme\_abandon}$, $M_{ind\_abandon}$, $M_{deme\_helped}$, $M_{ind\_helped}$ and $M_{interact}$ vary depending on the population frequency of an individual's tag. Furthermore, owing to genetic correlations between individuals (relatedness), $M_{ind\_helped}$ and $M_{deme\_helped}$ vary depending on an individual's trait (conditional helping/defection). We therefore index these "M-terms" with '*i*' to denote a given tag, and with '*j*' to denote a given trait, with *j* = 1 & *j* = 0 corresponding respectively to conditional helping & defection.

Having defined these "M-terms", we can now say that, each generation, a conditional helper with a given tag *i* will socially interact (in the actor role) with probability $M_{interact_i}$, resulting in an expected generational fecundity loss of $cM_{interact_i}$. Each generation, an individual with a given tag *i* and trait *j* will receive help an expected $M_{ind\_helped_{ij}}$ number of times, resulting in a generational fecundity benefit of $bM_{ind\_helped_{ij}}$. Each generation, an individual with a given tag *i* will accrue an expected generational partner search cost of $c_{search}M_{ind\_abandon_i}$.

Each generation, the total number of times help is given on an individual's deme is given by $NM_{deme\_helped_{ij}}$, where *N* is the number of individuals on the focal individual's deme (including itself), and *i* and *j* are the focal individual's tag and trait identity. This means that helping causes a net generational fecundity increase, summed across all individuals on the focal individual's deme, of $(b-c)NM_{deme\_helped_{ij}}$. This results in increased gamete production on the deme. A fraction of these gametes $(1-m)$ stay on the native deme, meaning the deme's net fecundity increase, after gamete dispersal, arising from helping on the native patch, is given by $(1-m)(b-c)NM_{deme\_helped_{ij}}$.

However, regardless of how big the deme's juvenile haploid population is, only *N* of these juveniles are (randomly) sampled from it to establish the deme's next adult population (local density dependence). Therefore, for any increase in fecundity due to helping, there will be an equal loss in fecundity due to competition. The deme's net fecundity decrease, arising from competition on the native patch, is therefore given by $(1-m)(b-c)NM_{deme\_helped_{ij}}$. Each of the *N* parents on the native deme bears this fecundity cost equally (stochastic deviations from equality are negligible because each adult produces lots of juvenile offspring). This cost arises from helping that has taken place on the native deme (helping-induced local competition). An individual with the genotype *ij* suffers the following generational fecundity cost of helping-induced local competition: $(1-m)(b-c)M_{deme\_helped_{ij}}$.

For analogous reasons, each parent on a native deme will also suffer a helping-induced fecundity cost arising from competitive displacement of their offspring by juveniles produced by gametes that have migrated in from non-native patches. This fecundity cost is borne equally by each individual in the population (stochastic deviations from equality are negligible because each adult produces lots of juvenile offspring). This cost arises from helping that has taken place on non-native demes

(helping-induced global competition). Each individual in the population suffers the following generational fecundity cost of helping-induced global competition: $m(b-c)M_{pop\_helped}$.

Partner abandoning also has consequences for competition. Specifically, partners will be abandoned a total of $NM_{deme\_abandon_i}$ times each generation in a focal (tag *i*) individual's deme, resulting in a net reduction of juveniles on the native deme, increasing the relative competitive success of the focal individual's juvenile offspring. Specifically, partner abandoning on the native deme will increase the focal individual's fitness by $(1-m)M_{deme\_abandon_i}c_{search}$ (partner abandoning-induced local competition). Furthermore, partner abandoning on non-native demes will increase the focal individual's fitness by $mM_{pop\_abandon}c_{search}$ (partner abandoning-induced global competition).

An individual's absolute fitness, defined as its number of offspring that survive through one iteration of the lifecycle, is denoted by $w_{ij}$, where the subscript *ij* denotes the individual's tag (*i*) and trait (*j*) identity[28]. Given our lifecycle assumption that population size is constant over generations, we note that: the population mean absolute fitness will be equal to 1; an individual's *relative* fitness (obtained by dividing its absolute fitness by the population mean absolute fitness) will be equal to its *absolute* fitness (because in a population of constant size, absolute fitness is converted to relative fitness by dividing by 1). We also note that, though our fitness definitions are suitable for current purposes, how best to define fitness in general is an ongoing research question[45,46]. We emphasise that the "M-terms" do not vary independently from each other; for instance, they relate to each other in such a way that the population average absolute fitness is given by 1 $\left(\sum_{l=1}^{L_{max}}(x_{l0}w_{l0} + x_{l1}w_{l1}) = 1\right)$.

A given genotype *ij* will, owing to selection, change in frequency according to $x_{ij}' = x_{ij}w_{ij}$. We can use the information given in this section to write absolute fitness ($w_{ij}$) explicitly in terms of our "M-terms". We give absolute fitness functions for both conditional helpers ($w_{i1}$) and defectors ($w_{i0}$):

$$w_{i1} = 1 - M_{interact_i}c + M_{ind\_helped_{i1}}b - (b-c)\left((1-m)M_{deme\_helped_{i1}} + mM_{pop\_helped}\right)$$
$$- c_{search}\left(M_{ind\_abandon_i} - (1-m)M_{deme\_abandon_i} - mM_{pop\_abandon}\right), \tag{2}$$

$$w_{i0} = 1 - M_{ind\_helped_{i0}}b - (b-c)\left((1-m)M_{deme\_helped_{i0}} + mM_{pop\_helped}\right)$$
$$- c_{search}\left(M_{ind\_abandon_i} - (1-m)M_{deme\_abandon_i} - mM_{pop\_abandon}\right). \tag{3}$$

*Probabilities of coalescence.* We now begin to close our expressions (Eqs. 2 and 3) by writing our M-terms as functions of various probabilities of identity by descent (IBD). Two genes sampled at a given locus in two different individuals are said to be identical by descent (IBD) if they converge, in finite time, to a single point of common ancestry (coalesce).

To explain this concept a bit more precisely, if we take two individuals from a common deme, and focus on a given locus, we can ask if the two genes at this locus are identical by descent. To work out if they are, we need to consider the ancestral lineages of each of the two genes. If, going backwards in time, through the parents, grandparents, great-grandparents, and so on, the two ancestral lineages eventually converge on the same individual (e.g. they have a common great, great, …, grandparent), then the lineages "coalesce" (converge) and the genes can be said to be "identical by descent". This only happens if the ancestral lineages of each gene "stay" in the same (common) deme long enough for the lineages to converge. If one ancestral lineage "moves" to a different deme, because, for instance, the great-great grandfather was a migrant, then coalescence takes an infinite amount of time, and the genes are said to be *not* identical by descent[28,44,47].

After R&R, we define the following probabilities of coalescence (identity by descent). See Supplementary Fig. 1 for a visual depiction of these coalescence probabilities. *F* is the probability that genes sampled at a given locus, in two individuals drawn at random (without replacement) from a common deme, are IBD (coalesce in finite time/coalesce in the same deme). *Φ is* the probability that, when genes are sampled at two loci (e.g. locus A and B), in two individuals drawn at random (without replacement) from a common deme, the pair of genes at locus A are IBD, and the pair of genes at locus B are also IBD. *G* is the probability that, if genes are sampled at a given locus in three individuals drawn from a common deme (individual 2 drawn without replacement of individual 1; individual 3 drawn with replacement of individuals 1 & 2), the triplet of genes are IBD. *γ* is the probability that, if three individuals are drawn from a common deme (individual 2 drawn without replacement of individual 1; individual 3 drawn with replacement of individuals 1 & 2), individuals 1 & 2 coalesce at one locus (e.g. locus A), and individuals 2 & 3 coalesce at a different locus (e.g. locus B).

With these coalescence probabilities defined, we can now write our M-terms in terms of these lower-level probabilities of coalescence. Before doing so, we need to make a simplifying assumption that selection and mutation are weak. That is—we need to assume that the magnitude of *b*, *c* & $c_{search}$ (coefficients of selection), as well as $\mu_{Trait}$ and $\mu_{Tag}$ (mutation rates), are small. This means that M-terms and IBD probabilities are functions of demography alone (*N*,*m*,*r*), not selection and mutation, which simplifies things[9,28].

$M_{interact_i}$. The per-generation probability of obtaining (as actor) a tag-matched ultimate partner, resulting in social interaction, can be written as follows. $x_i$ gives

the population frequency of the focal individual's tag, $i$.

$$M_{interact_i} = \frac{F + (1-F)x_i}{1 - \alpha(1-x_i)(1-F)}. \tag{4}$$

To interpret Eq. 4, note that there are a number of different ways for an individual to obtain a tag-matched ultimate partner. These are (i) obtain an ultimate partner that is tag-matched due to common ancestry; (ii) obtain an ultimate partner that is tag-matched despite lacking common ancestry. Given that we are ignoring stochastic variation in the genetic composition of demes, the probabilities of these outcomes occurring are proportional to (scale with): (i) $F$, (ii) $(1-F)x_i$.

However, these probabilities (i) and (ii) must be up-scaled to account for the fact that, before settling on an ultimate partner, an individual may have multiple social encounters with tag-mismatched partners, which are abandoned in favour of new social encounters. Because newly encountered partners are chosen with the replacement of individuals that were previously encountered during the social search, the probability of abandoning a given partner is the same, no matter how many encounters the focal individual has already had that generation. Given that we are ignoring stochastic variation in the genetic composition of demes, the per-encounter probability of abandoning a partner is given by $\alpha(1-x_i)(1-F)$. Social searches where partners are abandoned may ultimately progress into one of the outcomes (i) or (ii). Therefore, the total probability of finding a tag-matched partner to socially interact with, as actor ($M_{interact_i}$), is given by summing the probabilities of outcomes i–ii and dividing this through by $1 - \alpha(1-x_i)(1-F)$.

$M_{ind\_helped_{ij}}$. The expected number of times, per generation, that a focal individual receives help, can be written as follows. This expectation differs depending on whether the focal individual is itself a helper (Eq. 5) or defector (Eq. 6). $p_i$ gives the proportion of individuals bearing the tag $i$ who are helpers ('helper proportion' for short).

$$M_{ind\_helped_{i1}} = \frac{\phi + (F-\phi)p_i + (F-\phi)x_i + (1-2F+\phi)p_i x_i}{1 - \alpha(1-x_i)(1-F)}, \tag{5}$$

$$M_{ind\_helped_{i0}} = \frac{(F-\phi)p_i + (1-2F+\phi)p_i x_i}{1 - \alpha(1-x_i)(1-F)}. \tag{6}$$

To interpret Eqs. 5 and 6, note that there are a number of different ways for a focal individual to receive help (be chosen by a tag-matched helper). These are (i) be chosen by an actor that is IBD at the tag but not trait locus, and who happens to be a helper; (ii) be chosen by an actor that is IBD at neither the trait nor tag locus, and who happens to be a tag-matched helper.

If the focal individual is itself a conditional helper, rather than a defector, there are two more ways to receive help: (iii) be chosen by an actor that is IBD at both tag and trait loci; (iv) be chosen by an actor that is IBD at the trait but not tag locus, and who happens to be tag-matched.

Given that we are ignoring stochastic variation in the genetic composition of demes, the expected number of times that each of these outcomes occur are proportional to (scale with): (i) $(F-\phi)p_i$, (ii) $(1-2F+\phi)p_i x_i$, (iii) $\phi$, (iv) $(F-\phi)x_i$. To obtain the exact values, we must up-scale these values (i–iv) to account for extra social encounters obtained through the social search.

If the focal individual is a helper, the total expected number of times help is received ($M_{ind\_helped_{i1}}$) is given by summing the probabilities of outcomes i–iv, and dividing this through by $1 - \alpha(1-x_i)(1-F)$. If the focal individual is a defector, the total expected number of times help is received ($M_{ind\_helped_{i0}}$) is given by summing the probabilities of outcomes i–ii, and dividing this through by $1 - \alpha(1-x_i)(1-F)$.

$M_{deme\_helped_{i1}}$. The expected number of times, per generation, that a random individual drawn from a focal individual's deme, who is henceforth referred to as the 'local competitor', receives help, can be written as follows. This probability differs depending on whether the focal individual is a helper (Eq. 7) or defector (Eq. 8).

$$\begin{aligned} M_{deme\_helped_{i1}} = {}& \frac{M_{interact_i}}{N} + \frac{M_{ind\_helped_{i1}}}{N} \\ & + \frac{N-2}{N}\left( \gamma + (F-\gamma)\sum_{l=1}^{L_{max}} (x_l p_l) \right. \\ & + (F-\gamma)\sum_{l=1}^{L_{max}} \left( \frac{x_l p_l}{\sum_{l=1}^{L_{max}}(x_l p_l)}\left(x_l + (1-x_l)\alpha M_{interact_l}\right) \right) \\ & \left. + (1-2F+\gamma)\sum_{l=1}^{L_{max}}\left( p_l x_l \left(x_l + (1-x_l)\alpha M_{interact_l}\right) \right) \right), \end{aligned} \tag{7}$$

$$\begin{aligned} M_{deme\_helped_{i0}} = {}& \frac{M_{ind\_helped_{i0}}}{N} \\ & + \frac{N-2}{N}\left( (F-\gamma)\sum_{l=1}^{L_{max}} (x_l p_l) \right. \\ & \left. + (1-2F+\gamma)\sum_{l=1}^{L_{max}}\left( p_l x_l \left(x_l + (1-x_l)\alpha M_{interact_l}\right) \right) \right). \end{aligned} \tag{8}$$

To interpret Eqs. 7 and 8, note that there are a number of different ways for the local competitor to receive help (be chosen by a tag-matched helper). These are (a) the local competitor is the focal individual (they are the same individual), and the focal individual (and therefore the local competitor, by nature of being the same individual) receives help; (b) the local competitor is not the focal individual or the focal individual's ultimate partner, and: (i) the local competitor is chosen by an actor that is IBD (to the local competitor) at the tag locus, but not IBD (to the focal individual) at the trait locus, and who happens to be a helper; (ii) the local competitor is chosen by an actor that is not IBD (to the local competitor) at the tag locus, nor IBD (to the focal individual) at the trait locus, and who happens to be tag-matched with the local competitor and a helper.

Furthermore, if the focal individual is itself a conditional helper, rather than a defector, there are several additional ways for the local competitor to receive help. These are (c) the local competitor is chosen by the focal individual as their ultimate partner, and the two individuals are tag-matched; (d) the local competitor is not the focal individual or the focal individual's ultimate partner, and: (i) the local competitor is chosen by an actor that is IBD (to the focal individual) at the trait locus, and IBD (to the local competitor) at the tag locus; (ii) the local competitor is chosen by an actor that is IBD (to the focal individual) at the trait locus, but not IBD (with the local competitor) at the tag locus, and who happens to be tag-matched with the local competitor.

Given that we are ignoring stochastic variation in the genetic composition of demes, the expected number of times that each of these outcomes occur, after scaling up where necessary to account for extra social encounters obtained through the social search, are given by (a) $\frac{M_{ind\_helped_i}}{N}$, (bi) $\left(\frac{N-2}{N}\right)(F-\gamma)\sum_{l=1}^{L_{max}}(x_l p_l)$, (bii) $\left(1-2F+\gamma\right)\sum_{l=1}^{L_{max}}\left( p_l x_l \left(x_l + (1-x_l)\alpha M_{interact_l}\right) \right)$, (c) $\frac{M_{interact_i}}{N}$, (di) $\left(\frac{N-2}{N}\right)\gamma$, (dii) $\left(\frac{N-2}{N}\right)(F-\gamma)\sum_{l=1}^{L_{max}}\left( \frac{x_l p_l}{\sum_{l=1}^{L_{max}}(x_l p_l)}\left(x_l + (1-x_l)\alpha M_{interact_l}\right) \right)$.

If the focal individual is a helper, the total expected number of times that the local competitor receives help ($M_{deme\_helped_{i1}}$) is given by summing the probabilities of outcomes a, bi, bii, c, di, dii. If the focal individual is a defector, the total expected number of times that the local competitor receives help ($M_{deme\_helped_{i0}}$) is given by summing the probabilities of outcomes a, bi, bii.

$M_{pop\_helped}$. The expected number of times, per generation, that a random individual drawn from a different (non-native) deme to the focal individual, who is henceforth referred to as the 'non-native competitor', receives help, can be written as follows. This probability is the same, regardless of the focal individual's tag or trait identity.

$$M_{pop\_helped} = \sum_{l=1}^{L_{max}} \left( p_l x_l M_{interact_l} \right). \tag{9}$$

To interpret Eq. 9, note that, given that the non-native competitor is drawn from a different deme to the focal individual, the two individuals are not identical by descent. The RHS of Eq. 9 gives, when ignoring stochastic variation in the genetic composition of demes, the expected number of times that the non-native is chosen (to be an ultimate recipient) by an actor who is a tag-matched helper.

$M_{ind\_abandon_i}$. The expected number of times, per generation, that a focal individual (with tag $i$) abandons a social partner for a new encounter, can be written as follows.

$$M_{ind\_abandon_i} = \frac{\alpha(1-F)(1-x_i)}{1 - \alpha(1-F)(1-x_i)}. \tag{10}$$

We derived this term as follows. Given that we are ignoring stochastic variation in the genetic composition of demes, the probability of abandoning *exactly* one partner in a given social search (generation) is given by $\alpha(1-x_i)(1-F)(1-\alpha(1-x_i)(1-F))$, which is the per-encounter probability of abandoning a partner ($\alpha(1-x_i)(1-F)$), multiplied by the per-encounter probability of not abandoning a partner, bringing the social search to a close $(1-\alpha(1-x_i)(1-F))$. Generalising this argument, the probability of abandoning *exactly* $\eta$ partners in a given social search (generation) is given by $\alpha(1-x_i)(1-F)(1-\alpha(1-x_i)(1-F))^{\eta}$, which is the per-encounter probability of abandoning a partner, raised to the power $\eta$, and multiplied by the probability of not abandoning a partner, bringing the social search to a close. The *expected* number of times that a partner is abandoned is then given by summing the integers 1, 2, 3, …, ∞, where each integer is weighted by the probability of abandoning exactly that many partners. Formally, the expected number of times that a partner is abandoned is given by the infinite sum: $\alpha(1-x_i)(1-F)(1-\alpha(1-x_i)(1-F))\sum_{\eta=1}^{\infty}(\eta(\alpha(1-x_i)(1-F))^{\eta-1})$, which is a converging series

that can be written as $\alpha(1-x_i)(1-F)\left(1-\alpha(1-x_i)(1-F)\right)\left(\frac{1}{(1-\alpha(1-x_i)(1-F))^2}\right)$, which simplifies to Eq. 10.

$M_{deme\_abandon_i}$. The expected number of times, per generation, that a random individual drawn from a focal (tag $i$) individual's deme (the 'local competitor') abandons a social partner for a new encounter, can be written as follows. This probability varies with the focal individuals' tag identity, but not with the focal individual's trait identity.

$$M_{deme\_abandon_i} = \frac{1}{N}M_{ind\_abandon_i} + \frac{1}{N}(1-F)(1-x_i)\alpha\left(1+\frac{\sum_{l\neq i}^{L_{max}}(x_l M_{ind\_abandon_l})}{1-x_i}\right)$$
$$+ \frac{N-2}{N}\left((F-G)(1-x_i)\alpha\left(1+M_{ind\_abandon_i}\right)\right.$$
$$+ (F-G)(1-x_i)\alpha\left(1+\frac{\sum_{l\neq i}^{L_{max}}(x_l M_{ind\_abandon_l})}{1-x_i}\right)$$
$$\left.+ (1-3F+2G)\sum_{l=1}^{L_{max}}\left(x_l(1-x_l)\alpha\left(1+M_{ind\_abandon_l}\right)\right)\right). \tag{11}$$

To interpret Eq. 11, note that, with probability $1/N$, the local competitor and the focal individual are the same individual. If this is the case, it means that the local competitor abandons $M_{ind\_abandon_i}$ partners that generation.

With probability $1/N$, the focal individual is the local competitor's initial partner (i.e. the first individual that the local competitor encounters on its social search is the focal individual). If this is the case, it means that the local competitor goes on to abandon $(1-F)(1-x_i)\alpha\left(1+\frac{\sum_{l\neq i}^{L_{max}}(x_l M_{ind\_abandon_l})}{1-x_i}\right)$ partners that generation. To interpret this term, note that we know that the local competitor's initial social partner has the tag $i$, and so, if the local competitor is to abandon its initial partner, it must have a different tag (not tag $i$), and this occurs with probability $(1-F)(1-x_i)$. The local competitor then abandons its initial partner with probability $\alpha$, and goes on to abandon, in expectation, another $\frac{\sum_{l\neq i}^{L_{max}}(x_l M_{ind\_abandon_l})}{1-x_i}$ after this. This leads to a total of $(1-F)(1-x_i)\alpha\left(1+\frac{\sum_{l\neq i}^{L_{max}}(x_l M_{ind\_abandon_l})}{1-x_i}\right)$ partners abandoned that generation.

With probability $(N-2)/N$, the focal individual, local competitor and local competitor's initial partner are three different individuals. In this case, with probability $F$ the local competitor and the local competitor's initial partner are tag-matched due to identity by descent. If this is the case, it means that the local competitor does not abandon any partners that generation. Alternatively, with probability $F-G$, the focal individual and local competitor are IDB at the tag locus, with the local competitor and local competitor's initial partner *not* IDB at the tag locus. If this is the case, it means that the local competitor has the tag $i$, meaning it abandons its social partner $(1-x_i)\alpha\left(1+M_{ind\_abandon_i}\right)$ times that generation. Alternatively, with probability $F-G$, the focal individual and local competitor's initial partner are IDB at the tag locus, with the local competitor and local competitor's initial partner *not* IDB at the tag locus. If this is the case, it means that the local competitor's initial partner has the tag $i$, meaning the local competitor must have a different tag (not tag $i$) if it is to abandon any partners. This means the local competitor abandons $(1-x_i)\alpha\left(1+\frac{\sum_{l\neq i}^{L_{max}}(x_l M_{ind\_abandon_l})}{1-x_i}\right)$ social partners that generation. Alternatively, with probability $1-3F+2G$, the focal individual, local competitor, and local competitor's initial partner, are *not* IDB at the tag locus. If this is the case, it means that we have no information about the tag identity of the local competitor or its initial social partner, meaning the local competitor abandons $\sum_{l=1}^{L_{max}}\left(x_l(1-x_l)\alpha\left(1+M_{ind\_abandon_l}\right)\right)$ social partners that generation.

Summing across each of these possibilities gives the total number of partners abandoned by the 'local competitor' in a generation.

$M_{pop\_abandon}$. The expected number of times, per generation, that a random individual drawn from a different (non-native) deme to the focal individual (the 'non-native competitor') abandons a social partner for a new encounter, can be written as follows. This probability is the same, regardless of the focal individual's tag or trait identity.

$$M_{pop\_abandon} = \sum_{l=1}^{L_{max}}(x_l M_{ind\_abandon_l}). \tag{12}$$

To interpret Eq. 12, note that it is simply taking an average over the population of the expected number of times per generation that an individual abandons a social partner for a new encounter.

We check that our explicit M-term functions (Eqs. 4–12) are formulated correctly by deriving the population average absolute fitness $\left(\sum_{l=1}^{L_{max}}(x_{l0}w_{l0}+x_{l1}w_{l1})\right)$ as an explicit function of coalescence probabilities, and checking that this equals 1. We reiterate, however, that these M-term functions are only exact for either: the case where deme size is infinite ($N=\infty$), as this eliminates stochastic

variation in the genetic composition of demes; or, the case where there is no partner search ($\alpha=0$), as this means that stochastic variation in the genetic composition of demes, despite existing (for finite $N$), has no effect on the expected identity of social partners, and therefore has no effect on selection. We note that, in the special $\alpha=0$ case (no partner search), our model mathematically reduces to the weak selection version of the model considered by R&R (though, as explained in 'Model assumptions', we prefer a slightly different verbal description of the equations to R&R). We note that, in 'Finite population (agent-based) simulation', we use an agent-based simulation to verify that the theoretical conclusions drawn from this model tend to hold even when deme size ($N$) is low.

*Demographic parameters.* Having written our high-level "M-terms" in terms of coalescence (IBD) probabilities, we now finish closing our selection equations (Eqs. 2 and 3) by writing our coalescence probabilities ($F$, $\varphi$, $\gamma$) in terms of fundamental demographic parameters. Our demographic parameters are $N$ (deme size), $m$ (migration rate) and $r$ (recombination rate). These explicit coalescence probabilities were given in R&R, and we follow their derivations exactly.

To get an expression for $F$ (probability of being identical by descent, IBD, at a given locus), we first write a recursion describing how $F$ changes across a generation (under the assumption of weak selection and mutation)[7]:

$$F' = (1-m)^2\left(\frac{1}{N}+\left(1-\frac{1}{N}\right)F\right). \tag{13}$$

We derived Eq. 13 by drawing two (haploid) individuals from a deme (central pair). Each individual in the central pair has a single (haploid) parent. With probability $1-(1-m)^2$, the haploid parents were in different demes from each other, meaning the central pair's genes are not IBD (coalescence takes an infinite amount of time). With probability $(1-m)^2$, the parents were in the same deme, meaning the central pair's genes may be IBD (coalescence may occur in finite time). Given that the parents were in the same deme, there is a $\frac{1}{N}$ chance that the two parents are in fact the same individual (i.e. the same parent gave rise to the both individuals in the central pair), meaning the central pair is IBD with certainty (coalescence occurs). Conversely, given still that the parents were in the same deme, there is a $1-\frac{1}{N}$ chance that the two parents are different individuals, meaning the central pair is IBD with probability $F$ (coalescence occurs in finite time with probability $F$). Combining these potentialities gives the recursion (Eq. 13).

We solve Eq. 13 to get the following equilibrium expression for the single-locus IBD probability ($F$):

$$F = \frac{(1-m)^2}{(1-m)^2+N(1-(1-m)^2)}. \tag{14}$$

We obtain expressions for $\varphi$, $\gamma$ and $G$ in a similar way, by writing recursions describing how they change across a generation, and then solving them. Full details of this methodology, including the recursions for $\varphi$, $\gamma$ & $G$, are given in the Supplementary Information of R&R, in the section titled 'Probabilities of Coalescence'. We do not reproduce the details here, nor do we write out the full equilibrium expressions for $\varphi$, $\gamma$ & $G$, which are too long to be illuminating.

Instead, in Supplementary Fig. 2, we plot $\varphi$, $\gamma$ and $G$, alongside $F$, for different parameter values, to show how these functions behave. We see that the coalescence probabilities ($F$, $G$, $\varphi$, $\gamma$) decrease with migration, from a maximum of $F$, $G$, $\varphi$, $\gamma=1$ (groups comprised solely of full-kin) when there is no migration ($m=0$), to a minimum of $F$, $G$, $\varphi$, $\gamma=0$ (groups comprised solely of non-kin) when there is certain migration ($m=1$). It is intuitive that migration should have this effect—the more that individuals migrate, the less likely they are to interact with individuals with genes that are identical by descent (relatives).

When recombination rate between two loci is zero ($r=0$; Supplementary Fig. 2a), the probabilities of single and two-locus coalescence converge ($F=\varphi$), as do the probabilities of one-locus-three-individual and two-locus-three-individual coalescence ($G=\gamma$). This is because, with zero recombination ($r=0$), genes at each of the two loci are inherited as a single unit. For increased recombination (Supplementary Fig. 2b, c), the probability of two-locus coalescence ($\varphi$) falls below the probability of single-locus coalescence ($F$), and the probability of two-locus-three-individual coalescence ($\gamma$) falls below the probability of one-locus-three-individual coalescence ($G$). This is because genes at the two loci are less likely to be inherited as a single unit, meaning identity by descent at one locus gives less information about identity at the second locus.

Intuitively, the probability of single-locus coalescence is greater if the locus is measured in two ($F$) rather than three ($G$) individuals. Analogously, the probability of two-locus coalescence is greater if the two loci are measured in two ($\varphi$) rather than three ($\gamma$) individuals. Coalescence probabilities ($F$, $\varphi$, $\gamma$, $G$) decrease with deme size ($N$) (Supplementary Fig. 2d). The reason for this is that, with more individuals in a deme, the probability that a given individual in the deme is derived (in recent ancestral history) from that deme is reduced, meaning the probability that any two individuals drawn from that deme have IBD genes is reduced.

Having expressed coalescence probabilities in terms of fundamental model parameters, our equations for the effect of selection on genotype frequencies (Eqs. 2–14) are now closed, meaning there is enough information contained in them that they can be used to calculate genotype frequencies in the next time step (dynamically sufficient).

*Recombination.* Having formalised the effect of selection, we now formalise the effect of recombination on generational genotype frequency change (from $x_{ij}{}'$ to $x_{ij}{}''$). $k$ denotes the alternative allele to $j$ at the trait locus (i.e. $k = 1$ if $j = 0$; $k = 0$ if $j = 1$).

$$
\begin{aligned}
x_{ij}{}'' = \; & x_{ij}{}'(2F - \varphi)(1 - m)^2 \\
& + \left(1 - (2F - \phi)(1 - m)^2\right)\left(x_{ij}{}'\left(x_{ij}{}' + x_{ik}{}' + (1 - r)\sum_{l \neq i}^{L_{\max}} x_{lk}{}' + \sum_{l \neq i}^{L_{\max}} x_{lj}{}'\right)\right. \\
& \left. + rx_{ik}{}'\sum_{l \neq i}^{L_{\max}} x_{lj}{}'\right).
\end{aligned}
\tag{15}
$$

To interpret this equation, note that, of all zygotes (haploid-haploid associations) that form, a proportion of these—given by $(2F - \varphi)(1 - m)^2$—comprise haploid components that are identical by descent at either one, or both, of the tag and trait loci (specifically, $\varphi(1 - m)^2$ are IBD at both loci; $2(F - \varphi)(1 - m)^2$ are IBD at just one locus). When zygotes are comprised of haploid components that share alleles at one or both loci, disassociation of the zygote into haploid offspring does not result in genotype frequency change, regardless of whether recombination occurred in the zygote. This gives rise to the first term in Eq. 15—it states that, with probability $(2F - \varphi)(1 - m)^2$, recombination does not alter genotype frequency ($x_{ij}{}'' = x_{ij}{}'$).

A proportion of zygotes—given by $1 - (2F - \phi)(1 - m)^2$—comprise haploid components that are not identical by descent (non-IBD) at either the tag or trait locus. Of these zygotes, a proportion, given by $x_{ij}{}'^2$, comprise two $ij$ haploid components. These zygotes disassociate exclusively into $ij$ haploid offspring, meaning the first term in the brackets following $\left(1 - (2F - \phi)(1 - m)^2\right)$ is $x_{ij}{}'^2$.

The proportion of non-IBD zygotes comprising an $ij$ haploid component and an $ik$ haploid component (same tag; different trait allele) is $2x_{ij}{}'x_{ik}{}'$, and half of the haploid progeny from these associations have the $ij$ genotype, regardless of recombination, meaning the second term in the brackets following $\left(1 - (2F - \phi)(1 - m)^2\right)$ is $x_{ij}{}'x_{ik}{}'$.

The proportion of non-IBD zygotes comprising an $ij$ haploid, and a haploid with a different tag and trait allele, is $2x_{ij}{}'\sum_{l \neq i}^{L_{\max}} x_{lk}{}'$. From these zygotes, the proportion $(1 - r)/2$ of haploid progeny have the $ij$ genotype, meaning the third term in the brackets following $(1 - (2F - \phi)(1 - m)^2)$ is $(1 - r)x_{ij}{}'\sum_{l \neq i}^{L_{\max}} x_{lk}{}'$.

The proportion of non-IBD zygotes comprising an $ij$ haploid, and a haploid with a different trait allele but the same tag, is $2x_{ij}{}'\sum_{l \neq i}^{L_{\max}} x_{lj}{}'$. From these zygotes, half of the haploid progeny have the $ij$ genotype, regardless of recombination, meaning the fourth term in the brackets following $(1 - (2F - \phi)(1 - m)^2)$ is $x_{ij}{}'\sum_{l \neq i}^{L_{\max}} x_{lj}{}'$.

Finally, the proportion of zygotes comprising an $ik$ haploid, and a haploid with a different tag and different trait allele, is $2x_{ik}{}'\sum_{l \neq i}^{L_{\max}} x_{lj}{}'$. From these zygotes, the proportion $r/2$ of haploid progeny have the $ij$ genotype, meaning the fifth term in the brackets following $(1 - (2F - \phi)(1 - m)^2)$ is $rx_{ik}{}'\sum_{l \neq i}^{L_{\max}} x_{lj}{}'$.

*Mutation.* Having formalised the effects of selection and recombination, we now formalise the effect of mutation on generational genotype frequency change (from $x_{ij}{}''$ to $x_{ij}{}'''$). $k$ denotes the alternative allele to $j$ at the trait locus (i.e. $k = 1$ if $j = 0$; $k = 0$ if $j = 1$).

$$
x_{ij}{}''' = x_{ij}{}''\left(1 - \mu_{Trait} - \mu_{Tag}\right) + x_{ik}{}''\mu_{Trait} + \frac{\sum_{l \neq i}^{L_{\max}} x_{lj}{}''\mu_{Tag}}{L_{\max} - 1}.
\tag{16}
$$

We reiterate that, unless stated otherwise, tag mutation is assumed to be absent ($\mu_{Tag} = 0$), as it could lead to spurious (non-adaptive) tag diversity. We will reintroduce tag mutation later (see 'Finite population (agent-based) simulation'), in an agent-based simulation of the model.

*Dynamically sufficient recursions.* Our equations for selection, recombination and mutation, taken together, give recursions describing how genotype frequencies change over a single generation (from $x_{ij}$ to $x_{ij}{}'''$). By iterating these recursions over many generations, we can work out the amount of helping and tag diversity that evolves at evolutionary equilibrium.

**Individual-level analysis (finding the right area of parameter space).** Before presenting the numerical results of the population genetic model, we take a step back to ask, when does inclusive fitness theory predict that kin discrimination based on genetic cues will evolve? Inclusive fitness theory predicts that conditional (tag-based) altruism may only evolve if it leads to a higher inclusive fitness payoff, to the actor, per social interaction, than both indiscriminate (not tag-based) altruism and indiscriminate defection. In this section, we show when this is the case.

We note at the outset, however, that this is a necessary but not a sufficient condition for kin discrimination based on genetic cues to evolve. If recognising kin is too costly, either because it reduces social interaction rate too much, or because searching for partners is too costly, then genetic kin recognition will not evolve, even if kin discrimination based on genetic cues results in a higher (inclusive fitness) payoff per social interaction than indiscriminate strategies.

$IF_{conditional}$. First, we calculate the inclusive fitness payoff of conditional (tag-based) altruism, which we denote by $IF_{conditional}$. The inclusive fitness payoff of an action is calculated by: (1) identifying all individuals (including the actor) affected by the action; (2) weighting each of these individuals according to their genetic relatedness to the actor (similarity at the trait locus); (3) summing the (relatedness-weighted) fitness consequences of the action across all of the affected individuals. We note that fitness consequences are measured here relative to the non-social case (i.e. the fitness consequences of defecting[1]). For more discussion of our inclusive fitness payoff measure, and how it relates to a recent paper by Levin and Grafen[48], see the aside at the end of this section ('Individual-level analysis (finding the right area of parameter space)').

An act of conditional (tag-based) altruism has consequences for: (1) the altruist (actor); (2) its recipient (i.e. the altruist's ultimate social partner after the possibility of partner search); (3) competitors (those who suffer fecundity losses as a result of altruism exhibited by the actor). We note here that the altruist (actor) and its recipient are tag-matched—if they were not tag-matched, there would be no social interaction (no altruism exhibited), and therefore no fitness effect! However, the altruist (actor) and its competitors may be tag-mismatched.

The relatedness between the altruist (actor) and another individual ('affected individual') is given by:

$$
R = \frac{\lambda - \bar{p}}{1 - \bar{p}},
\tag{17}
$$

where $\bar{p}$ gives the population frequency of the conditional helping allele, and $\lambda$ gives the probability that the affected individual also has the conditional altruism allele[29]. Equation 17 is a simplified version of Eq. 7 in Grafen[29]; see ref. [29] for its derivation.

We see by plugging in $\lambda = 1$ that the actor is related to itself by 1 (complete genetic similarity).

To work out the relatedness between the actor (altruist bearing a given tag $i$), and its recipient, we need to plug in $\lambda = \frac{M_{ind\_helped,i}}{M_{interact,i}}$, which is the per-opportunity (i.e. per generation) probability that the actor's recipient is also an altruist. To interpret this expression for $\lambda$, note that the denominator gives the per-encounter probability of encountering a tag-matched individual, and the numerator gives the per-encounter probability of encountering a tag-matched altruist. Plugging this into Eq. 17, and writing it explicitly in terms of coalescence probabilities, we obtain the following expression for relatedness between actors and their recipients ($R_{tag}$). Following Grafen[29], we can interpret $R_{tag}$ as a regression of the actor's genic value on its recipient's (social partner's) genic value, where the regression line is forced through the population mean genic value (Supplementary Fig. 3).

$$
R_{tag} = \frac{\frac{x_i(-2Fp_i + F + (p_i - 1)\phi + p_i) + Fp_i - p_i\phi + \phi}{x_i(-F) + x_i + F} - \bar{p}}{1 - \bar{p}}.
\tag{18}
$$

We can write coalescence probabilities in terms of fundamental demographic parameters to obtain a fully explicit expression for $R_{tag}$. We plot this function to illustrate its behaviour (Supplementary Fig. 4a). We see that $R_{tag}$ increases as the actor's tag decreases in population frequency (the tag becomes a more reliable indicator of kinship), especially with reduced recombination ($r$) between tag and trait (the tag becomes a more reliable indicator of trait identity). $R_{tag}$ decreases with migration and deme size (social interactants are less likely to be relatives).

A notable feature of the relatedness expression (Eq. 18) is that, if helpers are evenly distributed across tags (no linkage disequilibrium), such that $p_i = \bar{p}$, helper frequency $(p_i, \bar{p})$ 'drops out' of the expression for relatedness. This leads to a relatedness of

$$
R_{tag}\big|_{p_i = \bar{p}} = \frac{\phi + (F - \phi)x_i}{F + (1 - F)x_i},
\tag{19}
$$

which is invariant (unchanging) with respect to the frequency of the conditional helping allele $(p_i, \bar{p})$. To interpret Eq. 19, note that the right-hand-side simply gives the probability that an individual's social partner is identical by descent at the trait locus[49].

Let us assume that each of the $L_{max}$ available tags is held at equal frequency in the population (we relax this assumption later, when solving our population genetic models), and that there is no linkage disequilibrium (i.e. $p_i = \bar{p}$). This means that each tag is at the population frequency $1/L_{max}$. Furthermore, it means that, if there are more available tags (increased $L_{max}$), each given tag is rarer in the population, resulting in a higher relatedness between altruists and their recipients ($R_{tag}$). Specifically, if each of the $L_{max}$ tags are maintained at equal frequency, relatedness at equilibrium is given by:

$$
R_{tag}\big|_{p_i = \bar{p}, x_i = 1/L_{\max}} = \frac{F + \phi(L_{\max} - 1)}{1 + F(L_{\max} - 1)}.
\tag{20}
$$

Having calculated the relatedness between the actor (altruist bearing a given tag $i$) and its recipient, we now calculate the (expected) relatedness between the actor (altruist bearing a given tag $i$) and its competitors. To do so, we first note that, with probability $m$, the competitor is drawn from a different deme to the actor (altruist),

resulting in a relatedness (obtained by be substituting $\lambda = \bar{p}$ into Eq. 17) of zero. With probability $(1-m)(1/N)$, the competitor is also the actor (they are the same individual), resulting in a relatedness of 1. With probability $(1-m)(1/N)$, the competitor is the actor's recipient, resulting in a relatedness of $R_{tag}$ (Eq. 18).

With probability $(1-m)(1-2/N)$, the competitor is in the same deme as the actor but is neither the actor herself, nor the actor's recipient. To calculate actor-competitor relatedness in these cases, we substitute $\lambda = \frac{\gamma}{M_{interact_i}} + \frac{(F-\gamma)\bar{p}}{M_{interact_i}} + (F-\gamma)\left(\frac{x_i+(1-x_i)\alpha M_{interact_i}}{M_{interact_i}}\right) + (1-2F+\gamma)\left(\frac{x_i+(1-x_i)\alpha M_{interact_i}}{M_{interact_i}}\right)\bar{p}$ into Eq. 17, to obtain $\left(\frac{1}{1-\bar{p}}\right)\left(\left(\frac{\gamma}{M_{interact_i}} + \frac{(F-\gamma)\bar{p}}{M_{interact_i}} + (F-\gamma)\left(\frac{x_i+(1-x_i)\alpha M_{interact_i}}{M_{interact_i}}\right) + (1-2F+\gamma)\left(\frac{x_i+(1-x_i)\alpha M_{interact_i}}{M_{interact_i}}\right)\bar{p}\right)-\bar{p}\right)$. To make sense of our $\lambda$ expression, note that $\frac{\gamma}{M_{interact_i}}$ (first term) gives the proportion of the altruist's (actor's) social interactions where the competitor is IBD (to the actor) at the trait locus, and IBD (to its ultimate partner) at the tag locus, and therefore is an altruist. $\frac{(F-\gamma)\bar{p}}{M_{interact_i}}$ (second term) gives the proportion of the altruist's (actor's) social interactions where the competitor is IBD (to its ultimate partner) at the tag locus, but not IBD (to the actor) at the trait locus, but is an altruist nevertheless. $(F-\gamma)\left(\frac{x_i+(1-x_i)\alpha M_{interact_i}}{M_{interact_i}}\right)$ (third term) gives the proportion of the altruist's (actor's) social interactions where the competitor is not IBD (to its ultimate partner) at the tag locus, but is IBD (to the actor) at the trait locus, and therefore is an altruist. $(1-2F+\gamma)\left(\frac{x_i+(1-x_i)\alpha M_{interact_i}}{M_{interact_i}}\right)\bar{p}$ (fourth term) gives the proportion of the altruist's (actor's) social interactions where the competitor is not IBD (to its ultimate partner) at the tag locus, nor IBD (to the actor) at the trait locus, but is an altruist nevertheless.

By averaging over each of these possible scenarios, we obtain the following expected relatedness between actors and competitors. Again, following Grafen[29], we can interpret $R_{competitor}$ as a regression of the actor's genic value on the competitor's genic value, where the regression line is forced through the population mean genic value (Supplementary Fig. 5).

$$R_{competitor} = (1-m)\left(\frac{1}{N} + \frac{R_{tag}}{N} + \left(\frac{1-\frac{2}{N}}{1-\bar{p}}\right)\left(\left(\frac{\gamma}{M_{interact_i}} + \frac{(F-\gamma)\bar{p}}{M_{interact_i}}\right.\right.\right.$$
$$+ (F-\gamma)\left(\frac{x_i+(1-x_i)\alpha M_{interact_i}}{M_{interact_i}}\right) \tag{21}$$
$$\left.\left.\left. + (1-2F+\gamma)\left(\frac{x_i+(1-x_i)\alpha M_{interact_i}}{M_{interact_i}}\right)\bar{p}\right)-\bar{p}\right)\right).$$

We can write coalescence probabilities in terms of fundamental demographic parameters to obtain a fully explicit expression for $R_{competitor}$. We plot this function to illustrate its behaviour (Supplementary Fig. 4b). We see that, as tag frequency $(x_i)$ decreases, $R_{competitor}$ increases, though not as starkly as $R_{tag}$ increases. Recombination $(r)$, migration $(m)$ and deme size $(N)$, as well as reducing the relatedness between social partners $(R_{tag})$, and for analogous reasons, also reduce the relatedness between competitors $(R_{competitor})$.

Partner search $(\alpha)$ reduces the relatedness between actors and competitors $(R_{competitor})$. The reason for this is, with high partner search $(\alpha)$, actors that are poorly related to their deme still, through partner search, have a good chance of socially interacting. This means that these incidences, where the actor and competitor are poorly related, are 'counted' in the relatedness calculation, reducing $R_{competitor}$. In contrast, with low partner search $(\alpha)$, actors that are poorly related to their deme are unlikely to socially interact (due to tag mismatching), meaning these incidences are not 'counted' in the relatedness calculation, resulting in a higher $R_{competitor}$. In this way, partner search $(\alpha)$ can actually reduce kin competition (reduce $R_{competitor}$), increasing the area of parameter space where conditional (tag-based) helping is favoured.

If helpers are evenly distributed across tags (no linkage disequilibrium), such that $p_i = \bar{p}$, helper frequency $(p_i, \bar{p})$ 'drops out' of the expression for relatedness, to give:

$$R_{competitor}|_{p_i=\bar{p}} = (1-m)\left(\frac{1}{N} + \frac{R_{tag}|_{p_i=\bar{p}}}{N} + \left(1-\frac{2}{N}\right)\right.$$
$$\left.\left(\frac{\gamma + x_i(F-\gamma) - \alpha(1-x_i)(\gamma-F^2)}{F+(1-F)x_i}\right)\right), \tag{22}$$

which is invariant (unchanging) with respect to the frequency of the conditional helping allele $(p_i, \bar{p})$. To interpret Eq. 22, note that the right-hand-side gives the probability that an individual, having engaged in a social interaction, is identical by descent at the trait locus to a random individual drawn from either the local deme (probability $1-m$) or a non-native deme (probability $m$). In the absence of partner search $(\alpha=0)$, this reduces to $R_{competitor}|_{p_i=\bar{p}, \alpha=0} = \frac{1}{N} + \frac{R_{tag}}{N} + \left(1-\frac{2}{N}\right)\left(\frac{\gamma+x_i(F-\gamma)}{F+(1-F)x_i}\right)$.

Let us assume that each of the $L_{max}$ available tags are held at equal frequency in the population, and that there is no linkage disequilibrium (i.e. $p_i = \bar{p}$). This gives a

relatedness of:

$$R_{competitor}|_{p_i=\bar{p}, x_i=1/L_{max}} = (1-m)\left(\frac{1}{N} + \frac{R_{tag}|_{p_i=\bar{p}}}{N}\right.$$
$$\left. + \left(1-\frac{2}{N}\right)\left(\frac{\gamma + x_i(F-\gamma) - \alpha(1-x_i)(\gamma-F^2)}{F+(1-F)x_i}\right)\right). \tag{23}$$

Having derived $R_{tag}|_{p_i=\bar{p}, x_i=1/L_{max}}$ and $R_{competitor}|_{p_i=\bar{p}, x_i=1/L_{max}}$, we can now write down the inclusive fitness payoff of conditional (tag-based) altruism, on the assumption that tags have equalised in frequency and helper proportions:

$$IF_{conditional} = R_{tag}|_{p_i=\bar{p}, x_i=1/L_{max}} b - (b-c)R_{competitor}|_{p_i=\bar{p}, x_i=1/L_{max}} - c, \tag{24}$$

where $R_{tag}|_{p_i=\bar{p}, x_i=1/L_{max}}$ is given in Eq. 20 and $R_{competitor}|_{p_i=\bar{p}, x_i=1/L_{max}}$ is given in Eq. 23.

$IF_{defection}$. Having derived, on the assumption that tags have equalised in frequency and helper proportions, the inclusive fitness payoff of conditional altruism (kin discrimination based on genetic cues), we can now ask when kin discrimination based on genetic cues will be favoured over defection. The inclusive fitness payoff of defection is zero, by definition, as it is a non-social trait[1]. This means that conditional (tag-based) altruistic helping will be favoured over defection whenever the following condition is satisfied (obtained by evaluating $IF_{conditional} > IF_{defection}$). Failure to satisfy this condition implies that defection will persist at evolutionary equilibrium:

$$R_{tag}|_{p_i=\bar{p}, x_i=1/L_{max}} b - (b-c)R_{competitor}|_{p_i=\bar{p}, x_i=1/L_{max}} - c > 0, \tag{25}$$

where $R_{tag}|_{p_i=\bar{p}, x_i=1/L_{max}}$ is given in Eq. 20 and $R_{competitor}|_{p_i=\bar{p}, x_i=1/L_{max}}$ is given in Eq. 23. This condition is a version of Hamilton's rule, which has been generalised from the more familiar form $Rb-c > 0$ to account for the effects of altruism on competitors[32,33,35]. Equation 25 is replicated in the main text, with simpler notation, as Eq. 1 (note that in Eq. 1, $R_{tag}|_{p_i=\bar{p}, x_i=1/L_{max}}$ is written simply as $R_{tag}$, and $R_{competitor}|_{p_i=\bar{p}, x_i=1/L_{max}}$ is written simply as $R_{comp}$).

$IF_{indiscriminate}$. To derive the inclusive fitness payoff for indiscriminate altruism $(IF_{indiscriminate})$, we assume that there is one single tag at fixation (no tag diversity; $x_i = 1, \bar{p} = p_i$). We evaluate our relatedness coefficients (Eqs. 18 and 21) under this assumption of no tag diversity, which gives $R_{tag}|_{x_i=1} = F$ and $R_{competitor}|_{x_i=1} = \frac{1}{N} + \frac{N-1}{N}F$. This leads to an inclusive fitness payoff of indiscriminate altruism, written explicitly in terms of model parameters, of:

$$IF_{indiscriminate} = -\left(\frac{m((b-c)(1-m) + c(N(2-m)))}{(m(2-m)(N-1)+1)}\right). \tag{26}$$

We can see by inspection of Eq. 26 that, given that helping is costly $(c > 0)$, the inclusive fitness payoff of indiscriminate altruism $(IF_{indiscriminate})$ is always negative (the numerator & denominator inside the brackets are both positive, meaning the overall expression is always negative).

Conditional (tag-based) helping confers a greater inclusive fitness return than indiscriminate helping whenever $IF_{conditional} > IF_{indiscriminate}$. Using Eqs. 24 and 26, we can evaluate this condition, and we see that it holds whenever there is diversity at the tag locus $(L_{max} > 1)$. This is intuitive—by discriminating who it interacts with, an individual is more likely to interact with kin, meaning there is a greater inclusive fitness return from helping. Therefore, as long as individuals are capable of differentiating individuals based on their tag $(L_{max} > 1)$, kin discrimination based on genetic cues will confer a greater inclusive fitness payoff, per social interaction, than indiscriminate helping.

Furthermore, given that the inclusive fitness payoff of indiscriminate helping is always negative, we can deduce that, in fact, indiscriminate helping will never evolve under these lifecycle assumptions—it is always less favourable than defection. This result was also given in R&R, and was first obtained in an inclusive fitness analysis of a very similar lifecycle by Taylor[33]. It is an example of a general result in social evolution theory, which is that, in 'homogeneous' population structures, where the scale of social interaction is equal to the scale of competition, kin selection (favouring altruism) is exactly offset by kin competition (disfavouring altruism), precluding the evolution of indiscriminate altruism[32,35,44,50–52].

*Predictions based on inclusive fitness theory.* This analysis reveals that, if tags evolve in such a way that their frequencies and helper proportions equalise, kin discrimination based on genetic cues will confer a greater inclusive fitness return, per social interaction, than indiscriminate strategies, wherever Eq. 25/1 is satisfied. However, we emphasise that kin discrimination based on genetic cues could evolve here; not that it will evolve here (it is a necessary rather than sufficient condition).

Specifically, if the costs of recognising kin (partner search cost/reduced social interaction rate) exceed the benefits (increased inclusive fitness payoff from a given social interaction), then tags will not equalise in frequency. Instead, tag diversity will be lost at equilibrium $(L^* = 1)$, and indiscriminate defection will evolve.

We have expressed these predictions, based on inclusive fitness theory, with minimal reference to genetic details. We will see that our prediction—that kin discrimination based on genetic cues evolves whenever Eq. 25/1 holds and there is a low cost of genetic kin recognition—is borne out by our explicit population genetic analyses. However, this prediction is high-level and somewhat vague, insofar that we have not examined the causal effects of model parameters such as partner search rate ($\alpha$), partner search cost ($c_{search}$), recombination ($r$), trait mutation ($\mu_{Trait}$), and so on, and we have not examined how evolution proceeds at the level of loci (tag & trait). For this, we need explicit genetic analyses, and we come onto this next.

*An aside regarding Levin and Grafen*[48]. We note as an aside that, when calculating the inclusive fitness payoff of a given action (trait), we followed Hamilton in measuring the fitness consequences of the trait relative to the hypothetical scenario where the actor does not express the trait[1]. A recent paper argues that, instead, when calculating the inclusive fitness payoff of a given action (trait), the fitness consequences of the trait should be measured relative to the population average trait value[48]. For instance, this proposed amendment to the calculation of inclusive fitness payoffs would mean, contrary to what we said in the above analysis, that indiscriminate defection could have an inclusive fitness payoff that is different from zero, because defection could still have fitness consequences relative to the population average trait value.

We stuck to Hamilton's original definition in our above calculations, for two reasons. Firstly, in the present model (and in the island model analysed below), the two approaches lead to quantitatively identical predictions regarding when kin discrimination based on genetic cues should be expected to evolve. Secondly, the equations are simpler and the mathematical argument is easier to follow when inclusive fitness calculations are made relative to case where the trait is absent (Hamilton's definition), rather than to the population average trait value (proposed amendment). More broadly, the two formulations are likely to give identical results in most models, with Hamilton's original definition only leading to potential pitfalls, and requiring amendment, in models where fitness effects combine non-additively[48,53,54].

**Full analysis (solving the model).** We numerically solve our population genetic model, for different parameter values ($N$, $m$, $r$, $\alpha$, $\mu_{Trait}$, $b$, $c$), and present the results[55]. We are primarily interested in whether, and when, kin discrimination based on genetic cues is found at equilibrium (multiple tags + conditional helping).

*Initial conditions.* For each numerical simulation of our population genetic recursions ('run'), we assume that one tag is initially dominant, and the remaining tags are rare. Specifically, we set the initial frequency of one tag to 0.9, and we randomly distribute the remaining 0.1 amongst the other $L_{max} - 1$ tags (with each tag frequency above zero). We also assume that the helping allele is initially rare (we set the initial helper proportion of each tag to 0.1). The start point of our runs is therefore indiscriminate defection. This allows us to examine the evolution as well as the maintenance of kin discrimination based on genetic cues. It can also be thought of as an unfavourable scenario for the evolution of kin discrimination based on genetic cues (negligible initial tag diversity).

We note that our model is only defined for certain parameter combinations. Specifically, our model is only defined for parameter combinations in which fitness is non-negative. The reason for this is simply that fitness is defined as "number of offspring who survive one iteration of the lifecycle", meaning negative fitness (negative offspring) is nonsensical. For a given rate of partner search ($\alpha$), the maximum generational cost of partner search is given by $\frac{\alpha}{1-\alpha} c_{search}$ (this is obtained by evaluating $M_{ind\_abandon_i}$ for $F = 0$ and $x_i = 0$ and multiplying by $c_{search}$). This leads to a lowest achievable fitness of $1 - \frac{\alpha}{1-\alpha} c_{search} - c$. We present results for parameter combinations in which $1 - \frac{\alpha}{1-\alpha} c_{search} - c > 0$, as these are the parameter combinations in which we can be sure that fitness is non-negative across all individuals in the population at all times, meaning our model is defined.

*Summary statistics.* For each run, we track two summary statistics. Summary statistics are measured across a given time period, starting at generation $T_{end} - T_{interval}$ and ending at generation $T_{end}$, where $T_{interval}$ gives the length of the time period, and $T_{end}$ gives the end-point ($T_{interval} \geq 0$ & $T_{end} \geq 1$). The two summary statistics are:

- Number of segregating tags.

- Frequency of the conditional helping allele.

Below, we give their precise mathematical definitions (Eqs. 27 and 28). We take $x_{ijt}$ to be the population frequency of genotype $ij$ (tag $i$; trait $j$) in generation $t$.

First, we calculate the average-over-time-and-over-tags tag frequency. Or more precisely, if an individual (tag not specified) is randomly chosen from a population, from a generation that is randomly chosen from within $T_{end} - T_{interval}$ to $T_{end}$, we calculate the expected frequency of the individual's tag as:

$$\frac{1}{L|_{T_{end}, T_{interval}}} = \frac{\sum_{i=1}^{L_{max}} \sum_{t=T_{end}-T_{interval}}^{T_{end}} \left( \sum_{j=0}^{1} x_{ijt} \right)^2}{1 + T_{interval}}. \tag{27}$$

The average-over-time number of segregating tags $\left( L|_{T_{end}, T_{interval}} \right)$ is then simply given by the inverse of the right-hand side of Eq. 27. This is our first summary statistic. We note that this metric is not the countable number of tags—such a measure would be misleading, because it would give equal weight to tags that are limitingly rare and exceedingly common. Rather, this metric is an effective tag number based on tag frequencies.

The average frequency of the helping allele can be written as follows (Eq. 28). This is our second summary statistic.

$$coop|_{T_{end}, T_{interval}} = \frac{\sum_{i=1}^{L_{max}} \sum_{t=T_{end}-T_{interval}}^{T_{end}} x_{i1t}}{1 + T_{interval}}. \tag{28}$$

We define equilibrium as the point at which the two summary statistics are no longer changing. At equilibrium, then, $T_{interval}$ and $T_{end}$ are both sufficiently large that, for a further increase in either $T_{interval}$ or $T_{end}$, there is negligible change in either of the summary statistics. Therefore, at equilibrium, $T_{interval}$ and $T_{end}$ are large, but they are arbitrary insofar that their precise values do not non-negligibly change any of the summary statistics. We can therefore drop the $T_{end}$, $T_{interval}$ indexing when writing our equilibrium summary statistics: $L^*$ & $coop^*$. We obtain these equilibrium summary statistics by iterating our recursions for a sufficiently long period of time, and using a sufficiently large interval to calculate them with respect to (sufficiently large $T_{interval}$ and $T_{end}$).

Having introduced our summary statistics, we can now move on to the results. Results for the case where partner search is uncostly and unrestricted ($\alpha = 1$ & $c_{search} = 0$) are plotted, for different parameter values, in Supplementary Fig. 6 ($r$ is varied across y-axes; $m$ is varied across x-axes; $N$ & $\mu_{Trait}$ are varied across panels a–d).

*Evolution of genetic kin recognition when partner search is uncostly and unrestricted* ($\alpha = 1$ & $c_{search} = 0$). R&R showed that, when individuals are incapable of searching for social partners ($\alpha = 0$), and when selection is weak, individuals bearing rare tags suffer a starkly reduced social interaction rate relative to individuals bearing common tags, and as a result, rare tags are always disfavoured relative to common tags, precluding genetic kin recognition—indiscriminate defection always evolves. We recover this result.

However, when individuals can search for social partners at no cost ($\alpha = 1$ & $c_{search} = 0$), and when there is a sufficient amount of mutation at the trait locus ($\mu_{Trait}$; only a small amount of trait mutation is required), genetic kin recognition evolves at evolutionary equilibrium whenever our Hamilton's Rule condition (Eq. 25/1) is satisfied, and indiscriminate defection evolves whenever our Hamilton's Rule condition (Eq. 25/1) is not satisfied (Supplementary Fig. 6). Insufficient trait mutation ($\mu_{Trait}$) can reduce the region where genetic kin recognition is found relative to the region where our Hamilton's Rule condition (Eq. 25/1) is satisfied—the reason for this, which is subtle, is given in the 'Key points and implications' section below.

We noted previously that satisfaction of the condition in Eq. 25/1 will only lead to the evolution of genetic kin recognition if tags equilibrate in frequency (Crozier's paradox is overcome), and this will require a low cost of kin recognition (high $\alpha$ & low $c_{search}$). We have seen that, given enough trait mutation ($\mu_{Trait}$), there is a perfect correspondence between the *predicted* (Eq. 25/1) and *actual* regions of parameter space where genetic kin recognition, as opposed to indiscriminate defection, evolves. This shows that trait mutation ($\mu_{Trait}$) and cheap partner search (high $\alpha$, low $c_{search}$) create the necessary context for our Hamilton's Rule condition (Eq. 25/1) to work as a predictor of long-term social evolution.

*Genetic constraint on tag diversification.* We make two points about tag availability ($L_{max}$). To reiterate, our parameter $L_{max}$ gives the maximum number of tags that may segregate in the population—it therefore captures an evolutionary constraint on tag diversification. In Supplementary Fig. 6, this parameter is set to $L_{max} = 100$. Our first point is that, if the evolutionary constraint on tag availability is relaxed, leading to a greater $L_{max}$, then each tag has the potential to (simultaneously) achieve a lower relatedness (given by $1/L_{max}$), which means that a greater relatedness can potentially be achieved between social partners ($R_{tag}$; see Supplementary Fig. 4 for the exact relationship between tag frequency and coefficients of relatedness). This means that, for increased tag availability ($L_{max}$), kin discrimination based on genetic cues is favoured (Eq. 25/1 satisfied) across an increased region of parameter space. If the evolutionary constraint on tag diversity is relaxed completely ($L_{max} \to \infty$), then kin discrimination based on genetic cues is favoured (Eq. 25/1 satisfied) across most of the parameter space (i.e. across most values of $m$, $N$ and $r$).

Our second point about tag availability is that, when partner search is cheap and prevalent (very high $\alpha$ & low $c_{search}$), and there is sufficient trait mutation ($\mu_{Trait}$), all available tags are maintained. For instance, if $\alpha$ and $\mu_{Trait}$ are sufficiently high and $c_{search}$ sufficiently low, when there are 100 available tags, all 100 tags are maintained at equilibrium when kin discrimination based on genetic cues evolves (Supplementary Fig. 6). Conversely, when there are e.g. 1000 available tags, all 1000 tags are maintained at equilibrium when kin discrimination based on genetic cues evolves. This arises because, if $\alpha$ and $\mu_{Trait}$ are sufficiently high and $c_{search}$ sufficiently low, there is genuine negative frequency dependence at the tag locus— the rarest tag is the fittest—which means that all available tags are 'striving' to

obtain the lowest population frequency possible, meaning all are maintained (no tags are displaced).

Putting these two points about tag availability together, we can see that, if there is no evolutionary constraint on tag diversification ($L_{max} \rightarrow \infty$); and $\alpha$, $\mu_{Trait}$ & $1/c_{search}$ are sufficiently high; genetic kin recognition will be: (i) evolvable across very broad parameter space, and (ii) open ended, leading to limitless tag diversification and hence a highly precise capacity of individuals to discriminate genealogical kin from non-kin. Of course, unlimited and non-costly partner search ($\alpha = 1$ & $c_{search} = 0$), and an unrestricted ability to differentiate tags ($L_{max} \rightarrow \infty$), will not literally apply to natural systems, so infallible kin discriminatory systems should not be generally expected.

*Evolution of genetic kin recognition when partner search is costly or restricted ($\alpha < 1$ or $c_{search} > 0$).* A limitation on the capacity of individuals to search for partners ($\alpha < 1$), and/or a cost of partner search ($c_{search}$), reduces the region where genetic kin recognition is found relative to the region where our Hamilton's Rule condition (Eq. 25/1) is satisfied.

These $\alpha < 1$ and $c_{search} > 0$ results are explained more fully in the main text (Figs. 4 and 5). We will note here, though, that the stability of genetic kin recognition is compromised less by an increase in the partner search cost ($c_{search}$) than by a decrease in the encounter parameter ($\alpha$). A principal reason for this is that, when the encounter parameter ($\alpha$) is reduced, individuals with rare tags suffer a cost of a reduced social interaction rate. But an additional factor is that the benefit of tag rarity—increased likelihood of receiving help in a given social interaction—also shines through less strongly, because there are few social opportunities from which to reap this benefit! In contrast, when the partner search cost ($c_{search}$) is increased, individuals with rare tags suffer an increased total cost of finding a social partner to interact with. But the benefit of tag rarity—increased likelihood of receiving help in a given social interaction—shines through more strongly, because individuals with rare tags still have many social opportunities from which to reap this benefit.

**Finite population (agent-based) simulation.** One issue with our mathematical model, which we are yet to address, is that we ignored stochastic variation in the genetic composition of demes. We did this to make our model mathematically tractable. This approximation does not matter when deme size ($N$) is infinite, because stochastic deme variation decreases to zero as deme size ($N$) increases to infinity. Also, this approximation does not matter when there is no partner search ($\alpha = 0$), because in this case, selection proceeds according to the expected genetic composition of demes, meaning stochastic deme variation would have no effect on evolution, even if we had modelled it (the genotype frequency recursions, once simplified, would be the same).

However, an inaccuracy is introduced when demes are finite and there is some partner search, and this inaccuracy increases as deme size ($N$) is reduced and partner search ($\alpha$) is increased. To see why this is the case, note that, if deme size ($N$) is low, an individual with a rare tag will, under conditions of low relatedness, often enter into demes with no tag-matched individuals. In such incidences, no amount of partner search ($\alpha$) will allow the individual with the rare tag to obtain a tag-matched partner to socially interact with. Conversely, if we ignore stochastic deme variation and assume that each deme has the same genetic variation in it as the population as a whole, as we did when constructing our mathematical model, then the rare-tag individual will be able to find a tag-matched social partner, given enough partner search ($\alpha$). Ignoring stochastic deme variation is therefore likely to make partner search ($\alpha$) more effective for finding a social partner to interact with, for purely artificial reasons. It is therefore pertinent that we verify our results when there is realistic (i.e. stochastic) deme variation. To do this, we need to construct an agent-based simulation.

We use an agent-based simulation to check the validity of our mathematical analysis, which was only technically accurate for: an infinite population; weak selection/mutation (low $b$, $c$, $c_{search}$, $\mu_{Trait}$, $\mu_{Tag}$); infinitely large demes ($N = \infty$). Our agent-based model is accurate for finite populations, stronger selection/mutation, and arbitrarily sized demes ($N$), including small ones. Furthermore, it is stochastic rather than deterministic, meaning each individual reproduces, interacts and mutates probabilistically each generation. As deme size ($N$) increases and selection/mutation decreases (low $b$, $c$, $c_{search}$, $\mu_{Trait}$, $\mu_{Tag}$), our agent-based and mathematical models converge.

Stochasticity means that tags may be lost from the finite population (by drift) even if they are favoured by selection. This could lead to the false impression that genetic kin recognition is unreachable by natural selection (un-evolvable), when in fact, it is just prohibited by excessive genetic drift. Therefore, in our agent-based model, we permit mutation at the tag locus ($\mu_{Tag} > 0$), which means that tag diversity can be maintained by selection even if tags are repeatedly lost through drift.

We find that the results of our population genetic analysis (Supplementary Fig. 6) are recovered in our agent-based simulation (Supplementary Fig. 7). This shows that, although our mathematical analysis ignored stochastic deme variation, and assumed that populations are infinite and selection/mutation is weak, our conclusions tend to hold even when populations are finite, selection/mutation is stronger, and demes are small.

*Balancing selection in the finite population model.* Our finite population (agent-based) model permitted tag mutation. As we pointed out when constructing our mathematical model, an issue with this is that tag mutation could allow tag

diversity to be maintained by mutational force, even if it is opposed by selection (spurious tag diversity). Therefore, we need to undertake further analysis of our finite population (agent-based) model to rule out the possibility that tag diversity is being maintained by mutational force (spurious).

We run a version of our finite population (agent-based) model that sets tag mutation to zero ($\mu_{tag} = 0$). Each trial, we set the initial tag diversity to maximum (each of the $L_{max}$ tags at an initial population frequency of $1/L_{max}$), and then iterate over successive generations, stopping only when tag diversity is lost (one tag reaches fixation). For each trial, we record the time taken (generations passed) for tag diversity to be lost (*tag fixation time*).

Note that, in a finite population model such as the one considered here, tag fixation time will necessarily be bounded (not infinite)—that is, tag diversity will eventually be lost, owing to drift. However, if balancing selection were very strong relative to drift, tag fixation time would be very large—possibly too large to track computationally (requiring too many iterations of the simulation)! We therefore focus on relatively small population sizes, where drift is strong, meaning tag fixation time is always small enough to measure without prohibitively high computational power.

In the region of parameter space where tag diversity is maintained (e.g. when there is prevalent and cheap partner search; high $\alpha$, low $c_{search}$), the first step towards ruling out mutational force as the cause of stable tag diversity is checking that the tag fixation times in these regions of parameter space are greater than in their corresponding neutral scenarios, where demographic parameters ($N$, $\mu_{trait}$) are held constant but selection coefficients ($c$, $b$ and $c_{search}$) are set to zero (note that $\alpha$ can technically be set to anything in the neutral scenarios, because partner search has no consequence in the absence of selection). If the tag fixation time is greater when there is selection, relative to when selection is absent (corresponding neutral case), then the implication is that balancing selection is in operation, prolonging the time period within which tag diversity persists[7]. Another way of putting this is to say that balancing selection is implied if the *tag fixation ratio*, obtained by dividing tag fixation time under selection by its corresponding neutral tag fixation time, is greater than one.

However, a positive tag fixation ratio, though necessary, is not a sufficient condition for demonstrating that balancing selection is in operation and responsible for the maintenance of tag diversity. A problem is that a positive tag fixation ratio could also arise in scenarios where selection eliminates tag diversity (directional tag selection), but via an extended path, for instance characterised by unstable tag frequency spiralling, where tag frequencies go up and down over time, with increasing amplitude, until one tag eventually reaches fixation. Unstable spiralling is not balancing selection—it is not maintaining multiple tags at equilibrium, but it nevertheless could cause tag fixation time to increase relative to the neutral case (which may be characterised by a more direct evolutionary path to tag fixation, not characterised by spiralling). We therefore need to undertake further analyses to rule out the possibility of directional selection via an extended path, before we can conclude that balancing selection is responsible for maintaining tag diversity[7].

We follow R&R and address this confounder by examining how the tag fixation ratio changes as population size is changed. If selection is causing tag diversity to be lost but via an extended path, then increasing the efficacy of selection relative to drift, for instance by increasing the population size, would cause the population to evolve along the (extended) path more quickly relative to the neutral case, meaning the tag fixation ratio would decrease. Conversely, if genuine balancing selection is in operation, then increasing the efficacy of selection relative to drift, for instance by increasing the population size, would cause the tag fixation ratio to increase.

We find that, in the areas of parameter space where our finite population (agent-based) model found stable tag diversity (Supplementary Fig. 7), the tag fixation ratio increases with population size, demonstrating that genuine balancing selection is in operation. Specifically, we recover key results obtained by R&R that, when migration ($m$) and recombination ($r$) are very small, balancing selection may maintain tag diversity even in the absence of partner search ($\alpha = 0$), and the strength of balancing selection increases with the magnitude of $b$ & $c$ (keeping the ratio $b/c$ constant) (Supplementary Fig. 8). In addition, we find both that increasing the rate of partner search ($\alpha$) and decreasing the cost of partner search ($c_{search}$) increases the strength of balancing selection, in turn increasing the likelihood that tag diversity is maintained (Supplementary Fig. 8).

**Key points and implications.** Overall, we found that kin discrimination based on genetic cues evolves in our island lifecycle if the following conditions are met:

1. Conditional altruism, predicated on the matching of tags with equal population frequencies ($x_i$) and helper proportions ($p_i$), gives a greater inclusive payoff to the actor than indiscriminate strategies would (Eq. 25/1 is satisfied).
2. There is negative frequency dependence at the tag locus, meaning tag frequencies equalise. This is facilitated by:

   a. Prevalent and cheap partner search (high $\alpha$ & low $c_{search}$).
   b. Some trait mutation ($\mu_{Trait}$).

*Condition 1: Inclusive fitness.* Condition 1 relates to whether there is anything to be gained by restricting interactions to genealogical kin. There might not be any reason to do so. For instance, the risk of being cheated by a random deme-mate might be miniscule anyway, removing the evolutionary incentive to restrict social

interactions to genealogical kin. Under the island lifecycle assumptions, condition 1 is more likely to be satisfied if recombination between tag and trait ($r$) is low and migration ($m$) is low.

Many previous studies have pointed out that recombination ($r$) and migration ($m$) hinder the evolution of genetic kin recognition—we accept and recover this result. However, it is generally thought that these parameters exert their effect by introducing evolutionary constraints, which prevent the population from evolving towards genetic kin recognition[7,8,24,56]. It is implicitly assumed that genetic kin recognition would evolve in the absence of such constraints.

We take issue with this interpretation in general, though it is sometimes true. For instance, recombination can sometimes break down linkage disequilibrium between rare tags and altruism, preventing genetic kin recognition from evolving when it otherwise (at lower recombination) would have. However, the capacity of recombination ($r$) to constrain the evolution of genetic kin recognition dissipates when individuals are capable of cheap partner search (high $\alpha$ & low $c_{search}$).

Furthermore, we have shown that recombination ($r$), migration ($m$) and deme size ($N$) affect the relatedness between social partners and local competitors (Supplementary Fig. 4), and therefore affect whether it is advantageous for an individual to restrict social interactions to their tag-mates. As a result, genetic kin recognition is sometimes an adaptive strategy for an inclusive fitness maximising agent, and sometimes it is not, and this will depend on $r$, $m$, $N$. Therefore, $N$, $m$ & $r$ influence the trajectory of evolution by affecting the adaptive value of genetic kin recognition, not (in general) by constraining the range of evolutionary paths that could be taken. You could say that the evolutionary path is *chosen* by freely evolving, fitness-maximising agents; it is not begrudgingly followed by diminished evolutionary agents with little opportunity to have done otherwise.

*Condition 2a: Partner search.* Condition 2a relates to whether or not Crozier's paradox (common-tag advantage) is overcome. This will depend on the extent to which individuals can abandon their social partners, for new social encounters, at not-too-great a fitness cost. Under our lifecycle assumptions, when individuals are incapable of partner search ($\alpha = 0$), and selection is weak (low magnitude of $b$ and $c$), rare tags suffer a starkly reduced social interaction rate, and consequently a prohibitively large direct fitness cost. This means that common tags are selected, eliminating diversity at the tag locus, and preventing the evolution of genetic kin recognition.

Conversely, when individuals are capable of cheap partner search (high $\alpha$, low $c_{search}$), rare tags do not suffer as stark a reduction in social interaction rate. When partner search is uncostly and unrestricted ($\alpha = 1$ & $c_{search} = 0$), rare tags suffer no (if demes are large) or little (if demes are smaller) reduction in social interaction rate (negligible direct cost). The dissipation of the prohibitively large direct cost of tag rarity means that rare tags may be selected, leading to tag equilibration, if the indirect benefit of tag rarity is large enough. Rare tags gain their (indirect) advantage over common tags by being more reliable indicators of kinship, and as a consequence, being less likely to lead to social interactions with cheaters[31].

*Condition 2b: Trait mutation.* Sometimes, partner search alone is not enough to generate an advantage for rare tags and lead to genetic kin recognition. Sometimes, mutation at the trait locus ($\mu_{Trait}$) is also required. Trait mutation takes effect in the following way. Defectors spread amongst individuals bearing common tags, and conditional altruists spread amongst individuals bearing rare tags (kin selection). If the conditional altruism allele spreads (via rare-tag groups) more quickly than it is removed (via common-tag groups), the conditional altruism allele will increase in population frequency alongside rare tags, leading to tag equilibration (genetic kin recognition) and conditional altruism[31].

However, if the conditional altruism allele spreads (via rare-tag groups) more slowly than it is removed (via common-tag groups), the conditional altruism allele will decrease in population frequency. In this case, in the absence of trait mutation, the frequency of the conditional altruism allele will fall to approximately zero (fixation of the defection allele), resulting in defection. This may even occur in regions of parameter space where Eq. 25/1 is satisfied. In other words, even if conditional altruism is favoured when all tags are at equal frequency (Eq. 25/1 satisfied), the quick purging of conditional altruists amongst individuals bearing common tags may mean that tag frequencies do not successfully equilibrate, and defection evolves as a result.

However, if the conditional altruism allele spreads (via rare-tag groups) more slowly than it is removed (via common-tag groups), but there is mutation at the trait locus, the frequency of the conditional altruism allele will not fall all the way to zero. Rather, it will only fall as low as mutation-selection balance (non-zero). This means that, even if conditional altruists are being removed (via common tags) faster than they are being added (by rare tag groups), a baseline proportion of altruists will persist in the population due to the net mutation of defectors into altruists. Furthermore, the altruists that persist will disproportionately bear rare tags (linkage disequilibrium)[31].

In this way, trait mutation ensures that fitness differences between tags persist despite the potentially fast purging of conditional altruists bearing common tags. The persistence of tag fitness differences allows rare tags to increase in frequency alongside the conditional altruism allele. The common tags decrease in frequency, eventually reaching low enough population frequencies that the conditional helping allele is selected amongst individuals bearing these tags. After this point, the conditional helping allele is universally favoured (i.e. selected amongst

individuals bearing all tags), resulting ultimately in tag equilibration and the fixation of the conditional helping allele. Therefore, trait mutation may promote the evolution of genetic kin recognition based on genetic cues.

*Theoretical significance.* As we mentioned earlier, a general result in social evolution theory is that, in 'homogeneous' population structures, where the scale of social interaction is equal to the scale of competition, kin selection (favouring altruism) is offset by kin competition (disfavouring altruism), precluding the evolution of altruism[32,44,50–52]. Altruism only evolves in 'heterogeneous' population structures, where the scale of social interaction is less than the scale of competition. Most theoretical demonstrations of the evolution of altruism have focused on lifecycles where the heterogeneity is there from the start (assumed)—for instance, some have assumed that generations overlap[57]; others have assumed that individuals disperse in groups[58]. By contrast, we have shown that heterogeneous population structure can evolve mechanistically (from scratch), under relatively permissive conditions, leading to the evolution of altruism.

## Additional definitions and derivations

*Definitions of the four outputs plotted in Fig. 2c.* Figure 2c plots, for a single trial, the following four outputs against time: (1) helping frequency; (2) tag diversity; (3) deviation from pedigree relatedness; (4) linkage disequilibrium. Precise mathematical definitions of these four outputs are given below.

Firstly, helping frequency is trivially given by $\sum_{l=1}^{L_{max}} x_l p_l$. This is the population frequency of the helping allele.

Secondly, tag diversity is given by $\frac{\frac{1}{\sum_{l=1}^{L_{max}}(x_l^2)} - 1}{L_{max} - 1}$. To interpret this definition, note that the average tag frequency is given by $\sum_{l=1}^{L_{max}}(x_l^2)$, meaning the number of segregating tags is given by $\frac{1}{\sum_{l=1}^{L_{max}}(x_l^2)}$, meaning the proportion of all available tag diversity that is present at a given time is given by $\frac{\frac{1}{\sum_{l=1}^{L_{max}}(x_l^2)} - 1}{L_{max} - 1}$, which is our tag diversity output.

Thirdly, deviation from pedigree relatedness is given by $\sum_{l=1}^{L_{max}} \left( x_l \left| R_{tag} \right|_{p_i = \bar{p}} - R_{tag} \right| \right)$. This is the average absolute difference between: the actual (locus-specific) relatedness between actors and social partners ($R_{tag}$; Eq. 18), and the pedigree relatedness between actors and social partners ($R_{tag} |_{p_i = \bar{p}}$; Eq. 19).

Fourthly, linkage disequilibrium is given by $-D$, which can be written explicitly as $-\sum_{l=1}^{L_{max}} \left( x_l \left( x_l - \sum_{l=1}^{L_{max}} x_l^2 \right) (p_l - \bar{p}) \right)$. This is derived in the following section.

*Derivation of linkage disequilibrium.* We derive an expression for the association (linkage disequilibrium) between tag frequency and helping. Our derivation follows the standard methodology for calculating linkage disequilibria in multi-locus population genetic models, as outlined in Kirkpatrick et al.[36].

We characterise trait deviation as follows. Let an individual have a trait value of 0 if it has the defection allele, and a trait value of 1 if it has the helping allele. The population average trait value is then given by $\bar{p}$, where $\bar{p}$ is the population average frequency of the helping allele $\left( \bar{p} = \sum_{l=1}^{L_{max}} x_{l1} \right)$. A given individual will therefore deviate in its trait value from the population average trait value. For individuals with the helping allele, this deviation will be $1 - \bar{p}$, and for individuals with the defection allele, this deviation will be $-\bar{p}$. The average trait deviation amongst individuals bearing a given tag $i$ will then be given by:

$$p_i - \bar{p}, \tag{29}$$

where $p_i$ is the frequency of the helping allele amongst individuals who have the tag $i$ $\left( p_i = x_{i1}/(x_{i0} + x_{i1}) \right)$. The average trait deviation amongst the whole population is trivially given by zero $\left( \sum_{l=1}^{L_{max}} \left( x_l (p_l - \bar{p}) \right) = 0 \right)$.

We characterise tag frequency deviation as follows. Let each individual have a tag frequency value that is (trivially) given by the population frequency of its tag $\left( x_i = x_{i0} + x_{i1} \right)$. A given individual may therefore deviate in its tag frequency value from the population average tag frequency value. For individuals with the tag $i$, this deviation will be:

$$x_i - \sum_{l=1}^{L_{max}} x_l^2, \tag{30}$$

where the $\sum_{l=1}^{L_{max}} x_l^2$ term gives the population average tag frequency (i.e. this will be one if there is only one tag at fixation in the population). The average tag frequency deviation amongst the whole population is trivially given by zero $\left( \sum_{l=1}^{L_{max}} \left( x_l \left( x_l - \sum_{l=1}^{L_{max}} x_l^2 \right) \right) = 0 \right)$.

For a given individual with a tag $i$, the product of deviations over the tag and trait loci is obtained by multiplying Eqs. 29 and 30 to give:

$$\left( x_i - \sum_{l=1}^{L_{max}} x_l^2 \right) (p_i - \bar{p}). \tag{31}$$

If this multiplicatively combined joint deviation is positive, it means either that: (i) the individual has a more-common-than-average neutral tag and a greater-than-average expectation (based on its tag identity) of being a helper; (ii) the individual has a less-common-than-average tag and a lower-than-average expectation (based on its tag identity) of being a helper. Conversely, if this multiplicatively combined joint deviation is negative, it means either that: (i) the individual has a more-common-than-average tag and a lower-than-average expectation (based on its tag identity) of being a helper; (ii) the individual has a less-common-than-average tag and a greater-than-average expectation (based on its tag identity) of being a helper.

The expected product of deviations over the tag and trait loci, measured across all individuals in the population, is then obtained by averaging Eq. 31 over all tags ($\{1, 2,...,L_{max}\}$), to give:

$$D = \sum_{l=1}^{L_{max}} \left( x_l \left( x_l - \sum_{l=1}^{L_{max}} x_l^2 \right) (p_l - \bar{p}) \right). \tag{32}$$

Equation 32 gives the association (linkage disequilibrium) between tag frequency and helping. It is a population-wide statistic, insofar that it refers to a characteristic of the population as a whole, rather than to a specific individual in the population. Technically, it is equal to the covariance in the allelic state of genes at the tag and trait loci. A positive value of $D$ indicates that common tags are associated with (technically: "covary with") the helping allele. Conversely, a negative value of $D$ indicates that rare tags are associated with (technically: "covary with") the helping allele. In Fig. 2c, we plot $-D$ rather than $D$, as the former represents the association between *rare* tags and helping.

*Definition of tag excess helping plotted in Fig. 3b.* The dotted lines in Fig. 3b show, for a given tag, the probability above the population average that a social interaction results in help being received (tag excess helping). Explicitly, tag excess helping is given by $x_i(p_i - \bar{p})$; where $p_i$ and $\bar{p}$ denote helping frequency amongst tag $i$ individuals, and all individuals, respectively, and where $x_i$ denotes the population frequency of tag $i$. The tag frequency weighting ($x_i$) is applied so that the curves are discernible when plotted.

**Reporting summary**. Further information on research design is available in the Nature Research Reporting Summary linked to this article.

## Data availability
The data generated during the current study are available at https://doi.org/10.5281/zenodo.6624184[55].

## Code availability
The Matlab code used to perform simulations and numerical calculations is available at https://doi.org/10.5281/zenodo.6624184[55]. We include programs for implementing our weak-selection mathematical model, and our agent-based simulation model. We also include code for generating figures.

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

## Acknowledgements

We thank Guy Cooper, Asher Leeks, Sam Levin and Geoff Wild for discussion and comments on the manuscript, NERC (grant NE/V011537/1; T.W.S., A.G., S.A.W.) and ERC (grant 834164; T.W.S., S.W.) for funding.

## Author contributions

Conceptualization: T.W.S., A.G. and S.A.W.; formal analysis: T.W.S.; writing—original draft: T.W.S.; writing—review & editing: T.W.S., A.G. and S.A.W.

## Competing interests

The authors declare no competing interests.
