## [Peer Review File · Nature Communications]

Multiple social encounters can eliminate Crozier's paradox and stabilise genetic kin recognitionReviewers' Comments:

Reviewer #1:

Remarks to the Author:

The paper presents a model of genetic kin recognition and cooperation based on highly variable, inherited "tags" that allow individuals to pair up with same-tag individuals, who are more likely than average to be close relatives. Inclusive fitness theory predicts that costly altruism should evolve more easily when it is directed at relatives, and earlier work has examined the coevolution between these tags and alleles that encode altruistic and non-altruistic behaviours. In 1986, Ross Crozier noted that individuals with rare recognition tags should receive less help than common-tag individuals, and might have a more difficult time locating social partners, giving them low fitness - this causes common tags to become more common, until the recognition cues become uniform and thus useless as cues of relatedness, creating what is now called Crozier's paradox.

This model extends that earlier work by changing how cooperative interactions are modelled before, and finds that Crozier's paradox no longer holds - this is the paper's main conclusion. Specifically, the model assumes that individuals can form a pair with one other individual, and potentially help each other, and that pair formation depends upon genetic tags. There are two types of individual (conditional helpers, and non-helpers), controlled by a genetic polymorphism, and there is a second locus controlling the recognition tag. Individuals form their pairs based on the tag, in an iterative process. If the first individual they meet has the same tag, a pair is formed. If not, they potentially continue to inspect other individuals, which happens with probability α , until they meet a same-tag individual. If the focal individual has the helping allele and its partner has the same tag, it helps the other individual at a cost to itself, otherwise no helping takes place. When α tends to 1, rare-tagged individuals always manage to find a partner, no matter how rare their tag is, and under this high- α scenario, Crozier's paradox does not hold. This is because when rare-tagged individuals always manage to pair up (and there is no cost to searching them out), there is a net benefit to carrying a rare tag, because these tags are better indicators of relatedness and hence of altruism strategy.

One odd aspect is that the paper has supplementary material running to 75 pages, which contains many important details and results that are not discussed or even alluded to in the main paper. I read the main paper first, and had several questions that could be answered by reading the supplement. For example, I was confused why the authors chose to model a scenario where searching through 100s of individuals to find one with a matching, rare tag was assumed to have no fitness costs, relative to being able to easily find a common-tagged partner, because this seems to completely undermine the model. In the supplement, I see that there was indeed a parameter called C_{search} , which controls the costliness of this searching process, and indeed setting $C_{\text{search}} > 0$ makes it less likely that the model resolves Crozier's paradox. However, C_{search} is never mentioned and is barely alluded to in the main manuscript, and I assume that all the figures in the main manuscript were generated assuming that $C_{\text{search}} = 0$. I am puzzled why the authors do not discuss this work in the main paper, especially because C_{search} is a biologically reasonable addition to the model, and its value completely determines the conclusions. Instead, the paper simply notes the the results are "robust" to the assumption that there are costs of partner searching (citing the supplement), although this claim seems to be contradicted by Figure S6 in the supplement (which illustrates that $C_{\text{search}} > 0$ reduces the parameter space where genetic kin recognition and altruism can evolve).

Overall I think the paper would be greatly improved by properly describing all the work in the supplement in the main paper. I wonder what it's there for if it doesn't warrant a mention in the paper (e.g. the simple vs island model distinctions).

Specific comments:

Line 57-59: "Previous theory has assumed..." This isn't true of the Holman et al. simulation model, which assumed that individuals interact with all N individuals within their own patch and potentially help (or receive help) from each of them depending on their recognition cues (sort of like $\alpha = 1$ in your model, but with partnerships forming with all matching-tag individuals in the patch, instead of just one as in your model). So, it seems untrue that this is a 'first' for the present model. However your model is an advance over that one in the sense that you consider the possibility that individuals get to inspect with between 1 and all N members of their patch in their search for a single social partner, instead of all or nothing.

Line 63: "If multiple encounters occur, then individuals with rare tags could find individuals with the same tag and receive as much help as individuals with common tags" But surely, the individuals with common tags would more easily find same-tag individuals, and find more of them, and thus probably still have higher fitness than rare-tag individuals under many realistic assumptions? Your model assumes that there is no cost to searching for partner with the same tag, and so you assume that individuals finding a partner on their first try have the same fitness as those that have to check all N individuals to find one. [update: you relax this assumption in the supplement, but never mention it in the main text - as I expected, adding search costs can restore Crozier's paradox if they are high enough]

Line 75: Is the group size N finite or infinite? If it's finite, what happens to individuals who are the only one with their tag within their patch - do they never associate with a partner? If this never happens because groups are so large, you may have removed one of the potential fitness costs of having a rare tag (i.e. higher risk of being unable to engage in social interactions).

Line 89: Again this doesn't apply to the Holman et al model, which assumes that individuals check all others in their patch and socially interact with all of them that share the tag.

Line 101: don't -> do not

Line 129: isn't -> is not

Line 147: "greater payoff" -> "greater average payoff"

Line 198: "linkage disequilibrium favouring rare tags". It's not strictly the LD [with the cooperation locus] that provides a fitness advantage to the rare tags. Instead, individuals that have a rare tag are more likely to receive help from their social partner than individuals with common tags are (i.e. there is LD at the population level between the tag and helping loci), and this extra help to rare-tagged individuals increases their fitness. I know it's a semantic point but you might be able to word this bit more clearly.

Line 209: "The probability of social interactions needs to drop significantly below 1.0 before genetic kin discrimination is likely to be less favoured" This is completely dependent on the specific parameters used here (e.g. the values of b and c and L_max), and also on the biology assumed in your model. For example, I think this run of the model assumes that an individual that has to check 100 partners before finding a matching tag one has the same fitness as individual who gets it right the first time. Relaxing that assumption would presumably change Figure 4d and other results. At present, the results could be read as "If we assume some of the main effects previously proposed to lead to Crozier's paradox are absent, such as greater search costs and more failures to find a partner for rare-tag individuals, then the paradox is resolved"; this doesn't invalidate your paper in my view, but it changes it from "we've solved the paradox" to "we've identified which biological assumptions lead to the paradox more clearly than before".

Line 226: "Previous studies found that tag diversity could not be maintained by selection alone". This

is not correct. I think you mean something like “could not be maintained by selection to avoid associating with non-cooperators”, or similar. For example, the Holman et al. model considered a second source of diversifying selection acting on the tag locus (from sexual selection) and found (unsurprisingly) that it could counteract the Crozier effect if that second source of selection was strong enough - but it’s still selection. What else maintains genetic diversity, if not selection? (mutation I guess, but that’s not what you mean here).

Line 229: “worst case scenario” Conversely, you assume there are no costs to rare-tag individuals that have to check dozens of partners before finding a match, and have little or no risk of going unpaired because patches are large or infinite (I’m not sure which it is), which is a best-case scenario for them.

Line 238: “Second, our theory emphasises the need to measure the frequency with which individuals encounter other individuals with the same tag (Fig. 4d). When this frequency is high, corresponding to a high alpha...” This seems incorrect. In the methods, you describe alpha differently, as the probability an individual goes off to keep searching when it encounters a mis-matched partner. Also, your theory does not mean that researchers should go out and measure how often tag-matched individuals encounter one another, I think. What matters is the relative fitness effects of cooperation for individuals with common vs rare tags - if the rare-tagged individuals have higher socially-based inclusive fitness (due to avoiding pairing up with unhelpful cheaters, directing their help to other cooperators, etc) then it would demonstrate that the Crozier effect (from being unable to easily find and be found by same-tag individuals) is outweighed by the LD effect you mention.

Line 241: “Third, Equation 1...” But that equation is basically just Hamilton’s rule, and so it’s not a prediction that stems from your theory. I suggest removing this.

Line 245: Could also note that genetic kin recognition is more favoured when there is little cost to checking large numbers of individuals to find a rare match, and when it’s actually possible to check many individuals. However as noted above, this is similar to saying that Crozier’s paradox doesn’t hold if the biological assumptions that led to its formulation are not true.

Reviewer #2:

Remarks to the Author:

This is a fantastic paper. It presents new theory on selection for tag-based altruism and a resolution for Crozier’s paradox, a longstanding dilemma in social evolution theory. Crozier’s paradox states that though genetic cues or tags are used to identify relatives, using them in this way is unstable, because common tags have higher fitness than rare ones. This comes about because common cues will match more often and get more kin altruism. Thus, theory tends to say that tag-based kin recognition should be absent or uncommon, but we have many empirical examples. The authors’ theoretical innovation is very simple in principle; allow mismatched individuals (mostly rare tags) to break off and search further for a matching partner. This turns out to stabilize genetic kin recognition, thus solving an important problem. The paper is also very clearly written.

The mathematical theory is of course more daunting and most of it is relegated to a very long supplement. My initial reaction as a reviewer was that a 75-page supplement was going to be too much. But on reading it, I changed my mind. It is a superb exposition of the models, extremely clear in its goals, assumptions, methods, and conclusions. While not every reader will want to dig through the detailed models, this way of writing it up maximizes the audience that could understand it if they put in the effort. I only wish that all theoretical papers were written like this.

My only significant comment is that I’d like to see a little more clearly how different values of alpha affect outcomes. In Fig. 4a, having a very high alpha of 0.975 already means a significant portion of the area where kin disclination is favored does not support stable recognition. So what about lower

values? Figure 4d gives us an idea about this, but for probability of interaction rather than alpha. Going back to 4c gives us the relationship between those two variables, and together c and d seems to confirm that for tag frequency near zero, you need very high alphas. So, does that mean, for lower alphas, that low tag frequencies (high tag numbers) won't be stable, but higher frequencies (lower tag numbers) will be? A little more discussion of this point would be helpful particularly since you consider only $\alpha=1$ in the detailed model. Alpha is the parameter that makes your model work where other models did not, and yet it seems to be the most poorly explored parameter.

I don't really like "networks" in the title. Networks are not the key to the solution of the problem and the word is hardly mentioned in the text. It will also lead readers into expecting a kind of model that this is not, with emphasis on nodes and edges etc. The real key to this model is partner searching or partner choice (though based on tags rather than past behavior). Shouldn't that be highlighted in the title.?

line 75. Later on I found that I had forgotten the important distinction here between encounters and interactions. It might help to insert here something along the lines of the following piece from the supplement "individuals can engage in sufficiently many social encounters before committing to a given social interaction".

105. What does the assumption of minimal diversity mean? Truly minimal would be no diversity.

SUPPLEMENT

p 1 bottom. "Partner recycling" seems a confusing term for this process. It implies a partner is going to be used again, which is not true here, but is true in standard partner-choice models where a partner is kept if it shows the right behavior. Consider alternative like "partner search", "tag search" or "tag choice".

p 5 par 3. I have trouble figuring out how search costs work. Are you making assumptions here for mathematical convenience or to match biology?

p 6 top. "density dependence is bland" seems on odd description

p. 7 par 1. The two probabilities making up the cost are given in one order verbally and in the opposite order in the math which is unnecessarily confusing.

eqns 1 and 2. I'm having trouble seeing that how the denominator of the fractions work. Is there a series argument here?

Added later. I get this now after reading your later explanation of equation 25. So maybe some of that explanation needs to be moved up to here. But I would also more explicitly add, because this is what I was failing to see, that this same probability applies to every step of the search process – even as the numbers of new interactions decline, the probability stays the same.

eqns 1 and 2. Doesn't your A term imply that the fitness components like b and c effectively vary over the generations as the number of altruistic interactions changes? Normally b and c are defined in terms of relative fitness and they go into the calculation of mean fitness, so mean fitness changes while b and c remain constant. Here it seems the reverse, mean fitness stays constant, implying that b and c change? I guess it boils down to I'm thinking about hard selection (of the global population) and you are modeling soft selection. It might be worth pointing out.

p 13, an aside regarding Levin & Grafen. You cite reference 42 for both the old and the newer way of calculating inclusive fitness. Should it be two different references?

Fig. 5 It's not clear to me what the blue text about alpha and csearch is saying about the figure.

eqn 17 and below. So, your number of tags is not the countable number of tags but a sort of effective tag number based on frequencies?

p 33 par 6 add a reference to Fig S3 when you mention being stretched on the y-axis?

Figure 6. Nice figure, but you do not actually refer to it very much in the text. Do you want to point out any more features of these results more explicitly?

p 41 bottom. Fix "even when indiscriminate cooperation is favored over indiscriminate defection"

p 42 par 1 "in our simple model, density-dependent population regulation was global as opposed to local (hard selection)". It's not clear if the "hard selection" is referring to global or local. But the description does not cord with my understanding of hard and soft selection. Hard and soft is not fundamentally about local vs global but an apply to either level. At the global level, I think your model is soft selection, as mean absolute fitness is fixed at 1 and is independent of allele frequency. Your social groups (interacting pairs) are under local hard selection because their contribution to the global pool is proportional to their mean fitness. This comment doesn't affect the results of course.

p 45 par 4. Are you really not going to show results for $c_{search} \neq 0$? See my comment on alpha at the beginning. These are the two parameters that matter most for you model.

p 45 last par. You define M_{helped} , but I think you use M_{ind} helped in the text that follows.

p 57 par 1 I note that Fig. 11 is referenced here, before Fig 10 is first referenced.

p 64 par 2. "and this will require a low cost of kin recognition (high α)" – is "high α " what you mean here?

p 64 par 4. "If the evolutionary constraint on tag diversity is relaxed completely ($L_{max} \rightarrow \infty$), then genetic kin recognition may evolve (Equation 43 satisfied) across most of the parameter space (i.e. across most values of m , N and r)." Do you mean for most values of m , N , and r , for which inclusive fitness predicts cooperation?

p 65 par 2. Is this agent-based model identical to your island model, other than being agent-based?

p 65 par 5-6 I'm not seeing the connection between these two paragraphs. the first says that conditional altruism will fall to close to zero (but held above by mutation). This will maintain rare tags. The second paragraph has those rare tags and conditional altruism being favored, the latter to fixation. What specifically has changed to cause conditional altruism to switch from being selected against to being selected to fixation?

Reviewer #3:

Remarks to the Author:

The manuscript is very interesting question, though I have a few comments.

1. Some researchers argue that there are no examples of genetic kin recognition, including one of the authors (Grafen 1990 argued that the only evidence comes from one species of marine invertebrates, *Botryllus schlosseri*), and yet this manuscript ignores this issue.

If there are good examples in the literature, then these should be addressed, but if genetic kin recognition does not exist or there are few or no good examples, then the point of such a model is

unclear – and its importance is dubious.

2. The introduction (Crozier's Paradox) summarizes the theoretical models that have addressed this topic, but it does not accurately summarize the field or previous theoretical work.

Contrary to what the authors assert, it is not widely accepted that kin recognition via genetic cues is evolutionarily unstable (p.2, line 29), as papers cited and the authors themselves have helped to keep this hypothesis viable. So, this assertion is inaccurate, which can be easily shown by any review on this topic.

The authors state that previous models find that genetic kin recognition is only stable under 'very restrictive conditions' (p. 2, line 39; line 227). The model by Rousset and Roze (2007), which is cited, found that kin recognition can be maintained given certain assumptions, i.e., spatial population structure (from low dispersal) and recombination between matching and helping loci, and negative-frequency dependent selection from parasites, but these are realistic, not unrealistically restrictive. It is unclear which aspects of this and other models are unrealistic or overly restrictive.

Grafen (1990) proposed that the main problem with Crozier's Paradox is that it assumes that there are no cheats, but this issue is buried in the supplement.

3. The assumptions of the authors' model allow multiple encounters, which is expected with realistic spatial populations structure, which is in line with previous models.

4. The authors assert that their conclusions are robust to several It is crucial to show that tag diversity is not eroded by genetic drift (p 7, line 151), and this issue is too important for the supplements and should be addressed in the manuscript. Also, the paper should address how robust their results are against these assumptions.

Many other important issues are found only in the supplement.

5. The authors state that their model makes only conservative assumptions (p 13, line 230), but they do not explain why -- or why they are more conservative than previous models.

6. The main question here is: 'can genetic recognition evolve with fewer assumptions of previous models?' The authors suggest that host-parasite interactions, the solution usually cited to solve Crozier's Paradox, 'may be a red herring' (p 13, line 237). If so, then the authors should clarify why their model assumptions are not only simpler but also more realistic, and acknowledge that their model is not a mutually exclusive alternative.

7. There are many terms that are not defined in the manuscript, such as 'kin recognition', 'pedigree relatedness', etc, and it is critical to define terms because they are used differently by different researchers. e.g. Grafen (1990) was criticized for generating semantic confusion over the term 'kin recognition'. The authors contrast genetic kin recognition with environmentally determined phenotypic tags, but then use an example of 'grew up in the same nest' (p. 2, line 41), which is not a phenotypic label. There are reviews that have addressed these semantic issues, but they are not cited.

8. Fig 1 is unnecessary. The point is rather simple and does not need a figure (and this one does not really help).

9. The final paragraph is very good.

Reviewer #1 (Remarks to the Author):

The paper presents a model of genetic kin recognition and cooperation based on highly variable, inherited “tags” that allow individuals to pair up with same-tag individuals, who are more likely than average to be close relatives. Inclusive fitness theory predicts that costly altruism should evolve more easily when it is directed at relatives, and earlier work has examined the coevolution between these tags and alleles that encode altruistic and non-altruistic behaviours. In 1986, Ross Crozier noted that individuals with rare recognition tags should receive less help than common-tag individuals, and might have a more difficult time locating social partners, giving them low fitness - this causes common tags to become more common, until the recognition cues become uniform and thus useless as cues of relatedness, creating what is now called Crozier’s paradox.

This model extends that earlier work by changing how cooperative interactions are modelled before, and finds that Crozier’s paradox no longer holds - this is the paper’s main conclusion. Specifically, the model assumes that individuals can form a pair with one other individual, and potentially help each other, and that pair formation depends upon genetic tags. There are two types of individual (conditional helpers, and non-helpers), controlled by a genetic polymorphism, and there is a second locus controlling the recognition tag. Individuals form their pairs based on the tag, in an iterative process. If the first individual they meet has the same tag, a pair is formed. If not, they potentially continue to inspect other individuals, which happens with probability α , until they meet a same-tag individual. If the focal individual has the helping allele and its partner has the same tag, it helps the other individual at a cost to itself, otherwise no helping takes place. When α tends to 1, rare-tagged individuals always manage to find a partner, no matter how rare their tag is, and under this high- α scenario, Crozier’s paradox does not hold. This is because when rare-tagged individuals always manage to pair up (and there is no cost to searching them out), there is a net benefit to carrying a rare tag, because these tags are better indicators of relatedness and hence of altruism strategy.

One odd aspect is that the paper has supplementary material running to 75 pages, which contains many important details and results that are not discussed or even alluded to in the main paper. I read the main paper first, and had several questions that could be answered by reading the supplement. For example, I was confused why the authors chose to model a scenario where searching through 100s of individuals to find one with a matching, rare tag was assumed to have no fitness costs, relative to being able to easily find a common-tagged partner, because this seems to completely undermine the model. In the supplement, I see that there was indeed a parameter called C_{search} , which controls the costliness of this searching process, and indeed setting $C_{\text{search}} > 0$ makes it less likely that the model resolves Crozier’s paradox. However, C_{search} is never mentioned and is barely alluded to in the main manuscript, and I assume that all the figures in the main manuscript were generated assuming that $C_{\text{search}} = 0$. I am puzzled why the authors do not discuss this work in the main paper, especially because C_{search} is a biologically reasonable addition to the model, and its value completely determines the conclusions. Instead, the paper simply notes the the results are “robust” to the assumption that there are costs of partner searching (citing the supplement), although this claim seems to be contradicted by Figure S6 in the supplement (which illustrates that $C_{\text{search}} > 0$ reduces the parameter space where genetic kin recognition and altruism can evolve).

We fully take on board this concern, and have added an extensive analysis of partner search cost (c_{search}) to the island model (i.e. the model presented in the main text). The mathematical details are presented in the SI (pages 13-15, 31 & interspersed text), and the results have been added to the main text (under the “Encounter rate and search cost” heading on p11, & Fig. 5).

In particular, we explain what we meant by the claim that the results are robust to an increase in c_{search} (though we no longer use the overly-strong word “robust”). We show that genetic kin recognition is less likely to be destabilised by an increase in the search cost (c_{search}) than by a decrease in the partner search rate (α). This is because the benefit of tag rarity – more cooperative social interactions – shines through more strongly when rare-tag individuals have more social opportunities. Increasing c_{search} does not limit these social opportunities, whereas decreasing α does, which is why c_{search} is less likely to destabilise genetic kin recognition.

Overall I think the paper would be greatly improved by properly describing all the work in the supplement in the main paper. I wonder what it’s there for if it doesn’t warrant a mention in the paper (e.g. the simple vs island model distinctions).

We have taken this on board, removing all non-essential information (e.g. the “simple model” analysis) from the SI, and making sure that all key results in the SI are also described in the main text (e.g. examination of c_{search} ; a more detailed analysis of the encounter parameter, α ; additional examination of the strength of balancing selection in finite populations).

Specific comments:

Line 57-59: “Previous theory has assumed...” This isn’t true of the Holman et al. simulation model, which assumed that individuals interact with all N individuals within their own patch and potentially help (or receive help) from each of them depending on their recognition cues (sort of like $\alpha = 1$ in your model, but with partnerships forming with all matching-tag individuals in the patch, instead of just one as in your model). So, it seems untrue that this is a ‘first’ for the present model. However your model is an advance over that one in the sense that you consider the possibility that individuals get to inspect with between 1 and all N members of their patch in their search for a single social partner, instead of all or nothing.

We see that our sentence was badly worded. We have changed it to read: “Previous theory has assumed that, when an individual encounters a partner with a different tag, the opportunity to socially interact is wasted”. It is the ability to abandon tag-mismatched partners in favour of new social encounters (partner search) that is the ‘first’ for our model – we hope this is clearer now.

Line 63: “If multiple encounters occur, then individuals with rare tags could find individuals with the same tag and receive as much help as individuals with common tags” But surely, the individuals with common tags would more easily find same-tag individuals, and find more of them, and thus probably still have higher fitness than rare-tag individuals under many realistic assumptions? Your model assumes that there is no cost to searching for partner with the same tag, and so you assume that individuals finding a partner on their first try have the same fitness as those that have to check all N individuals to find one. [update: you relax this assumption in the supplement, but never mention it in the main text - as I expected, adding

search costs can restore Crozier's paradox if they are high enough]

See above reply regarding our new analyses of partner search cost (c_{search}).

Line 75: Is the group size N finite or infinite? If it's finite, what happens to individuals who are the only one with their tag within their patch - do they never associate with a partner? If this never happens because groups are so large, you may have removed one of the potential fitness costs of having a rare tag (i.e. higher risk of being unable to engage in social interactions).

We have two points in reply to this:

- 1) Regarding whether group size is infinite or finite, we constructed our mathematical model under the assumption that there is no stochastic variation in the genetic composition of demes. This means that our mathematical model is only accurate for the case where group size (N) is infinite (this eliminates stochastic variation), and / or when partner search (α) is zero (this renders stochastic variation unimportant for selection). In the case where $\alpha > 0$, the model becomes less accurate as deme size (N) is reduced. So the key point here is that, technically, deme size is assumed to be finite (this is necessary because we vary deme size when presenting certain results), but we also recognize that there is a mathematical inaccuracy here. This mathematical inaccuracy is inevitable in order to construct an analytically tractable model. Importantly, we check via agent-based simulation that our results tend to hold when stochastic deme variation is accounted for, even when deme size (N) is low. This more nuanced description of the assumptions behind deme size (N) and deme genetic composition have been added to the SI (e.g. pages 4, 5, 31) and main text (lines 73-75, 155-157).
- 2) Your point about rare-tag individual sometimes entering demes without tag-mates, and therefore being unable to find a tag-matched partner, irrespective of partner search, is a valid one – we completely agree. We have added a detailed account of this concern to the SI (page 31). Fortunately, this extra cost for rare tags does not tend to alter results, even for small deme size (N), as shown in our agent-based simulation analyses, which account for this (SI pages 31-34).

Line 89: Again this doesn't apply to the Holman et al model, which assumes that individuals check all others in their patch and socially interact with all of them that share the tag.

We see that we have introduced some semantic confusion here. It is true that, in Holman et al., "individuals check all others in their patch and socially interact with all of them that share the tag". Indeed, this is a standard assumption in models of tag-based helping, including Rousset and Roze (2007). However, these models are all still considering the $\alpha=0$ case, because social interaction rate is fully determined by the frequency of the individual's tag. For instance, in these models, if an individual has a rare tag, such that only one other member of its group has the same tag, that individual will socially interact once that generation. Conversely, an individual with a more common tag, such that e.g. 10 other individuals in its group have the same tag, will socially interact 10 times that generation.

This contrasts with the $\alpha=1$ case, in which social interaction rate is independent of tag frequency. The $\alpha=1$ case has not been permitted in any theoretical treatment prior to ours, as far as we are aware.

We recognise that this issue may have arisen because we said in places that our study, but not previous ones, allowed multiple social encounters. What we really mean is – our study, but not previous ones, allowed multiple social encounters *before each social interaction* (partner search). We have made sure to add in this “before each social interaction” qualification throughout the text.

Line 101: don't -> do not

Amended.

Line 129: isn't -> is not

Amended.

Line 147: “greater payoff” -> “greater average payoff”

Amended.

Line 198: “linkage disequilibrium favouring rare tags”. It's not strictly the LD [with the cooperation locus] that provides a fitness advantage to the rare tags. Instead, individuals that have a rare tag are more likely to receive help from their social partner than individuals with common tags are (i.e. there is LD at the population level between the tag and helping loci), and this extra help to rare-tagged individuals increases their fitness. I know it's a semantic point but you might be able to word this bit more clearly.

We have changed the wording to “more cooperative social interactions (due to linkage disequilibrium) favouring rare tags.”

Line 209: "The probability of social interactions needs to drop significantly below 1.0 before genetic kin discrimination is likely to be less favoured" This is completely dependent on the specific parameters used here (e.g. the values of b and c and L_max), and also on the biology assumed in your model. For example, I think this run of the model assumes that an individual that has to check 100 partners before finding a matching tag one has the same fitness as individual who gets it right the first time. Relaxing that assumption would presumably change Figure 4d and other results. At present, the results could be read as “If we assume some of the main effects previously proposed to lead to Crozier's paradox are absent, such as greater search costs and more failures to find a partner for rare-tag individuals, then the paradox is resolved”; this doesn't invalidate your paper in my view, but it changes it from “we've solved the paradox” to “we've identified which biological assumptions lead to the paradox more clearly than before”.

We have changed the wording to “the probability of social interactions often needs to drop significantly below 1.0” – i.e. we have added in the word “often”, to indicate that these results hold for most biologically reasonable parameterisations of the model.

Line 226: “Previous studies found that tag diversity could not be maintained by selection alone”. This is not correct. I think you mean something like “could not be maintained by selection to avoid associating with non-cooperators”, or similar. For example, the Holman et al. model considered a second source of diversifying selection acting on the tag locus (from sexual selection) and found (unsurprisingly) that it could counteract the Crozier effect if that

second source of selection was strong enough - but it's still selection. What else maintains genetic diversity, if not selection? (mutation I guess, but that's not what you mean here).

We have changed the wording to “Previous studies found that, in general, tag diversity could not be maintained by selection on social behaviour alone” – i.e. we have added in “in general” and “on social behaviour”. This statement, to the best of our knowledge, is accurate. We have also added more explanation of this statement to the SI (pages 38-39), which is referenced in the main text (lines 280-286).

In particular, Rousset & Roze (2007) found that 2 tags can be maintained, under very low recombination and mutation (restrictive conditions), by selection on social behaviour alone (i.e. without tag mutation), but no more than 2 tags can be maintained without tag mutation.

Furthermore, Holman et al. found that tag diversity can be maintained by extrinsic balancing selection, if tags have an additional role in mate choice. This is similar to Crozier’s suggestion that tag diversity can be maintained if tags have an additional role in host-parasite interactions. We have added a detailed discussion of these theories to the SI (pages 38-39) and main text (lines 292-312), but these invoke selective pressures extrinsic to the evolution of social behaviour.

Line 229: “worst case scenario” Conversely, you assume there are no costs to rare-tag individuals that have to check dozens of partners before finding a match, and have little or no risk of going unpaired because patches are large or infinite (I’m not sure which it is), which is a best-case scenario for them.

We have changed the sentence to “Additionally, our theory modelled a relatively unfavourable scenario for genetic kin recognition, and so our finding that it can be stable may be conservative.” We see that “worst case scenario” was overly strong language. However, the issues you raise – regarding partner search cost, and infinite deme size – are both accounted for, either in our mathematical model, or in our agent-based simulation.

Line 238: “Second, our theory emphasises the need to measure the frequency with which individuals encounter other individuals with the same tag (Fig. 4d). When this frequency is high, corresponding to a high α ...” This seems incorrect. In the methods, you describe α differently, as the probability an individual goes off to keep searching when it encounters a mis-matched partner. Also, your theory does not mean that researchers should go out and measure how often tag-matched individuals encounter one another, I think. What matters is the relative fitness effects of cooperation for individuals with common vs rare tags - if the rare-tagged individuals have higher socially-based inclusive fitness (due to avoiding pairing up with unhelpful cheaters, directing their help to other cooperators, etc) then it would demonstrate that the Crozier effect (from being unable to easily find and be found by same-tag individuals) is outweighed by the LD effect you mention.

We have changed this passage to “Second, our theory emphasises the need to measure the frequency with which individuals encounter other individuals to allow them a reasonable chance of encountering one with the same tag (Fig. 4d). When this frequency is high, corresponding to a high α ...” (lines 318-321). We see that our previous wording was not quite right – it is the rate with which individuals encounter each other (our α parameter) that should be measured in general (or a proxy for it), rather than the rate with which individuals *with the same tag* encounter each other.

With regards to your suggestion, one issue with it is that it seems to require knowing the population frequency of recognition tags (though you may have ideas for getting around this). This is likely to be empirically difficult, particularly if the genetic basis of genetic kin recognition is unknown. We agree though that examining the inclusive fitness consequences of conditional versus unconditional (or less conditional) behaviour is important, and indeed this is a key result of our analysis, encapsulated by our Hamilton's Rule condition (Equation 1). However, this is not the whole story, because genetic kin recognition will often not be stable if partner search is limited (low α) or costly (high c_{search}), even if conditional altruism is a favourable strategy. This is why measuring α in natural populations – or a proxy for it, such as time spent searching for a social partner or aggregating into a multicellular group, etc – is also important.

Line 241: "Third, Equation 1..." But that equation is basically just Hamilton's rule, and so it's not a prediction that stems from your theory. I suggest removing this.

The reviewer is correct to note that Equation 1 is very similar in form to the versions of Hamilton's rule derived by Taylor & Queller amongst others. In these formulations, we have $Rb - c - R'(b-c) > 0$, where R is the relatedness between social partners, and R' is the relatedness between competitors. Our version is similar to this. However, the novelty in our formulation is that R & R' are derived as functions of tag frequency (with R being more strongly affected by a change in tag frequency than R'). Our derivation of this modified Hamilton's Rule gives a necessary condition for kin discrimination based on genetic cues to evolve. This link between Hamilton's rule and genetic kin recognition is important and, we think, worth pointing out. It is explicitly derived from our population genetic model in the SI (Section 3b), and is a novel prediction, so we disagree with the statement that "it's not a prediction that stems from your theory". We have added "(derived in Supp. 3b)" to the main text when introducing Equation 1, to emphasise that it is a prediction that stems from our theory.

Line 245: Could also note that genetic kin recognition is more favoured when there is little cost to checking large numbers of individuals to find a rare match, and when it's actually possible to check many individuals. However as noted above, this is similar to saying that Crozier's paradox doesn't hold if the biological assumptions that led to its formulation are not true.

We have changed the sentence to: "Genetic cues could be more likely to be favoured when there is greater opportunity for multiple low-cost social encounters (higher α & lower c_{search}), for instance, when social groups are more compact (dense social networks)." i.e. we have added in "low-cost" and "& lower c_{search} ".

With regards to whether this is similar to saying that Crozier's paradox doesn't hold if the biological assumptions that led to its formulation are not true – Crozier's argument did not mention costs of searching for partners. Being explicit about the biological features that can lead to positive versus negative frequency dependence at recognition loci is important, and can inform empirical work.

Reviewer #2 (Remarks to the Author):

This is a fantastic paper. It presents new theory on selection for tag-based altruism and a resolution for Crozier's paradox, a longstanding dilemma in social evolution theory. Crozier's paradox states that though genetic cues or tags are used to identify relatives, using them in this way is unstable, because common tags have higher fitness than rare ones. This comes about because common cues will match more often and get more kin altruism. Thus, theory tends to say that tag-based kin recognition should be absent or uncommon, but we have many empirical examples. The authors' theoretical innovation is very simple in principle; allow mismatched individuals (mostly rare tags) to break off and search further for a matching partner. This turns out to stabilize genetic kin recognition, thus solving an important problem. The paper is also very clearly written.

The mathematical theory is of course more daunting and most of it is relegated to a very long supplement. My initial reaction as a reviewer was that a 75-page supplement was going to be too much. But on reading it, I changed my mind. It is a superb exposition of the models, extremely clear in its goals, assumptions, methods, and conclusions. While not every reader will want to dig through the detailed models, this way of writing it up maximizes the audience that could understand it if they put in the effort. I only wish that all theoretical papers were written like this.

Thank you very much for your kind words.

My only significant comment is that I'd like to see a little more clearly how different values of alpha affect outcomes. In Fig. 4a, having a very high alpha of 0.975 already means a significant portion of the area where kin disclination is favored does not support stable recognition. So what about lower values? Figure 4d gives us an idea about this, but for probability of interaction rather than alpha. Going back to 4c gives us the relationship between those two variables, and together c and d seems to confirm that for tag frequency near zero, you need very high alphas. So, does that mean, for lower alphas, that low tag frequencies (high tag numbers) won't be stable, but higher frequencies (lower tag numbers) will be? A little more discussion of this point would be helpful particularly since you consider only $\alpha=1$ in the detailed model. Alpha is the parameter that makes your model work where other models did not, and yet it seems to be the most poorly explored parameter.

We see that we went through the analysis of the encounter parameter (α) too quickly, so we have added more explanation and new results to the main text (lines 203-232).

Your suspicion that, as α is reduced from 1, the stability of genetic kin recognition quickly falls off, is correct. We have added in this specific plot (Fig. 4c), and we now explicitly state this result in the text (lines 203-204). However, this does not mean that genetic kin recognition is only likely to be stable in extreme cases, and we list two reasons for this.

The first reason is that we need to think about what our mathematical parameter α represents biologically. One way to think about it is that it gives a proxy for what the likelihood is of socially interacting, per opportunity to socially interact. We plotted this relationship between α and the probability of socially interacting, for different tag frequencies. This shows that the probability of socially interacting falls off as α decreases, and this fall-off is steeper for individuals that are using rarer tags.

We then focus on the relationship between α and the social interaction probability that is obtained when individuals are using a limitingly rare tag (i.e. the line labelled “~0” in Fig. 4d). We can then ask, how does the stability of genetic kin recognition vary as the probability of social interaction, for an individual using a limitingly rare tag, varies. This is what we plot in Fig. 4e, and this is the key result, which shows that, on a biological interpretation of our mathematical parameter α , kin discrimination based on genetic cues is likely to be stable, unless the opportunity to socially interact when an individual has a rare tag is heavily diminished. We have gone through these steps more slowly now, which should hopefully make things a bit clearer.

You also ask, “So, does that mean, for lower alphas, that low tag frequencies (high tag numbers) won’t be stable, but higher frequencies (lower tag numbers) will be?” This is sometimes the case. But more often, for lower alphas, one single tag runs all the way to fixation, meaning no tags are maintained.

The second reason why it is not the case that genetic kin recognition is only likely to be stable in extreme cases, is that the results of our mathematical model are based on the case where selection on social behaviour (magnitude of b and c) is weak. When the strength of selection is increased, it becomes more likely that genetic kin recognition can be stable for lower values of α . We have added discussion of this point to lines 222-232, and a figure (Fig. 4f).

I don’t really like “networks” in the title. Networks are not the key to the solution of the problem and the word is hardly mentioned in the text. It will also lead readers into expecting a kind of model that this is not, with emphasis on nodes and edges etc. The real key to this model is partner searching or partner choice (though based on tags rather than past behavior). Shouldn’t that be highlighted in the title.?

We have taken your advice and changed the title to: “Multiple social encounters can eliminate Crozier’s paradox and stabilise genetic kin recognition”.

line 75. Later on I found that I had forgotten the important distinction here between encounters and interactions. It might help to insert here something along the lines of the following piece from the supplement "individuals can engage in sufficiently many social encounters before committing to a given social interaction".

We have inserted “Individuals can potentially have many social encounters before committing to a given social interaction.”

105. What does the assumption of minimal diversity mean? Truly minimal would be no diversity.

This is true. Specifically, we assumed that one tag had an initial frequency of 0.9, and all remaining tags had a frequency of approximately $0.1 / (L_{max}-1)$. This is explained in the SI, but in the main text we have changed “minimal” to “low” tag diversity.

SUPPLEMENT

p 1 bottom. “Partner recycling” seems a confusing term for this process. It implies a partner is going to be used again, which is not true here, but is true in standard partner-choice models where a partner is kept if it shows the right behavior. Consider alternative like “partner

search”, “tag search” or “tag choice”.

We have removed all mention of “partner recycling” from the main text and supplementary information, and have used alternatives like “partner search”, as you suggest.

p 5 par 3. I have trouble figuring out how search costs work. Are you making assumptions here for mathematical convenience or to match biology?

Our previous generational search cost function was an approximation, taken for mathematical convenience. However, we have opted to change this function, so that now, c_{search} straightforwardly refers to the cost of abandoning one social partner and re-associating for a new social encounter. This results in a more complicated generational search cost function: $\frac{\alpha(1-F)(1-x_i)}{1-\alpha(1-F)(1-x_i)} c_{search}$. In this function, $\frac{\alpha(1-F)(1-x_i)}{1-\alpha(1-F)(1-x_i)}$ gives the number of partner re-associations per generation. The mathematical derivation of $\frac{\alpha(1-F)(1-x_i)}{1-\alpha(1-F)(1-x_i)}$ is now given explicitly in the SI (page 13). We hope this clarifies how the search cost works.

p 6 top. “density dependence is bland” seems on odd description

We have removed this description, which was superfluous anyway.

p. 7 par 1. The two probabilities making up the cost are given in one order verbally and in the opposite order in the math which is unnecessarily confusing.

We have removed the “simple model” from the SI, which means that the text / maths you are referring to no longer features.

eqns 1 and 2. I’m having trouble seeing that how the denominator of the fractions work. Is there a series argument here?

Added later. I get this now after reading your later explanation of equation 25. So maybe some of that explanation needs to be moved up to here. But I would also more explicitly add, because this is what I was failing to see, that this same probability applies to every step of the search process – even as the numbers of new interactions decline, the probability stays the same.

We have added the text to the SI (page 11), which hopefully clarifies: “Because newly encountered partners are chosen with replacement of individuals that were previously encountered during the social search, the probability of abandoning a given partner is the same, no matter how many encounters the focal individual has already had that generation.” The series argument is also now made explicit in the derivation of the generational partner search cost in the SI (page 13).

eqns 1 and 2. Doesn’t your A term imply that the fitness components like b and c effectively vary over the generations as the number of altruistic interactions changes? Normally b and c are defined in terms of relative fitness and they go into the calculation of mean fitness, so mean fitness changes while b and c remain constant. Here it seems the reverse, mean fitness stays constant, implying that b and c change? I guess it boils down to I’m thinking about hard

selection (of the global population) and you are modeling soft selection. It might be worth pointing out.

In our model, b & c do indeed stay fixed, as is standard in social evolution models, as you rightly say. Mean fitness stays constant at 1, as is standard in population genetic models where population size is constant. The two things are consistent. Specifically, any increase in mean fitness caused by cooperative interactions is exactly countered by increased competition. In other words, the “A term” captures the effect of competition on fitness. The “A term” is a function of, amongst other things, the number of social interactions that have taken place across the population that generation. The A term accordingly varies each generation such that mean fitness remains at 1 (in other words, the A term is not a constant). We hope it is now clear how it is both true that b & c are constant and mean fitness is 1.

We note also that, in our model, because population size is constant, absolute fitness (defined as the number of offspring that survive through one iteration of the lifecycle) and relative fitness (defined as absolute fitness divided by mean absolute fitness) are identical quantities. Our approach is therefore fully consistent with social evolution approaches where “ b and c are defined in terms of relative fitness”. We have added some extra explanation of these issues to the SI (bottom of page 8).

p 13, an aside regarding Levin & Grafen. You cite reference 42 for both the old and the newer way of calculating inclusive fitness. Should it be two different references?

Yes, there are two references – the old way of calculating inclusive fitness is the original Hamilton (1964) calculation of inclusive fitness. This Hamilton (1964) reference was already there, but it was written inline, so you may have missed it! We have moved the reference to the end of the sentence so it stands out more.

Fig. 5 It’s not clear to me what the blue text about α and c_{search} is saying about the figure.

We have removed the “simple model” from the SI, which means that the figure you are referring to no longer features.

eqn 17 and below. So, your number of tags is not the countable number of tags but a sort of effective tag number based on frequencies?

Yes, that is right. We have added the following text to the SI (bottom of page 28): “We note that this metric is not the countable number of tags – such a measure would be misleading, because it would give equal weight to tags that are limitingly rare and exceedingly common. Rather, this metric is an effective tag number based on tag frequencies.”

p 33 par 6 add a reference to Fig S3 when you mention being stretched on the y-axis?

We have removed the “simple model” from the SI, which means that the text you are referring to no longer features.

Figure 6. Nice figure, but you do not actually refer to it very much in the text. Do you want to

point out any more features of these results more explicitly?

As above.

p 41 bottom. Fix “even when indiscriminate cooperation is favored over indiscriminate defection”

As above.

p 42 par 1 “in our simple model, density-dependent population regulation was global as opposed to local (hard selection)”. It’s not clear if the “hard selection” is referring to global or local. But the description does not cord with my understanding of hard and soft selection. Hard and soft is not fundamentally about local vs global but an apply to either level. At the global level, I think your model is soft selection, as mean absolute fitness is fixed at 1 and is independent of allele frequency. Your social groups (interacting pairs) are under local hard selection because their contribution to the global pool is proportional to their mean fitness. This comment doesn’t affect the results of course.

We have removed all reference to hard / soft jargon completely to minimise unnecessary confusion.

p 45 par 4. Are you really not going to show results for $c_{search} \neq 0$? See my comment on alpha at the beginning. These are the two parameters that matter most for you model.

As detailed above, we have added c_{search} to the island model (SI pages 13-15, 31 & interspersed text) and given these results in the main text (under the “Encounter rate and search cost” heading on p11, & Fig. 5).

p 45 last par. You define Mhelped, but I think you use Mind helped in the text that follows.

Thanks for catching this typo – amended.

p 57 par 1 I note that Fig. 11 is referenced here, before Fig 10 is first referenced.

Amended.

p 64 par 2. “and this will require a low cost of kin recognition (high α)” – is “high α ” what you mean here?

We have amended the text slightly to now read “and this will require a low cost of kin recognition (high α & low c_{search})”. The “cost of kin recognition” refers to the costs of reduced interaction rate (α) and increased investment in partner search (c_{search}).

p 64 par 4. “If the evolutionary constraint on tag diversity is relaxed completely ($L_{max} \rightarrow \infty$), then genetic kin recognition may evolve (Equation 43 satisfied) across most of the parameter space (i.e. across most values of m , N and r).” Do you mean for most values of m , N , and r , for which inclusive fitness predicts cooperation?

No, we mean most of the total parameter space – i.e. most combinations of m , N and r . This is because, if infinitely many tags are (hypothetically) segregating, then individuals will be able to pick out kin with no mistakes (highly precise kin recognition). In this scenario, conditional helping is nearly always favoured over defection, as long as $b > c$.

p 65 par 2. Is this agent-based model identical to your island model, other than being agent-based?

The differences are: (1) The mathematical model assumes weak selection (low magnitude of c , b , c_{search}), whereas the agent-based model permits stronger selection. (2) The mathematical model assumes that there is no stochastic variation in the genetic composition of demes, which is a reasonable assumption if deme size (N) is large. The agent-based model accounts for this stochastic variation, so is accurate for any deme size (N). We have emphasised these differences in SI (pages 31-34).

p 65 par 5-6 I'm not seeing the connection between these two paragraphs. the first says that conditional altruism will fall to close to zero (but held above by mutation). This will maintain rare tags. The second paragraph has those rare tags and conditional altruism being favored, the latter to fixation. What specifically has changed to cause conditional altruism to switch from being selected against to being selected to fixation?

(N.B. We assume you were referring to p68 rather than p65.) In the absence of trait mutation, the conditional helping allele may go to a population frequency of zero. However, in the presence of trait mutation, the conditional helping allele will not go to zero; rather, it will go to a mutation-selection balance equilibrium. Trait mutation therefore ensures that there is always trait variation in the population. And as long as there is trait variation, rare tags can gain an advantage over common tags, because they will be in linkage disequilibrium with the conditional helping allele.

This is why, in some cases, tag diversity is lost when there is no trait mutation, but sustained when there is trait mutation. The difference is that, when there is no trait mutation, trait variation may be completely lost (i.e. all individuals become defectors), which removes any benefit of tag rarity, preventing tags from equalising in frequency. However, when there is trait mutation, trait variation is never lost, meaning rare tags can always gain an advantage (more cooperative interactions) – if this advantage outweighs the costs of tag rarity (reduced interaction rate or costlier partner search), tags will begin to equalise in frequency. Eventually, tag equalisation will bring the previously-common tags to a low enough frequency that altruism is selected, even amongst individuals bearing these tags. At this point, altruism is universally favoured, resulting in high conditional altruism alongside stable tag diversity.

So to directly answer your question “what specifically has changed to cause conditional altruism to switch from being selected against to being selected to fixation?”, the relative frequencies of different tags has changed. In the no-trait-mutation scenario, the common tag never gets pulled to a low enough frequency that cooperation is favoured amongst individuals bearing this tag. In the trait mutation scenario, long term trait variation gives rare tags a long term advantage, pulling the common tag to a lower and lower frequency until altruism is universally favoured.

We hope that this clarifies things – we can appreciate that the effect of trait mutation is very subtle. We have added in some more explanation of this process to the SI (page 37), to hopefully make it clearer.

Reviewer #3 (Remarks to the Author):

The manuscript is very interesting question, though I have a few comments.

1. Some researchers argue that there are no examples of genetic kin recognition, including one of the authors (Grafen 1990 argued that the only evidence comes from one species of marine invertebrates, *Botryllus schlosseri*), and yet this manuscript ignores this issue.

If there are good examples in the literature, then these should be addressed, but if genetic kin recognition does not exist or there are few or no good examples, then the point of such a model is unclear – and its importance is dubious.

Since Grafen's 1990 review of genetic kin recognition, many instances of genetic kin recognition have been discovered, and these are cited in the main text (lines 45-46).

2. The introduction (Crozier's Paradox) summarizes the theoretical models that have addressed this topic, but it does not accurately summarize the field or previous theoretical work.

Contrary to what the authors assert, it is not widely accepted that kin recognition via genetic cues is evolutionarily unstable (p.2, line 29), as papers cited and the authors themselves have helped to keep this hypothesis viable. So, this assertion is inaccurate, which can be easily shown by any review on this topic.

The authors state that previous models find that genetic kin recognition is only stable under 'very restrictive conditions' (p. 2, line 39; line 227). The model by Rousset and Roze (2007), which is cited, found that kin recognition can be maintained given certain assumptions, i.e., spatial population structure (from low dispersal) and recombination between matching and helping loci, and negative-frequency dependent selection from parasites, but these are realistic, not unrealistically restrictive. It is unclear which aspects of this and other models are unrealistic or overly restrictive.

Rousset & Roze (2007) found that, under conditions of low migration ($m < 0.1$), tight linkage ($r < 0.1$), and strong selection (high magnitude of b and c), two recognition alleles (tags) can be maintained stably alongside low / intermediate frequency of a conditional cooperation allele. For any more than two tags to be maintained, there also needs to be tag mutation (i.e. tag diversity of more than 2 tags cannot be maintained by selection on social behaviour alone).

These requirements, in our view, are indeed restrictive, and this perspective was also conveyed in Rousset & Roze (2007) themselves, who say that: “in some cases, the population evolves toward an equilibrium where both the helping and matching locus are polymorphic. However, this only occurs under rather restrictive conditions”; “a stable polymorphism seems possible only under a restricted set of parameters and initial conditions. In most biological settings, nonhelper mutants at additional unlinked ($r = 0.5$) loci would be expected to destabilize any preexisting polymorphism not maintained by extrinsic forces”; “selection alone appears generally insufficient to maintain polymorphism at the recognition locus [although a low tag mutation rate may stabilise it]”.

Of course, it is true that Rousset & Roze (2007) found that genetic kin recognition can be stabilised if there exists extrinsic balancing selection due to an

additional role of recognition alleles in e.g. parasite resistance. We have therefore changed the sentence “It has become widely accepted that kin recognition via genetic cues is not usually evolutionarily stable”, and added “, except when genetic cues are maintained for some reason unrelated to social behaviour.” We have also changed “very restrictive” to “restrictive”, which is how Rousset & Roze (2007) described their own results.

More generally, we have described in detail why previous demonstrations of the evolution of kin discrimination based on genetic cues are unrealistic / restrictive, both in the SI (pages 38-39) and main text (under the “Alternative scenarios and genetic architecture” heading on p16).

Grafen (1990) proposed that the main problem with Crozier's Paradox is that it assumes that there are no cheats, but this issue is buried in the supplement.

We have added “Crozier’s original statement of the paradox did not permit cooperative cheats, meaning this coevolution between cooperation and kin recognition could not be captured” to the main text (lines 148-150).

3. The assumptions of the authors' model allow multiple encounters, which is expected with realistic spatial populations structure, which is in line with previous models.

Previous models have indeed permitted multiple encounters, but in all previous models, partners cannot be abandoned in favour of new encounters. In all previous models, therefore, social interaction rate is determined by the frequency of an individual’s tag (our $\alpha=0$ case). Our model is novel because it permits multiple social encounters *before each social interaction* (partner search), meaning social interaction rate is not necessarily fully determined by the frequency of an individual’s tag. We see that we have not always included the “before each social interaction” qualification when talking about multiple social encounters. We have now amended this in the text.

4. The authors assert that their conclusions are robust to several It is crucial to show that tag diversity is not eroded by genetic drift (p 7, line 151), and this issue is too important for the supplements and should be addressed in the manuscript. Also, the paper should address how robust their results are against these assumptions.

We had already shown, using an agent-based finite-population model, that tag diversity can be maintained in a finite population, implying that balancing selection on tags is overriding genetic drift (SI pages 31-32). However, to avoid any doubt that balancing selection is in operation and overrides drift, we have extended this analysis, adding the new section “Balancing selection in the finite population model” to the SI (pages 32-34).

In this new section, we run a version of our agent-based simulation, in which there is no tag mutation, for different parameter combinations. Tag diversity is eventually lost due to drift (this is inevitable in a finite population, if the simulation is run for long enough, even under balancing selection). We record this time taken for tag diversity to be lost (fixation time), and divide it by the time taken for tag diversity to be lost in the corresponding neutral scenario (i.e. where coefficients of selection are set to zero). We show that, in certain regions of parameter space, this ratio of fixation times is positive (i.e. fixation time is greater when there is selection) and increases with population size. As we explain in the SI, these properties provide strong evidence that balancing selection is in operation, and overrides genetic drift, allowing tag diversity to

be maintained. We also demonstrate, using an analogous approach, that balancing selection is more likely to override drift, sustaining tag diversity, when: partner search (α) is higher, partner search is less costly (c_{search}), and the strength of selection on social behaviour (b,c) is increased (this last result recovers an important finding of Rousset & Roze (2007)).

The methodological approach that we used to show that balancing selection on tags overrides genetic drift closely follows one utilised by Rousset & Roze (2007). The methodology and results are elaborated in more detail in the SI (pages 32-34), and the results are also conveyed in the main text (lines 229-231 & Fig. 4f).

5. The authors state that their model makes only conservative assumptions (p 13, line 230), but they do not explain why -- or why they are more conservative than previous models.

We do explain why – there is a reference in the main text to a section of the SI, in which we discuss these assumptions in detail – indeed, in too much detail for inclusion in the main text.

6. The main question here is: 'can genetic recognition evolve with fewer assumptions of previous models?' The authors suggest that host-parasite interactions, the solution usually cited to solve Crozier's Paradox, 'may be a red herring' (p 13, line 237). If so, then the authors should clarify why their model assumptions are not only simpler but also more realistic, and acknowledge that their model is not a mutually exclusive alternative.

We have added a detailed discussion of this to both the main text (lines 292-312) and SI (pages 38-39). In particular, we argue that host-parasite interactions is an incomplete theory, because it gives no account of why a locus under extrinsic balancing selection should be “chosen” as the kin recognition locus. In this sense, it may not be a mutually exclusive alternative to the solution presented here, but more theoretical work is needed to evaluate this properly.

This is in addition to the discussion in the SI (page 38) and main text (lines 280-291) that explains why our model assumptions are simpler and more realistic than competing hypotheses that don't rely on extrinsic balancing selection (i.e. our solution is the only one that works: (i) under weak selection; (ii) without tag mutation; (iii) for migration and recombination values that aren't unrealistically small).

7. There are many terms that are not defined in the manuscript, such as 'kin recognition', 'pedigree relatedness', etc, and it is critical to define terms because they are used differently by different researchers. e.g. Grafen (1990) was criticized for generating semantic confusion over the term 'kin recognition'. The authors contrast genetic kin recognition with environmentally determined phenotypic tags, but then use an example of 'grew up in the same nest' (p. 2, line 41), which is not a phenotypic label. There are reviews that have addressed these semantic issues, but they are not cited.

We have added the text “Individuals are therefore expected to evolve *kin discrimination*, which is the conditional helping of relatives that are identified (*kin recognition*) through either genetic or environmental cues” to the introduction of the main text.

We now introduce pedigree relatedness in the following context (lines 142-143), to emphasise that we just mean an individual's common ancestry: “As tags become more common, they will become less useful cues of the individual's common ancestry (pedigree relatedness; Supp. Info. 3e *iii*)”. We have also added the reference to Supp.

Info. 3e *iii*, in which pedigree relatedness, and how this differs from other causes of genetic relatedness, is discussed at length.

We have changed our “grew up in the same nest” example to “a song learnt from relatives”. This addresses each of the reviewer’s semantic concerns.

8. Fig 1 is unnecessary. The point is rather simple and does not need a figure (and this one does not really help).

We agree that the figure captures a simple point, but in our experience, the figure helps with explaining the theoretical problem (Crozier’s paradox) to empiricists. We would therefore prefer to keep it, in an attempt to make the paper as accessible as possible.

9. The final paragraph is very good.

Thank you.

Reviewers' Comments:

Reviewer #1:

Remarks to the Author:

I think the manuscript is much improved, and is ready for publication pending some minor further additions/changes.

Line 127: Presumably, the R terms refer to relatedness at the help/defect locus specifically? Is it worth being more explicit here?

Line 132: typo

Line 148: Maybe delete this, or mention that many subsequent models allowed for defectors (e.g. Rousset & Roze, the "beard chromodynamics" paper, Holman et al), since some readers will interpret this to mean your model was the first to do this

Line 176-177: Presumably, this "relatedness" refers to relatedness at the help/defect locus specifically?

Line 183: "More common tags are therefore worse indicators of pedigree relatedness". This is true, but it's not as salient as saying "More common tags are therefore worse indicators of relatedness at the cooperation locus". The thing that I take from the green beard thought experiment is that genome-wide allele sharing ('pedigree relatedness') matters in social evolution because it predicts allele-sharing at the loci that govern in social behaviours. You could consider revising this diagram so that the size of the coloured area on the birds illustrates the probability that each relative carries the same allele at the cooperate/defect locus as the focal individual.

Line 215: "When searching for partners is relatively cheap (low c_{search}), and individuals are using a limitingly rare tag, the probability of social interactions often needs to drop significantly below 1.0 before genetic kin recognition is likely to be less favoured (Fig. 4e)." I (and I think other readers) wonder how this conclusion changes when c_{search} is higher. You could add 1-2 more lines to this graph, showing the effects of varying c_{search} on the threshold α . I'd guess that increasing the search costs increases the effect of declining α , since low α means the cost is paid more often. Alternatively, you might prefer to examine this interaction in Figure 5 (since in that paragraph you talk more about how α and c_{search} interact to determine the realised cost of searching for a partner).

Line 302: I was a bit puzzled by this paragraph. I think all the former authors that focused on things like MHC did not think that kin discrimination was forced to use those loci, rather that only those loci would stay variable in the face of the eroding effects of Crozier's paradox and thus be viable candidates on which to base genetic kin discrimination. Furthermore, loci under diversifying selection might tend to be hyper variable, and therefore provide more accurate information on pedigree relatedness (and hence, relatedness at helping/social behaviour loci), than some other locus that is not under diversifying selection. I agree a model would be good here, and I would be interested to see if indeed the strategy "base kin discrimination on an externally-selected locus like the MHC" beats the strategy "base kin discrimination on a locus that is otherwise neutral".

Reviewer #2:

Remarks to the Author:

I like this paper the first time around, and most of my concerns have been addressed, with one exception. My main question was how widely applicable the model is likely to be given its dependence on very high encounter rate parameters. I take your point that the analytical model assumes small

fitness effects and the model can work better with large ones. But I don't think I buy the rest of the argument. From line 217:

"Therefore, although the stability of genetic kin recognition is susceptible to a drop off in the mathematical encounter parameter (α), a biological interpretation of our mathematical parameter – as a proxy for the probability of socially interacting – implies that genetic kin recognition should evolve relatively permissively, as long as the risk of foregoing social interactions whilst using a rare tag is relatively low".

I think this is argument obscures more than clarifies. You take a parameter that is biologically easy to think about – what is the probability that you search for a new partner – and then ask us to understand it via a less clear parameter (probability of social interaction) that depends on multiples of the tag frequency. It also presumably depends on the search costs (do you die before you find a tag match?), which you have assumed to be zero. Yes, this logic can help us understand downstream effects of α but it doesn't change the fact that the α parameter has to be very large – individuals have to re-search with near certainty. And it is much easier to judge that biologically. I really think this should not be swept under the rug, but instead highlighted so readers can make their own judgments. My own guess is that there would be many biological cases that fail to get a high enough α . But the model is still interesting and applicable to some cases.

The fundamental issue is that you need an advantage for rare tags and it is really improbable to get rare tags together unless they just keep searching and searching after mismatches. I don't see any argument that gets around that.

Another point is that you assume each individual gets only one interaction. But in many cases, interactions may be brief and repeatable. So, while rare-tag individuals are desperately searching for a match, common-tag individuals could sometimes be racking up repeated benefits. I did not mention this in my first review, but you might want to address it in the final version.

Fig 5c. You only allow search cost to get as high as 0.001. Might they not be higher?

235-234. I think "total search costs" would be better than "effective search costs". I don't think it is really analogous to the "effective" in "effective population size".

Reviewer #3:

Remarks to the Author:

The authors have done a fine job of replying to my concerns, and I have only a couple of minor points.

Given the semantic confusion with this topic, it is good that they have now defined their terms. Contrary to what they state in their reply, it isn't that there is new evidence for genetic kin recognition, but rather that the authors are not using Grafen's (1990) narrow definition for 'kin recognition' (or for 'genetic kin recognition'), and they cite examples that do not meet the narrow definition.

I suggest that the authors drop the term 'red herring' (p. 18) to refer to the host-parasite interaction hypothesis. This term implies that the hypothesis is intentionally misleading, but it is not. The authors are not arguing that the hypothesis is misleading, intentional or otherwise; they are merely claiming that they are not necessary.

I suggest that the authors re-state the assumption and clarify the testable empirical predictions of their model (e.g., kin recognition will only be found in species show X number of social encounters, etc.).

REVIEWER COMMENTS

Reviewer #1 (Remarks to the Author):

I think the manuscript is much improved, and is ready for publication pending some minor further additions/changes.

Thank you!

Line 127: Presumably, the R terms refer to relatedness at the help/defect locus specifically? Is it worth being more explicit here?

Yes it does, but it is important to also emphasise that relatedness (genetic similarity) at the trait locus will often be equal to standard pedigree / genealogical relatedness, i.e. relatedness that follows from a simple reading of a family tree (as shown in Supp. 3e *iii*). It is useful to make this point, because pedigree relatedness can be easier to measure empirically than genetic similarity at a specific locus (especially given that the loci underpinning social behaviour may not be known).

We have therefore added in (line 128) the following clarificatory sentence: “Here, relatedness technically means *genetic similarity at the trait locus*, but at evolutionary equilibrium, this will usually be equal to the probability that individuals share common ancestry (pedigree / genealogical relatedness; e.g., 1/2 for full siblings, 1/8 for cousins; Supp. 3e *iii*).”

Line 132: typo

Amended

Line 148: Maybe delete this, or mention that many subsequent models allowed for defectors (e.g. Rousset & Roze, the “beard chromodynamics” paper, Holman et al), since some readers will interpret this to mean your model was the first to do this

We have added (line 153) the sentence “More recent models have permitted defectors” and cite the papers you suggest, as well as other relevant papers.

Line 176-177: Presumably, this “relatedness” refers to relatedness at the help/defect locus specifically?

Yes it does – we have now made this explicit (line 179).

Line 183: “More common tags are therefore worse indicators of pedigree relatedness”. This is true, but it’s not as salient as saying “More common tags are therefore worse indicators of relatedness at the cooperation locus”. The thing that I take from the green beard thought experiment is that genome-wide allele sharing (‘pedigree relatedness’) matters in social evolution because it predicts allele-sharing at the loci that govern in social behaviours. You could consider revising this diagram so that the size of the coloured area on the birds illustrates the probability that each relative carries the same allele at the cooperate/defect locus as the focal individual.

We have taken on board your suggestion, and now explicitly refer to relatedness at the trait locus. However, we think that it is also important to emphasise pedigree relatedness, for two reasons. Firstly, as mentioned above, pedigree relatedness is much easier to measure empirically than genetic similarity at a particular locus. Secondly, emphasising pedigree relatedness serves as a good reminder to the reader that, as shown in Supp. 3e *iii*, the evolutionary force in action here is standard kin selection (selection to help genealogical kin), rather than selfish greenbeard selection (rogue genes helping copies of themselves against the interest of the individual as a whole).

We have accordingly amended the text (line 186) so that it now reads: “More common tags are therefore worse indicators of both relatedness at the trait locus and pedigree relatedness.”

Line 215: “When searching for partners is relatively cheap (low c_{search}), and individuals are using a limitingly rare tag, the probability of social interactions often needs to drop significantly below 1.0 before genetic kin recognition is likely to be less favoured (Fig. 4e).” I (and I think other readers) wonder how this conclusion changes when c_{search} is higher. You could add 1-2 more lines to this graph, showing the effects of varying c_{search} on the threshold α . I’d guess that increasing the search costs increases the effect of declining α , since low α means the cost is paid more often. Alternatively, you might prefer to examine this interaction in Figure 5 (since in that paragraph you talk more about how α and c_{search} interact to determine the realised cost of searching for a partner).

Good suggestion – this would be useful to clarify. You say that “low alpha means the [search] cost is paid more often”, but actually the opposite is true – alpha gives the rate of partner searching, so a higher alpha leads to more searching, meaning the cost of partner search (C_{search}) is incurred more often. The result of this is that, as the partner search cost (C_{search}) increases, the proportion of parameter space where genetic kin recognition is stable decreases, but this decrease is proportionally greater if there is more partner search (higher alpha). We have illustrated this by adding in an extra line to Figure 5, as you suggest. We have also added this result into the main text, with the following sentence (line 262): “Furthermore, if the partner search rate (α) is reduced, the partner search cost (C_{search}) has even less of a destabilising effect, simply because the search cost will be paid less often (Fig. 5b).”

Line 302: I was a bit puzzled by this paragraph. I think all the former authors that focused on things like MHC did not think that kin discrimination was forced to use those loci, rather that only those loci would stay variable in the face of the eroding effects of Crozier’s paradox and thus be viable candidates on which to base genetic kin discrimination. Furthermore, loci under diversifying selection might tend to be hyper variable, and therefore provide more accurate information on pedigree relatedness (and hence, relatedness at helping/social behaviour loci), than some other locus that is not under diversifying selection. I agree a model would be good here, and I would be interested to see if indeed the strategy “base kin discrimination on an externally-selected locus like the MHC” beats the strategy “base kin discrimination on a locus that is otherwise neutral”.

We have clarified and shortened this text, so that it now (line 319) reads: “However, there are possible complications [with the hypothesis that extrinsic balancing selection stabilises genetic kin recognition], such as host-parasite coevolution leading to fluctuating allele frequencies, and it has yet to be shown that natural selection would ‘choose’ a locus under diversifying selection for a kin recognition tag, as opposed to a locus that was otherwise neutral.”

Reviewer #2 (Remarks to the Author):

I like this paper the first time around, and most of my concerns have been addressed, with one exception.

Many thanks!

My main question was how widely applicable the model is likely to be given its dependence on very high encounter rate parameters. I take your point that the analytical model assumes small fitness effects and the model can work better with large ones. But I don’t think I buy the rest of the argument. From line 217:

“Therefore, although the stability of genetic kin recognition is susceptible to a drop off in the mathematical encounter parameter (α), a biological interpretation of our mathematical parameter – as a proxy for the probability of socially interacting – implies that genetic kin recognition should evolve relatively permissively, as long as the risk of foregoing social interactions whilst using a rare tag is relatively low”.

I think this is argument obscures more than clarifies. You take a parameter that is biologically easy to think about – what is the probability that you search for a new partner – and then ask us to understand it via a less clear parameter (probability of social interaction) that depends on multiples of the tag frequency. It also presumably depends on the search costs (do you die before you find a tag match?), which you have assumed to be zero. Yes, this logic can help us understand downstream effects of α but it doesn’t change the fact that the α parameter has to be very large – individuals have to re-search with near certainty. And it is much easier to judge that biologically. I really think this should not be swept under the rug, but instead highlighted so readers can make their own judgments. My own guess is that there would be many biological cases that fail to get a high enough α . But the model is still interesting and applicable to some cases.

The fundamental issue is that you need an advantage for rare tags and it is really improbable to get rare tags together unless they just keep searching and searching after mismatches. I don’t see any argument that gets around that.

We see what you mean here. It is true that a high alpha is required for genetic kin recognition to evolve, when selection is weak. Our argument, however, is that the key issue is that rare tags gain their advantage by socially interacting at a comparable rate to common tags, and in the context of our model, searching randomly (with replacement) for social partners is the way that this can be achieved. But in nature, partner search is likely to be more streamlined than this.

Firstly, in nature, partner search may not occur with *replacement* of social encounters. Rather, individuals may improve the efficiency of the social search by remembering previous failed social encounters, and focusing the social search on individuals that haven’t been ‘tested’ yet (this point was

made in the context of our model potentially reflecting an unfavourable scenario for the evolution of genetic kin recognition, on line 308 and Supp. 3e ii).

Secondly, in nature, partner search may work in conjunction with environmental cues, such that the efficiency of the social search is increased. For instance, individuals may narrow the social search so that only individuals who sing the same learnt song as themselves are 'tested'. This may mean that rare tags can achieve a comparable rate of social interaction as common tags, without having to engage in as much partner search.

So although our requirement of high alpha does make it seem like genetic kin recognition may not evolve permissively, the point we are trying to make here is that our model is only capturing one very specific (and very inefficient!) way of finding a tag-matched social partner, and we shouldn't necessarily take this as evidence that genetic kin recognition can only evolve in extreme cases. We see that we are technically speculating outside of the realm of what our mathematical model shows, but we feel that this is necessary in order to give a more complete biological picture. In other words, we are trying not to be too literal-minded when drawing conclusions from our model, which is (necessarily) very specific and simplified.

Another way of thinking about this is that alpha is a nice a simple clear parameter in our model, but it is the probability of finding a relative with the same tag that really matters for natural selection (which could depend upon other factors).

To address this problem we have expanded and clarified both these points. We have specified (line 207) that "Consequently, a high value of this encounter parameter (α) is required for genetic kin recognition to evolve." We have expanded and clarified our discussion of why we also examine the probability of finding a relative with the same tag. We have also changed (line 233) the "should" to "could", in recognition that we are technically speculating outside of the remit of the model: "a biological interpretation of our mathematical parameter... implies that genetic kin recognition should evolve relatively permissively, as long as the risk of foregoing social interactions whilst using a rare tag is relatively low".

Another point is that you assume each individual gets only one interaction. But in many cases, interactions may be brief and repeatable. So, while rare-tag individuals are desperately searching for a match, common-tag individuals could sometimes be racking up repeated benefits. I did not mention this in my first review, but you might want to address it in the final version.

This is true, and will undoubtedly often be the case empirically. However, if common-tag individuals are interacting many times per generation, and rare-tag individuals are interacting fewer times per generation, this will be highly analogous to the "low alpha" scenario covered by our models (the only difference would be that, with multiple opportunities to socially interact per generation, social interactions would occur at a greater rate relative to each round of recombination and mutation, but this should not appreciably impact the end point of adaptive evolution). We therefore suspect that our failure to allow multiple interactions per generation is not a particular limitation of the model.

Fig 5c. You only allow search cost to get as high as 0.001. Might they not be higher?

They might, yes. We didn't allow it to be higher than this, because it sometimes results in negative fitness, meaning our model is undefined. Nevertheless, it is true that excessively high partner search cost will destabilise genetic kin recognition, so we have added (line 264) the following text: "However, if the search cost (C_{search}) is increased high enough, kin recognition will eventually be destabilised".

235-234. I think "total search costs" would be better than "effective search costs". I don't think it is really analogous to the "effective" in "effective population size".

Yes, you are right – amended.

Reviewer #3 (Remarks to the Author):

The authors have done a fine job of replying to my concerns, and I have only a couple of minor points.

Given the semantic confusion with this topic, it is good that they have now defined their terms. Contrary to what they state in their reply, it isn't that there is new evidence for genetic kin recognition, but rather that the authors are not using Grafen's (1990) narrow definition for 'kin recognition' (or for 'genetic kin recognition'), and they cite examples that do not meet the narrow definition.

This is a fair point. We are happy to use the broader definition in this case, which we feel is more in line with most current usage.

I suggest that the authors drop the term 'red herring' (p. 18) to refer to the host-parasite interaction hypothesis. This term implies that the hypothesis is intentionally misleading, but it is not. The authors are not arguing that the hypothesis is misleading, intentional or otherwise; they are merely claiming that they are not necessary.

We take this point, and have removed “red herring”. We now simply state that the hunt for alternative factors to maintain tag diversity may “not be necessary”.

I suggest that the authors re-state the assumption and clarify the testable empirical predictions of their model (e.g., kin recognition will only be found in species show X number of social encounters, etc.).

We already have the following passage (line 329) in the final section of the text: “our theory emphasises the need to measure the frequency with which individuals encounter other individuals to allow them a reasonable chance of encountering one with the same tag (Fig. 4d). When this frequency is high, corresponding to a high α , genetic kin recognition is more likely to be stable.” Given the (necessarily) highly specific nature of our model, we would be wary of making more quantitative predictions, in particular to do with the specific number of social encounters necessary to favour genetic kin recognition. We envisage that our theory would be more appropriately tested through comparative analysis, in which the value of alpha in natural populations is used to predict the occurrence of genetic kin recognition in natural populations (i.e., we would expect a correlation between alpha and kin recognition, but we would be reluctant to predict, on the basis of our model, a threshold alpha after which kin recognition is favoured).

Reviewers' Comments:

Reviewer #1:

Remarks to the Author:

I am happy with the authors' responses to the previous review, and can now recommend publication.

Reviewer #2:

Remarks to the Author:

With the changes made in the current version of the ms, I fully support publication.

Reviewer #3:

Remarks to the Author:

The authors have addressed my criticisms (and also the comments of the other referees).

REVIEWERS' COMMENTS

Reviewer #1 (Remarks to the Author):

I am happy with the authors' responses to the previous review, and can now recommend publication.

Reviewer #2 (Remarks to the Author):

With the changes made in the current version of the ms, I fully support publication.

Reviewer #3 (Remarks to the Author): _____

The authors have addressed my criticisms (and also the comments of the other referees).

We are pleased that the reviewers are satisfied with our changes, and we thank them for a constructive peer review process.

We provide a brief summary of the main findings of the paper, as requested in the author checklist: “Crozier’s paradox suggests that genetic kin recognition will not be evolutionarily stable. Here, the authors show that allowing for multiple social encounters before each social interaction can eliminate Crozier’s paradox and stabilise genetic kin recognition.”